# Stochastic Primal-Dual Double Block-Coordinate for Two-way Partial AUC Maximization

**Linli Zhou**                                                                    *lynn94874@tamu.edu*
*Department of Computer Science and Engineering*
*Texas A&M University*

**Bokun Wang**                                                              *bokun.wang@utexas.edu*
*Department of Electrical and Computer Engineering*
*University of Texas, Austin*

**My T. Thai**                                                                      *mythai@cise.ufl.edu*
*Department of Computer and Information Science and Engineering*
*University of Florida*

**Tianbao Yang**                                                          *tianbao-yang@tamu.edu*
*Department of Computer Science and Engineering*
*Texas A&M University*

**Reviewed on OpenReview:** *https://openreview.net/forum?id=M3kibBFP4q*

## Abstract

Two-way partial AUC (TPAUC) is a critical performance metric for binary classification with imbalanced data, as it focuses on specific ranges of the true positive rate (TPR) and false positive rate (FPR). However, stochastic algorithms for TPAUC optimization remain under-explored, with existing methods either limited to approximated TPAUC loss functions or burdened by sub-optimal complexities. To overcome these limitations, we introduce two innovative stochastic primal-dual double block-coordinate algorithms for TPAUC maximization. These algorithms utilize stochastic block-coordinate updates for both the primal and dual variables, catering to both convex and non-convex settings. We provide theoretical convergence rate analyses, demonstrating significant improvements over prior approaches. Our experimental results, based on multiple benchmark datasets, validate the superior performance of our algorithms, showcasing faster convergence and better generalization. This work advances the state of the art in TPAUC optimization and offers practical tools for real-world machine learning applications.

## 1 Introduction

The area under the ROC curve, commonly referred to as AUC, is frequently utilized as a measure of the model's classification ability, without the explicit setting of a threshold. With a long history dating back to the late 90s (Herbrich et al., 1999), AUC is acknowledged as a more informative metric than accuracy for assessing the performance of binary classifiers in the context of imbalanced data and widely used in machine learning.

In many applications, there are large monetary costs due to high false positive rates (FPR) and low true positive rates (TPR), e.g., in medical diagnosis. Hence, a measure of primary interest is the region of the ROC curve corresponding to low FPR and high TPR, i.e., $\text{TPR} \geq 1 - \theta_0$, $\text{FPR} \leq \theta_1$, for some $\theta_0, \theta_1 \in (0, 1)$, which is referred to as two-way partial AUC (TPAUC). Nevertheless, research on efficient optimization algorithms to optimize TPAUC for learning a classifier remains underdeveloped.

Table 1: Comparison with prior works for optimizing the TPAUC loss, where $n_+$ is the number of positive examples, $S$ is the mini-batch size of positive examples, $B$ is the mini-batch size of negative examples, and $d$ is the dimension of the model parameter

| Method | Convexity | Loop | Iteration Complexity | Total Complexity |
|---|---|---|---|---|
| SONX (Hu et al., 2023) | non-convex | Single | $\mathcal{O}((B+S)d)$ | $\mathcal{O}\big(\frac{n_+}{B^{1/2}S\epsilon^6}\big)$ |
| SOTA (Zhu et al., 2022) | non-convex | Double | $\mathcal{O}((B+S)d + n_+)$ | $\mathcal{O}\big(\frac{n_+}{\epsilon^6}\big)$ |
| STACO1 (Ours) | convex | Single | $\mathcal{O}((B+S)d)$ | $\mathcal{O}\big(\frac{n_+}{S\epsilon^2}\big)$ |
| STACO2 (Ours) | non-convex | Double | $\mathcal{O}((B+S)d)$ | $\mathcal{O}\big(\frac{n_+}{BS\epsilon^6}\big)$ |

Compared with standard AUC maximization, optimizing TPAUC presents several unique technical challenges. First, the estimator of TPAUC requires selecting subsets of positives and negatives in the top and bottom ranks. Some earlier works have proposed heuristic approaches for TPAUC maximization, including selecting examples based on their ranks in the mini-batch or converting data selection into ad-hoc data weighting (Yang et al., 2021; Kar et al., 2014), which do not provide a guarantee of optimizing TPAUC losses.

Recently, Zhu et al. (2022) have initiated rigorous optimization of TPAUC losses. They converted data selection in top/bottom ranks into pairwise loss selection and reformulated it using the tool of distributionally robust optimization. They have proposed two algorithms for two different formulations: SOTAs for solving a smooth coupled compositional objective that corresponds to a soft TPAUC loss and SOTA for solving a non-smooth min-max objective that corresponds to an exact TPAUC loss. Nevertheless, SOTAs is not for optimizing the exact TPAUC loss, and SOTA is inefficient for large datasets as it requires updating all coordinates of an auxiliary variable corresponding to all positive examples at every iteration. Additionally, its convergence rate analysis fails to demonstrate any mini-batch speedup.

Hu et al. (2023) has developed an algorithm for solving non-convex non-smooth coupled compositional objective of the exact TPAUC loss as formulated in (Zhu et al., 2022). However, their method cannot achieve linear speedup in terms of the mini-batch size of negative examples. In addition, since their method does not exploit convexity, its convergence guarantee still exhibits a complexity of $O(1/\epsilon^6)$ even in the convex setting.

To overcome these difficulties, this paper proposes improved algorithms and analysis over SOTA for solving the non-smooth min-max objective of the exact TPAUC loss. Our key idea is to design stochastic double block-coordinate updates that simultaneously act on both primal and dual variables. We propose two methods: STACO1 for convex objectives and STACO2 for non-convex objectives. Our convergence analysis introduces novel techniques for handling non-bilinear min-max objectives with stochastic block-coordinate updates, establishing state-of-the-art complexity bounds. Our algorithms enable scalable updates and provable mini-batch parallel speedup. We compare our results with prior works in Table 1.

We summarize the main contributions of our work below:

- We propose novel primal-dual double block-coordinate algorithms STACO (**S**tochastic **T**wo-way partial **A**UC block-**C**ordinate **O**ptimizer) designed for convex functions (STACO1) and non-convex functions (STACO2). These algorithms leverage double block-coordinate updates for both the primal and dual variables.

- We provide a novel convergence analysis of STACO1 for convex functions. To the best of our knowledge, this is the first work to analyze double block-coordinate updates for both primal and dual variables for min-max optimization without a bilinear structure. We extend this analysis to STACO2 for non-convex cases, demonstrating its ability to find (nearly) stationary solutions. We demonstrate our algorithm enjoys better convergence rate than existing results Hu et al. (2023); Zhu et al. (2022) by improving the block-size dependency, achieving full mini-batch speedup and time efficiency.

- We conduct comprehensive experiments on both linear and deep models for image classification and graph classification tasks involving imbalanced data. Our algorithms consistently demonstrate better performance compared to existing TPAUC maximization methods and various baselines. Additionally, we perform ablation studies to verify the improved convergence rates of our methods.

## 2 Related Work

**Two-way Partial AUC (TPAUC).** AUC has been studied for more than two decades (Hanley & McNeil, 1982), and a huge amount of work has been devoted to AUC maximization (Yang & Ying, 2022). Compared to AUC maximization, two-way partial AUC (TPAUC) maximization is much more challenging due to that it involves the selection of examples whose prediction scores are in a certain range. Recently, studies on TPAUC have emerged, as researchers have argued that for certain tasks, only the TPR or FPR within a specific range is of interest (Narasimhan & Agarwal, 2013; Yang et al., 2019; Yuan et al., 2021a; Zhu et al., 2022; Xie et al., 2024). In particular, by replacing TPR and FPR with surrogate losses, TPAUC maximization problem can be further transformed into coupled compositional optimization and min-max optimization (Zhu et al., 2022). Some other works are also focusing on TPAUC (Zhang et al., 2023; Shao et al., 2023; Yang et al., 2023b; 2022; Shao et al., 2022). Zhang et al. (2023) focuses on optimizing a compositional formulation for AUC maximization, Shao et al. (2023) considers a weighted AUC formulation for cost-sensitive learning, and Yang et al. (2023b) considers AUC maximization with certified robustness. Yang et al. (2022); Shao et al. (2022) focus on TPAUC maximization with the following differences: Yang et al. (2022) tackles the data selection challenge by a weighting scheme, which does not yield the exact TPAUC surrogate objective; Shao et al. (2022) considers TPAUC maximization with a special square loss. In contrast, we directly tackle solving the exact TPAUC surrogate objective without further approximation and our result applies to any non-decreasing loss function.

**Compositional Optimization.** Compositional optimization has gained substantial attention in recent years. This area of optimization deals with objective functions that are composed of multiple nested functions, leading to challenges in efficient evaluation and optimization. Several papers (Wang et al., 2017a;b; Zhang & Lan, 2020; Zhang & Xiao, 2022) have considered standard compositional optimization, where the inner function does not depend on the random variable of the outer level. However, simply applying these algorithms to TPAUC maximization would suffer a high cost (Qi et al., 2021). To address this issue, Zhu et al. (2022) have formulated TPAUC maximization as FCCO (Finite-Sum Coupled Compositional Optimization) as introduced in (Qi et al., 2021). Hu et al. (2023) have proposed an algorithm termed SONX for solving a non-smooth FCCO optimization where the outer function is non-smooth and applied it to TPAUC maximization.

**Min-Max Optimization.** Many stochastic primal-dual algorithms have been proposed to solve non-convex min-max optimization since the seminal work (Rafique et al., 2022). Built on their proximal-guided algorithmic framework, Zhu et al. (2022) developed SOTA for solving the min-max formulation of TPAUC loss. However, their algorithm suffers from the limitations mentioned before. To address its limitations, we have to consider double block-coordinate updates for both primal and dual variables and develop advanced techniques to derive a complexity that has a parallel speed-up, which means complexity is linearly dependent on both positive and negative mini-batch size . Several works (Zhang & Xiao, 2015; Alacaoglu et al., 2022) have considered stochastic primal-dual block-coordinate algorithms for solving finite-sum min-max problems with a bilinear structure, where the block-coordinate update is only applied to the dual variable. Hamedani et al. (2023); Jalilzadeh et al. (2019) have considered more general min-max problems using block-coordinate updates for the primal variable only or for both primal and dual variables. However, their algorithm and analysis require the coupled function to be smooth in terms of both the primal and dual variables, which is not applicable to TPAUC maximization. In addition, Li et al. (2025) propose a Smoothed Proximal Linear Descent-Ascent (Smoothed PLDA) algorithm for deterministic nonsmooth nonconvex-nonconcave minimax problems with convergence guarantees under the KL property. However, PLDA is not directly applicable to large-scale stochastic problems with composite structure, where full dual updates and deterministic computations are infeasible. Recently, Wang & Yang proposed a novel stochastic primal-dual block-coordinate algorithm to solve convex finite-sum compositional optimization problems, which only employs the block-coordinate update on the dual variable.

### 2.1 Notations and Definitions

We present notations in this section. For any $\mathbf{w} \in \mathcal{W}$, the subdifferential $\partial_{\mathbf{w}} f(\mathbf{w})$ is the set of subgradients of $f$ at point $\mathbf{w}$. For a vector $\mathbf{y} \in \mathbb{R}^n$, $\mathbf{y}^{(i)} \in \mathbb{R}$ represents the $i$-th coordinate (block) of $\mathbf{y}$, i.e., $\mathbf{y} = (\mathbf{y}^{(1)}, \cdots, \mathbf{y}^{(n)})^{\mathbb{T}}$.

We use $f_i^*$ to denote the convex conjugate of $f_i$. For a function $g(\mathbf{x}) = \mathbb{E}_{\xi \sim \mathbb{P}}[g(\mathbf{x}; \xi)]$, we define the stochastic estimator based on the mini-batch $\mathcal{B}$ as $g(\mathbf{x}; \mathcal{B}) := \frac{1}{|\mathcal{B}|} \sum_{\xi \in \mathcal{B}} g(\mathbf{x}; \xi)$.

# 3 Primal-dual Double Block-Coordinate Algorithms for TPAUC Maximization

Let $\mathbf{x}$ denote an input example and $h_{\mathbf{w}}(\mathbf{x})$ denote a prediction of a parameterized model such as a deep neural network or a linear model on data $\mathbf{x}$. Denote by $\mathcal{S}_+$ the set of $n_+$ positive examples and by $\mathcal{S}_-$ the set of $n_-$ negative examples. TPAUC measures the area under the ROC curve where the TPR is higher than $1 - \theta_0$ and the FPR is lower than an upper bound $\theta_1$. A surrogate loss for optimizing TPAUC with TPR $\geq 1 - \theta_0$, FPR$\leq \theta_1$ is given by:

$$\min_{\mathbf{w} \in \mathbb{R}^d} \frac{1}{n_+ n_-} \sum_{\mathbf{x}_i \in \mathcal{S}_+^{\uparrow}[1,k_1]} \sum_{\mathbf{x}_j \in \mathcal{S}_-^{\downarrow}[1,k_2]} \ell(h_{\mathbf{w}}(\mathbf{x}_j) - h_{\mathbf{w}}(\mathbf{x}_i)), \tag{1}$$

where $\ell(\cdot)$ is a convex, monotonically non-decreasing surrogate loss of the indicator function $\mathbb{I}(h_{\mathbf{w}}(\mathbf{x}_j) \geq h_{\mathbf{w}}(\mathbf{x}_i))$, $\mathcal{S}_+^{\uparrow}[1,k_1]$ is the set of positive examples with $k_1 = \lfloor n_+ \theta_0 \rfloor$ smallest scores, and $\mathcal{S}_-^{\downarrow}[1,k_2]$ is the set of negative examples with $k_2 = \lfloor n_- \theta_1 \rfloor$ largest scores. To tackle the challenge of selecting examples for $\mathcal{S}_+^{\uparrow}[1,k_1]$ and $\mathcal{S}_-^{\downarrow}[1,k_2]$, we use the following lemma to reformulate (1) (Zhu et al., 2022).

**Lemma 3.1.** *If $\ell(\cdot)$ is non-decreasing, then the TPAUC loss minimization problem (1) is equivalent to the following:*

$$\min_{\mathbf{w}, s', \mathbf{s}} \frac{1}{n_+} \sum_{\mathbf{x}_i \in \mathcal{S}_+} f_i(g_i(\mathbf{w}, \mathbf{s}^{(i)}), s'), \tag{2}$$

*where $\mathbf{s} = (\mathbf{s}^{(1)}, \cdots, \mathbf{s}^{(n_+)})^{\top}$, $f_i(g, s') = s' + \frac{1}{\theta_0}[g - s']_+$, and $g_i(\mathbf{w}, \mathbf{s}^{(i)}) = \frac{1}{n_-} \sum_{\mathbf{x}_j \in \mathcal{S}_-} \mathbf{s}^{(i)} + \frac{[\ell(h_{\mathbf{w}}(\mathbf{x}_j) - h_{\mathbf{w}}(\mathbf{x}_i)) - \mathbf{s}^{(i)}]_+}{\theta_1}$.* The reformulation above uses an equivalent form of the conditional-value-at-risk (CVaR) loss, $\frac{1}{n\gamma} \sum_{i=1}^{n\gamma} \ell_{[i]}(\cdot) = \min_s s + \frac{1}{n\gamma} \sum_{i=1}^{n} [\ell_i(\cdot) - s]_+$, where $\gamma = k/n$ for some integer $k \in [n]$, $\ell_{[i]}(\cdot)$ denotes the $i$-th largest value in $\{\ell_1, \cdots, \ell_n\}$. (Ogryczak & Tamir, 2003, Lemma 1). Since $[t]_+ = \max_{y \in [0,1]} ty$, we cast (2) into an equivalent min-max problem:

$$\min_{\substack{\mathbf{w} \in \mathbb{R}^d, s' \in \mathbb{R} \\ \mathbf{s} \in \mathbb{R}^{n_+}}} \max_{\mathbf{y} \in [0,1]^{n_+}} \frac{1}{n_+} \sum_{\mathbf{x}_i \in \mathcal{S}_+} \mathbf{y}^{(i)} \cdot \frac{g_i(\mathbf{w}, \mathbf{s}^{(i)}) - s'}{\theta_0} + s'. \tag{3}$$

This problem presents unique challenges that make existing algorithms unsuitable for direct application: (i) the objective function is non-smooth with respect to $\mathbf{w}$ and $\mathbf{s}$ due to the hinge function in $g_i$; (ii) both the primal variable $\mathbf{s}$ and the dual variable $\mathbf{y}$ are high-dimensional and depend on all positive examples, preventing their full coordinate updates in each iteration; and (iii) the coupled term is not bilinear with respect to the primal and dual variables.

## 3.1 Algorithms

Now we present our efficient algorithms designed to solve the min-max problem (3) in convex and non-convex settings.

**STACO1 for convex functions.** We first consider the convex case when $\ell(h_{\mathbf{w}}(\mathbf{x}_j) - h_{\mathbf{w}}(\mathbf{x}_i))$ is a convex function of $\mathbf{w}$. This is true when we learn a linear model such that $h_{\mathbf{w}}(\mathbf{x}) = \mathbf{w}^{\top}\mathbf{x}$. Hence, $g_i(\mathbf{w}, s)$ is convex w.r.t. $(\mathbf{w}, s)$ for any $i \in [n]$, and (3) is a convex-concave min-max problem.

A challenge of solving (3) is that updating all coordinates for $\mathbf{s}, \mathbf{y}$ would require computing $g_i(\mathbf{w}, \mathbf{s}^{(i)})$ and its gradient for all positive examples $\mathbf{x}_i \in \mathcal{S}_+$, which is prohibited when the number of positive examples is large. Hence, we have to use block-coordinate updates for both $\mathbf{s}$ and $\mathbf{y}$. Let us consider how to update $\mathbf{y}^{(i)}$ and $\mathbf{s}^{(i)}$ for a sampled coordinate $i$. A simple method is to use gradient ascent to update $\mathbf{y}^{(i)}$ and use gradient descent to update $\mathbf{s}^{(i)}$, which require computing $g_i(\mathbf{w}, \mathbf{s}^{(i)})$ and $\partial_{\mathbf{s}^{(i)}} g_i(\mathbf{w}, \mathbf{s}^{(i)})$. However, this would require processing all negative examples $\mathcal{S}_-$ as $g_i(\mathbf{w}, \mathbf{s}^{(i)})$ depends on all negative examples. To reduce this cost, we

---

**Algorithm 1** STACO1

---

1: Initialize $\mathbf{w}_0 \in \mathcal{W}$, $\mathbf{y}_0 = \mathbf{1}^{n_+}$, $\mathbf{s}_0 = \mathbf{1}^{n_+}$, $s'_0 = 1$,
2: **for** $t = 0, 1, \ldots, T-1$ **do**
3:     Sample a batch $\mathcal{S}_t \subset \mathcal{S}_+$ with $|\mathcal{S}_t| = S$
4:     Sample independent mini-batches $\mathcal{B}_t, \tilde{\mathcal{B}}_t \subset \mathcal{S}_-$
5:     **for** each $i \in \mathcal{S}_t$ **do**
6:         Update $\mathbf{y}_{t+1}^{(i)}$ according to (4)
7:         Update $\mathbf{s}_{t+1}^{(i)}$ according to (5)
8:     **end for**
9:     For each $i \notin \mathcal{S}_t$, $\mathbf{y}_{t+1}^{(i)} = \mathbf{y}_t^{(i)}$ and $\mathbf{s}_{t+1}^{(i)} = \mathbf{s}_t^{(i)}$
10:    Update $\mathbf{w}_{t+1}$ according to (6)
11:    Update $s'_{t+1}$ according to (7)
12: **end for**
13: $\bar{\mathbf{w}} = \frac{1}{T} \sum_{t=0}^{T-1} \mathbf{w}_{t+1}, \bar{\mathbf{s}} = \frac{1}{T} \sum_{t=0}^{T-1} \mathbf{s}_{t+1}, \bar{s}' = \frac{1}{T} \sum_{t=0}^{T-1} s'_{t+1}$
14: Return $\bar{\mathbf{w}}, \bar{\mathbf{s}}, \bar{s}'$

---

need to use stochastic estimators of their gradients. For a random mini-batch of negative samples $\mathcal{B} \subset \mathcal{S}_-$, we let

$$g_i(\mathbf{w}, \mathbf{s}^{(i)}; \mathcal{B}) = \frac{1}{|\mathcal{B}|} \sum_{\mathbf{x}_j \in \mathcal{B}} \mathbf{s}^{(i)} + \frac{[\ell(h_{\mathbf{w}}(\mathbf{x}_j) - h_{\mathbf{w}}(\mathbf{x}_i)) - \mathbf{s}^{(i)}]_+}{\theta_1}.$$

At the $t$-th iteration, we sample a mini-batch of $S$ positive examples $\mathcal{S}_t \subset \mathcal{S}_+$ and a mini-batch of $B$ negative examples $\mathcal{B}_t \subset \mathcal{S}_-$. We update $\mathbf{y}_{t+1}^{(i)}$ according to

$$\mathbf{y}_{t+1}^{(i)} = \arg\max_{\mathbf{y}^{(i)} \in [0,1]} \left\{ \mathbf{y}^{(i)} \cdot \frac{g_i(\mathbf{w}_t, \mathbf{s}_t^{(i)}; \mathcal{B}_t) - s'_t}{\theta_0} - \frac{1}{2\alpha} \left( \mathbf{y}^{(i)} - \mathbf{y}_t^{(i)} \right)^2 \right\}, \forall \mathbf{x}_i \in \mathcal{S}_t \tag{4}$$

where $\alpha$ is a step size parameter. Then we update $\mathbf{s}_{t+1}^{(i)}, i \in \mathcal{S}_t$ and $\mathbf{w}_{t+1}$ using stochastic gradient descent:

$$\mathbf{s}_{t+1}^{(i)} = \mathbf{s}_t^{(i)} - \frac{\beta}{\theta_0} \mathbf{y}_{t+1}^{(i)} \partial_{\mathbf{s}^{(i)}} g_i(\mathbf{w}_t, \mathbf{s}_t^{(i)}; \tilde{\mathcal{B}}_t), \forall \mathbf{x}_i \in \mathcal{S}_t \tag{5}$$

$$\mathbf{w}_{t+1} = \mathbf{w}_t - \frac{\eta}{\theta_0} \frac{1}{S} \sum_{i \in \mathcal{S}_t} \mathbf{y}_{t+1}^{(i)} \partial_{\mathbf{w}} g_i(\mathbf{w}_t, \mathbf{s}_t^{(i)}; \tilde{\mathcal{B}}_t) \tag{6}$$

$$s'_{t+1} = s'_t - \beta'\left(1 - \frac{1}{\theta_0 S} \sum_{i \in \mathcal{S}_t} \mathbf{y}_{t+1}^{(i)}\right) \tag{7}$$

where $\beta, \eta, \beta'$ are step size parameters, and we use another mini-batch of negative samples $\tilde{\mathcal{B}}_t$ independent of $\mathcal{B}_t$ to decouple the dependence between $\mathbf{y}_{t+1}^{(i)}$ and $\tilde{\mathcal{B}}_t$. The detailed steps of **STACO1** are presented in Algorithm 1.

**STACO2 for non-convex functions.** Next we consider the non-convex case. We assume $\ell(h_{\mathbf{w}}(\mathbf{x}_j) - h_{\mathbf{w}}(\mathbf{x}_i))$ is weakly-convex with respect to $\mathbf{w}$, which holds true when $\ell$ is a convex non-smooth function and $h_{\mathbf{w}}(\mathbf{x})$ is a smooth function of $\mathbf{w}$ (Hu et al., 2023). Hence, $g_i(\mathbf{w}, s)$ is weakly-convex with respect to $(\mathbf{w}, s)$, and (3) is a weakly-convex concave min-max problem. Inspired by the proximal-guided algorithm (Rafique et al., 2022) for non-smooth weakly-convex concave problems, we propose a double-loop algorithm **STACO2** for solving problem (3). The inner loop updates apply STACO1 to solve the following problem approximately at the $t$-th outer iteration:

$$\min_{\substack{\mathbf{w} \in \mathbb{R}^d, s' \in \mathbb{R} \\ \mathbf{s} \in \mathbb{R}^{n_+}}} \max_{\mathbf{y} \in [0,1]^{n_+}} \frac{1}{n_+} \sum_{\mathbf{x}_i \in \mathcal{S}_+} \mathbf{y}^{(i)} \cdot \frac{g_i(\mathbf{w}, \mathbf{s}^{(i)}) - s'}{\theta_0} + s' + \frac{1}{2\gamma} \|\mathbf{w} - \mathbf{w}_{t,0}\|_2^2 + \frac{1}{2n_+\gamma} \|\mathbf{s} - \mathbf{s}_{t,0}\|_2^2, \tag{8}$$

where $\mathbf{w}_{t,0}, \mathbf{s}_{t,0}$ are initial value of $\mathbf{w}, \mathbf{s}$ at $t$-th stage, $\gamma > 0$ is a proper parameter. The addition of quadratic functions is to ensure the function becomes convex in terms of $\mathbf{w}, \mathbf{s}$. At $k$-th iteration in $t$-th stage, we utilize

---

**Algorithm 2** STACO2

---

1: Initialize $\mathbf{w}_0 \in \mathcal{W}$, $\mathbf{s}_0 = \mathbf{1}^{n_+}$, $s_0' = 1$
2: **for** $t = 0, 1, \dots, T-1$ **do**
3:      Initialize $\mathbf{y}_{t,0} = \mathbf{1}^{n_+}$
4:      Set $\mathbf{w}_{t,0} = \mathbf{w}_t, \mathbf{s}_{t,0} = \mathbf{s}_t, s_{t,0}' = s_t'$
5:      **for** $k = 0, 1, \dots, K_t - 1$ **do**
6:          Sample a batch $\mathcal{S}_{t,k} \subset \mathcal{S}_+$, where $|\mathcal{S}_{t,k}| = S$
7:          Sample independent mini-batches $\mathcal{B}_{t,k}, \tilde{\mathcal{B}}_{t,k} \subset \mathcal{S}_-$
8:          **for** each $i \in \mathcal{S}_{t,k}$ **do**
9:              Update $\mathbf{y}_{t,k+1}^{(i)}$ according to (9)
10:         Update $\mathbf{s}_{t,k+1}^{(i)}$ according to (10)
11:          **end for**
12:          For each $i \notin \mathcal{S}_{t,k}$, $\mathbf{y}_{t,k+1}^{(i)} = \mathbf{y}_{t,k}^{(i)}$ and $\mathbf{s}_{t,k+1}^{(i)} = \mathbf{s}_{t,k}^{(i)}$
13:          Update $\mathbf{w}_{t+1}$ according to (11)
14:          Update $s_{t+1}'$ according to (12)
15:      **end for**
16:      $(\bar{\mathbf{w}}_t, \bar{\mathbf{s}}_t, \bar{s}_t') = \frac{1}{K_t} \sum_{k=0}^{K_t-1} (\mathbf{w}_{t,k+1}, \mathbf{s}_{t,k+1}, s_{t,k+1}')$
17:      Set $\mathbf{w}_{t+1} = \bar{\mathbf{w}}_t, \mathbf{s}_{t+1} = \bar{\mathbf{s}}_t, s_{t+1}' = \bar{s}_t'$
18: **end for**
19: Return $\mathbf{w}_T, \mathbf{s}_T, s_T'$

---

following updates:

$$\mathbf{y}_{t,k+1}^{(i)} = \arg\max_{\mathbf{y}^{(i)} \in [0,1]} \left\{ \mathbf{y}^{(i)} \cdot \frac{g_i(\mathbf{w}_{t,k}, \mathbf{s}_{t,k}^{(i)}; \mathcal{B}_{t,k}) - s_{t,k}'}{\theta_0} - \frac{1}{2\alpha_t} \left( \mathbf{y}^{(i)} - \mathbf{y}_{t,k}^{(i)} \right)^2 \right\}, \forall \mathbf{x}_i \in \mathcal{S}_{t,k} \tag{9}$$

$$\mathbf{s}_{t,k+1}^{(i)} = \mathbf{s}_{t,k}^{(i)} - \frac{\beta_t}{\theta_0} \left( \mathbf{y}_{t,k+1}^{(i)} \partial_{\mathbf{s}^{(i)}} g_i(\mathbf{w}_{t,k}, \mathbf{s}_{t,k}^{(i)}; \tilde{\mathcal{B}}_{t,k}) + \frac{1}{\gamma} (\mathbf{s}_{t,k}^{(i)} - \mathbf{s}_{t,0}^{(i)}) \right), \forall \mathbf{x}_i \in \mathcal{S}_{t,k} \tag{10}$$

$$\mathbf{w}_{t,k+1} = \mathbf{w}_{t,k} - \frac{\eta_t}{\theta_0} \left( \frac{1}{S} \sum_{i \in \mathcal{S}_{t,k}} \mathbf{y}_{t,k+1}^{(i)} \partial_{\mathbf{w}} g_i(\mathbf{w}_{t,k}, \mathbf{s}_{t,k}^{(i)}; \tilde{\mathcal{B}}_{t,k}) + \frac{1}{\gamma} (\mathbf{w}_{t,k} - \mathbf{w}_{t,0}) \right) \tag{11}$$

$$s_{t,k+1}' = s_{t,k}' - \beta_t' (1 - \frac{1}{\theta_0 S} \sum_{i \in \mathcal{S}_{t,k}} \mathbf{y}_{t,k+1}^{(i)}), \tag{12}$$

where $\alpha_t, \beta_t, \eta_t, \beta_t'$ are step size parameters.

We would like to highlight the difference between STACO2 and SOTA (Zhu et al., 2022), where we use block-coordinate update for $\mathbf{s} \in \mathbb{R}^+$. In contrast, SOTA needs to update all coordinates of $\mathbf{s}$. This difference is caused by different techniques for handling all coordinates: they compute an unbiased sparse stochastic gradient for $\mathbf{s}$ by sampling and then update $\mathbf{s}$ using a stochastic proximal gradient method. The unbiased sparse stochastic gradient used in SOTA cannot enjoy a variance bound that scales with the mini-batch size. In contrast, we just compute an unbiased stochastic gradient for the sampled coordinate of $\mathbf{s}$, and perform a stochastic gradient descent on sampled coordinates and leave other coordinates unchanged. It is this difference that makes our analysis more involved and leads to a parallel speed-up.

## 4 Analysis

In this section, we present the convergence results for our algorithms. We emphasize the contributions of our convergence analysis for both convex and non-convex settings compared to Zhu et al. (2022): (i) our convergence analysis for the convex case is more refined, leading to an optimal convergence rate which implies a parallel speed-up in terms of mini-batch size; (ii) our analysis for the non-convex case is also improved, which not only enjoys a parallel speed-up but also removes strong boundedness assumptions of $\mathbf{s}_{t,k}, s_{t,k}'$ and the pairwise loss values at all iterations.

For analysis, we consider the following optimization problem:

$$\min_{\mathbf{u} \in \mathcal{U}, \mathbf{s} \in \mathcal{S}} F(\mathbf{u}, \mathbf{s}) \coloneqq \frac{1}{n} \sum_{i=1}^{n} f_i(g_i(\mathbf{u}, \mathbf{s}^{(i)})), \tag{13}$$

where $f_i : \mathbb{R} \to \mathbb{R}$ is closed proper convex and lower-semicontinuous, $g_i : (\mathcal{U}, \in \mathcal{S}_i) \to \mathbb{R}$ is possibly non-convex, and $\mathcal{U}, \mathcal{S}$ are convex closed sets, $g_i(\mathbf{u}, \mathbf{s}^{(i)}) \coloneqq \mathbf{E}_{\zeta_i \sim \mathbb{P}_i} \left[ g_i(\mathbf{u}, \mathbf{s}^{(i)}; \zeta_i) \right]$. It is equivalent to the following min-max problem:

$$\min_{\mathbf{u} \in \mathcal{U}, \mathbf{s} \in \mathcal{S}} \max_{\mathbf{y} \in \mathcal{Y}} L(\mathbf{u}, \mathbf{s}, \mathbf{y}) \coloneqq \frac{1}{n} \sum_{i=1}^{n} \mathbf{y}^{(i)} g_i(\mathbf{u}, \mathbf{s}^{(i)}) - f_i^*(\mathbf{y}^{(i)}). \tag{14}$$

Compared to problem (2), (13) excludes parameter $s'$. Since the update of $s'$ is almost the same as $\mathbf{w}$, our analysis for solving (13) can be easily extended to STACO1 and STACO2.

## 4.1 Assumptions

We first outline assumptions underlying our analysis. Notably, these assumptions are easily satisfied for TPAUC maximization when the loss function $\ell$ is Lipchitz continuous.

**Assumption 4.1.** For any $i \in [n]$, we suppose $f_i, g_i$ is Lipschitz continuous, i.e., there exists $C_f, C_g > 0$ such that

$$|f_i(u) - f_i(\bar{u})| \le C_f |u - \bar{u}|$$
$$\left| g_i(\mathbf{u}, \mathbf{s}^{(i)}) - g_i(\bar{\mathbf{u}}, \bar{\mathbf{s}}^{(i)}) \right| \le C_g \left( \|\mathbf{u} - \bar{\mathbf{u}}\|_2 + \left| \mathbf{s}^{(i)} - \bar{\mathbf{s}}^{(i)} \right| \right),$$

for any $u, \bar{u} \in \mathbb{R}$, $\mathbf{u}, \bar{\mathbf{u}} \in \mathcal{U}$ and $\mathbf{s}^{(i)}, \bar{\mathbf{s}}^{(i)} \in \mathcal{S}_i$.

**Assumption 4.2.** For any $i \in [n]$, there exists finite $\sigma_0^2, \sigma_1^2, \sigma_2^2$ such that

$$\mathbf{E}_{\zeta_i} \left| g_i(\mathbf{u}, \mathbf{s}^{(i)}) - g_i(\mathbf{u}, \mathbf{s}^{(i)}; \zeta_i) \right|^2 \le \sigma_0^2,$$
$$\mathbf{E}_{\zeta_i} \left\| \hat{G}_1^{(i)}(\zeta_i) - G_1^{(i)} \right\|_2^2 \le \sigma_1^2, \quad \mathbf{E}_{\zeta_i} \left\| \hat{G}_2^{(i)}(\zeta_i) - G_2^{(i)} \right\|_2^2 \le \sigma_2^2,$$

for stochastic subgradients $\hat{G}_1^{(i)}(\zeta_i) \in \partial_{\mathbf{u}} g_i(\mathbf{u}, \mathbf{s}^{(i)}; \zeta_i)$, $\hat{G}_2^{(i)}(\zeta_i) \in \partial_{\mathbf{s}^{(i)}} g_i(\mathbf{u}, \mathbf{s}^{(i)}; \zeta_i)$ at any $\mathbf{u} \in \mathcal{U}$, and $\mathbf{s}^{(i)} \in \mathcal{S}_i$. Besides, there exists $\delta^2$ such that

$$\mathbf{E}_j \left\| y^{(j)} G_1^{(j)} - \frac{1}{n} \sum_{i=1}^{n} y^{(i)} G_1^{(i)} \right\|_2^2 \le \delta^2,$$

for any $G_1^{(i)} \in \partial_1 g_i(\mathbf{u}, \mathbf{s}^{(i)})$, $\mathbf{u} \in \mathcal{U}, \mathbf{s}^{(i)} \in \mathcal{S}_i$, and $\mathbf{y} \in \mathcal{Y}$. Note that under Assumption 4.1, we have $\delta^2 \le C_f^2 C_g^2$.

## 4.2 Convex Case

We first analyze the Algorithm 3, which aims to solve the problem (14) when both $f_i$ and $g_i$ are convex for any $i \in [n]$. The analysis is motivated by techniques proposed in Wang & Yang. However, the problem they considered is $\frac{1}{n} \sum_{i=1}^{n} f_i(g_i(\mathbf{u}))$, which excludes the primal parameter $\mathbf{s}$. Notably, the analysis of convergence of primal parameter $\mathbf{u}$ is more tricky than $\mathbf{w}$ since its updating only lies in selected coordinates each iteration.

**Theorem 4.3.** *Under Assumptions 4.1 and 4.2, when $g_i(\mathbf{u}, \mathbf{s}^{(i)})$ is convex w.r.t $\mathbf{u}, \mathbf{s}^{(i)}$, let $\eta = \mathcal{O}(\epsilon)$, $\beta = \mathcal{O}(\epsilon)$, and $\alpha = \mathcal{O}(B\epsilon)$, STACO1 can make $\mathbf{E}[F(\bar{\mathbf{u}}, \bar{\mathbf{s}}) - F(\mathbf{u}^*, \mathbf{s}^*)] \le \epsilon$ after $T = \mathcal{O}\left( \frac{nC_g^2 C_f^2}{S\epsilon^2} + \frac{C_f^2 \sigma_1^2}{B\epsilon^2} + \frac{nC_f^2 \sigma_2^2}{BS\epsilon^2} + \frac{\delta^2}{S\epsilon^2} + \frac{n\sigma_0^2}{BS\epsilon^2} \right)$ iterations, where $\bar{\mathbf{u}} = \frac{1}{T} \sum_{t=0}^{T-1} \mathbf{u}_{t+1}, \bar{\mathbf{s}} = \frac{1}{T} \sum_{t=0}^{T-1} \mathbf{s}_{t+1}$.*

**Remark.** The proof is included in Appendix C.2.2. The above convergence rate implies a parallel speed-up in terms of the positive batch size $S$ and negative batch size $B$. When we use full information at each iteration, which means $\sigma_0 = 0, \sigma_1 = 0, \sigma_2 = 0, \delta = 0, S = n_+$, the above complexity reduces to $O(1/\epsilon^2)$, which is a standard complexity for non-smooth convex optimization (Nesterov et al., 2018). In addition, the dominating term $O(n/(S\epsilon^2))$ matches the lower bound proved in Wang & Yang.

### 4.3 Non-convex Case

Now we consider the non-convex case when $g_i$ is weakly convex as stated in the following assumption.

**Assumption 4.4** (weakly convexity of $g_i$). For any $i \in [n]$, we suppose that $g_i(\mathbf{u}, \mathbf{s}^{(i)})$ is $\rho$-weakly convex to $\mathbf{u}$ and $\mathbf{s}^{(i)}$ for any $\mathbf{u} \in \mathcal{U}$ and $\mathbf{s}^{(i)} \in \mathcal{S}_i$, i.e., $g_i(\cdot) + \frac{\rho}{2} \|\cdot\|_2^2$ is convex, where $\rho$ is a positive number.

It is sometimes difficult to find an $\epsilon$-stationary point $(\mathbf{u}, \mathbf{s})$ of the non-smooth function $F$, i.e., $\mathrm{dist}(0, \partial F(\mathbf{u}, \mathbf{s})) \le \epsilon$. For example, an $\epsilon$-stationary point of function $f(\mathbf{x}) = |\mathbf{x}|$ does not exist for $0 \le \epsilon < 1$ unless it is the optimal solution. To address this problem, (Davis & Drusvyatskiy, 2018) proposed using the stationarity of the Moreau envelope of the problem as the convergence metric, which has become a standard metric for solving weakly convex problems.

Given a $\rho$-weakly convex function $f : \mathbb{R}^d \to \mathbb{R}$, its Moreau envelope is constructed as

$$f_\gamma(\mathbf{x}) := \min_{\mathbf{w} \in \mathbb{R}^d} \left\{ f(\mathbf{w}) + \frac{1}{2\gamma} \|\mathbf{w} - \mathbf{x}\|_2^2 \right\}, \tag{15}$$

where $\gamma$ is a positive constant. For a $\rho$-weakly convex function $f$, it can be shown that $f_\gamma$ is smooth when $\frac{1}{\gamma} > \rho$ (Davis & Grimmer, 2019) and its gradient is

$$\nabla f_\gamma(\mathbf{x}) = \frac{1}{\gamma}(\mathbf{x} - \mathrm{prox}_\gamma f(\mathbf{x})), \tag{16}$$

where

$$\mathrm{prox}_\gamma f(\mathbf{x}) := \arg\min_{\mathbf{w}} \{ f(\mathbf{w}) + \frac{1}{2\gamma} \|\mathbf{w} - \mathbf{x}\|_2^2 \}. \tag{17}$$

Notice that when $\frac{1}{\gamma} > \rho$, the minimization in problem (15) is strongly convex, which ensures $\mathrm{prox}_\gamma f(\mathbf{x})$ is uniquely defined. Moreover, for any point $\mathbf{x} \in \mathbb{R}^d$, the proximal point $\mathbf{x}^\dagger := \mathrm{prox}_\gamma f(\mathbf{x})$ satisfies (Hu et al., 2023)

$$\left\| \mathbf{x}^\dagger - \mathbf{x} \right\|_2 = \gamma \left\| \nabla f_\gamma(\mathbf{x}) \right\|_2, \quad f_\gamma(\mathbf{x}^\dagger) \le f_\gamma(\mathbf{u}), \quad \mathrm{dist}(0, \partial f(\mathbf{x}^\dagger)) \le \left\| \nabla f_\gamma(\mathbf{x}) \right\|_2. \tag{18}$$

Thus if $\|\nabla f_\gamma(\mathbf{x})\|_2 \le \epsilon$, we can say $\mathbf{x}$ is close to a point $\mathbf{x}^\dagger$ that is $\epsilon$-stationary, which is called nearly $\epsilon$-stationary solution of $f(\mathbf{x})$. Given an iterate $\mathbf{x}_t$, a common idea is using the stochastic subgradient method (SSG) to approximately solve (15) with $\mathbf{x} = \mathbf{x}_t$, namely, to compute a solution $\mathbf{x}_{t+1}$ such that

$$\mathbf{x}_{t+1} \approx \mathrm{prox}_\gamma(\mathbf{x}_t) = \arg\min_{\mathbf{x}} \left\{ f(\mathbf{x}) + \frac{1}{2\gamma} \|\mathbf{x} - \mathbf{x}_t\|_2^2 \right\}. \tag{19}$$

Then $\mathbf{x}_{t+1}$ returned by the SSG method will then be used in the next iterate. Inspired by Rafique et al. (2022), we consider the following update according to equation (19)

$$(\mathbf{u}_{t+1}, \mathbf{s}_{t+1}, \mathbf{y}_{t+1}) \approx \arg\min_{\mathbf{u} \in \mathcal{U}, \mathbf{s} \in \mathcal{S}} \arg\max_{\mathbf{y} \in \mathcal{Y}} \left\{ L_\gamma(\mathbf{u}, \mathbf{s}, \mathbf{y}; \mathbf{u}_t, \mathbf{s}_t) \right\},$$

$$\text{where } L_\gamma(\mathbf{u}, \mathbf{s}, \mathbf{y}; \mathbf{u}', \mathbf{s}') := \frac{1}{n} \sum_{i=1}^{n} \left( \mathbf{y}^{(i)} g_i(\mathbf{u}, \mathbf{s}^{(i)}) - f_i^*(\mathbf{y}^{(i)}) \right) + \frac{1}{2\gamma} \|\mathbf{u} - \mathbf{u}'\|_2^2 + \frac{1}{2n\gamma} \|\mathbf{s} - \mathbf{s}'\|_2^2. \tag{20}$$

**Theorem 4.5.** *Under Assumptions 4.1, 4.2 and 4.4, STACO2 with* $\gamma \le \frac{1}{2C_f \rho}$, $\eta_t = \mathcal{O}(\epsilon^2)$, $\beta_t = \mathcal{O}(\epsilon^2)$, $\alpha_t = \mathcal{O}(B\epsilon^2)$*, and* $K_t = \mathcal{O}\left( \frac{n}{BS\epsilon^4} \vee \frac{1}{\eta_t} \vee \frac{n}{S\beta_t} \right)$ *can converge to an* $\epsilon$-*stationary point of* $\Phi_\gamma(\mathbf{u}, \mathbf{s})$ *in* $\mathcal{O}\left( \frac{C_f^2 \sigma_1^2}{B\epsilon^4} + \frac{\delta^2}{S\epsilon^4} + \frac{nC_g^2 C_f^2}{S\epsilon^4} + \frac{nC_f^2 \sigma_2^2}{BS\epsilon^4} + \frac{n\sigma_0^2}{BS\epsilon^6} \right)$ *iterations, where* $\Phi_\gamma(\mathbf{u}, \mathbf{s}) = \min_{\tilde{\mathbf{u}}, \tilde{\mathbf{s}}} F(\tilde{\mathbf{u}}, \tilde{\mathbf{s}}) + \frac{1}{2\gamma} \|\tilde{\mathbf{u}} - \mathbf{u}\|_2^2 + \frac{1}{2\gamma n} \|\tilde{\mathbf{s}} - \mathbf{s}\|_2^2$ *is a Moreau envelope of* $F(\mathbf{u}, \mathbf{s})$.

**Remark** The proof is included in Appendix C.3.4. We compare the above result with the complexity of SOTA and SONX. In particular, SOTA has a complexity of $\mathcal{O}\left(\frac{n}{\epsilon^6}\right)$ result, which cannot show any mini-batch speedup. SONX has an iteration complexity of $\mathcal{O}\left(\frac{n}{B^{1/2} S\epsilon^6}\right)$ in Theorem C.4 (Hu et al., 2023). In comparison, our complexity $\mathcal{O}\left(\frac{n}{BS\epsilon^6}\right)$ has a better dependence on $B$.

# 5 Experiments

We evaluate the empirical performance of our proposed algorithm against baselines for Two-way Partial AUC Maximization (TPAUC) in a convex setting for learning linear models and a non-convex setting for learning deep models.

## 5.1 Settings

**Datasets.** For linear model experiments, we use three datasets in (Chang & Lin, 2011), namely HIGGS, SUSY, and ijcnn1. For SUSY and HIGGS, we use the first 80% of the data as the training dataset and the remaining 20% as the testing dataset. For ijcnn1, we follow the existing split in (Chang & Lin, 2011). To create imbalanced datasets for HIGGS and SUSY (ijcnn1 itself is imbalanced), we randomly remove 99.5% positive data. For deep learning model experiments, we use two molecule datasets from the Stanford Open Graph Benchmark (OGB) website (Hu et al., 2020) and two biomedical image datasets from MedMNIST (Yang et al., 2023a), namely moltox21 (the No.0 target), molmuv (the No.1 target), nodulemnist3d, and adrenalmnist3d. Those four datasets are naturally imbalanced. The task in molecular datasets is to predict certain properties of molecules, and the task in biomedical image datasets is binary classification. The statistics of datasets are presented in Table 5 in Appendix B.

**Models.** For linear model experiments, we let $h_{\mathbf{w}}(\mathbf{x}) = \mathbf{w}^\top \mathbf{x}$. In deep model experiments, for molecule datasets moltox21 and molmuv, we use Graph Isomorphism Network (GIN) (Xu et al., 2018) as the backbone model, which has 5 mean-pooling layers with 64 hidden units and 0.5 dropout rate. For image datasets nodulemnist3d and adrenalmnist3d, we learn a convolutional neural network (CNN) and use ResNet18 (He et al., 2016). We utilize the sigmoid function for the final output layer to generate the prediction score and set the surrogate loss $\ell(\cdot)$ as the squared hinge loss with a margin parameter (Zhu et al., 2022).

**Baselines.** We evaluate our algorithms, STACO1 and STACO2, by comparing their training and testing performance against various baselines, while STACO1 is for linear model and STACO2 is for deep model. Specifically, we benchmark our methods against other approaches that optimizes different objectives, including CE for optimizing the cross-entropy loss, AUCM for optimizing an AUC min-max margin loss (Yuan et al., 2021b), SOTAs for optimizing a soft TPAUC loss (Zhu et al., 2022), SOTA (Zhu et al., 2022), and SONX for optimizing the same TPAUC loss as ours (Hu et al., 2023), and PAUCI for optimizing an instance-wise TPAUC loss (Shao et al., 2022).

**Evaluation Metrics.** For linear and deep learning model experiments, we evaluate TPAUC with two settings, i.e., TPR $\geq 0.5$ and FPR $\leq 0.5$, and TPR $\geq 0.25$ and FPR $\leq 0.75$.

**Hyperparameter Tuning.** In linear model experiments, the model is trained by 3000 iterations, and the learning rate is decreased by 10-fold on the 500th, 1500th, and 2500th iterations for all methods. For deep learning experiments, the model is trained by 60 epochs and the learning rate is decreased by 10-fold after every 20 epochs for all methods. In addition, we pre-train the model for deep learning experiments following previous studies (Yuan et al., 2021b; Zhu et al., 2022). The pre-trained model is trained for 60 epochs using CE loss with an Adam optimizer on the training datasets, and the initial learning rate is 1e-3 which is decreased by 10-fold on the 30th and 45th epochs. We tune the step sizes of STACO1, STACO2, SOTA, PAUCI, and AUCM in the range {1e-2, 1e-1, 5e-1}, and tune the step sizes of SONX, SOTAs, and CE in the range {1e-3, 1e-2, 1e-1}. For STACO1, STACO2, SOTA, and SONX, we fix the margin parameter of the surrogate loss $\ell$ as 0.5, and tune the rate parameter $\theta_0, \theta_1$ in $\{0.4, 0.5, 0.75\}$ for reporting testing performance. For SONX, we fix the moving average parameter as 0.9 and tune the momentum parameter in the range {0, 1e-3, 1e-2, 1e-1}. For AUCM, we choose the momentum parameter as 0.9, the margin parameter of the surrogate loss as 0.5, and tune the hyperparameter $\gamma$ that controls consecutive epoch-regularization in {100, 500, 1000}. For SOTAs, we fix $\gamma_0 = \gamma_1 = 0.9$ and tune $\lambda, \lambda'$ in $\{0.1, 1.0, 10\}$. For PAUCI, we tune $k$ in $[1, 10]$, $c_1, c_2, \mu, \lambda$ in $[0, 1]$, $m$ in $[10, 100]$ and $\kappa$ in $[2, 6]$. For all algorithms, we choose the weight decay parameter as 2e-4. Without specific statements, each algorithm samples 64 data points in each iteration. We execute all experiments using 5-fold-cross-validation to evaluate testing performance based on the best validation performance and report the average and standard deviation over multiple runs.

## 5.2   Results

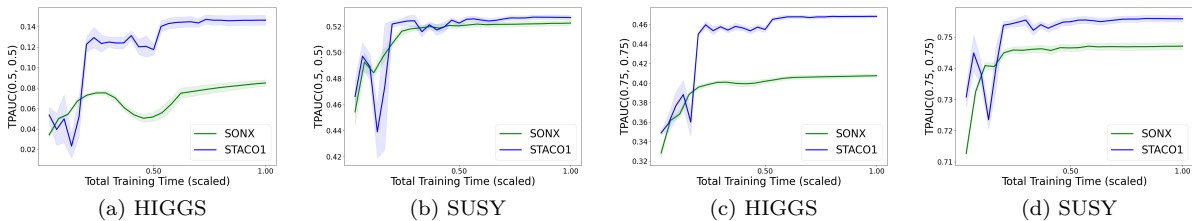

Figure 1: Training TPAUC Curves of STACO1 and SONX on two different datasets. The first two shows the TPAUC (0.5, 0.5) results, and the last two shows the TPAUC (0.75, 0.75) results.

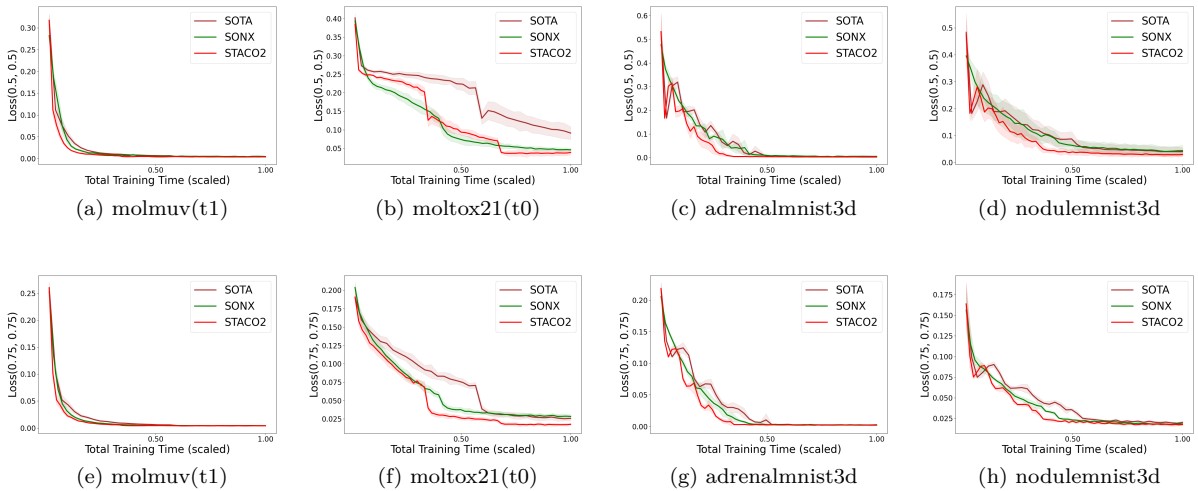

Figure 2: Training Loss Curves of STACO2, SOTA, and SONX on four different datasets. The first row shows the Loss (0.5, 0.5) results, and the second row shows the Loss (0.75, 0.75) results.

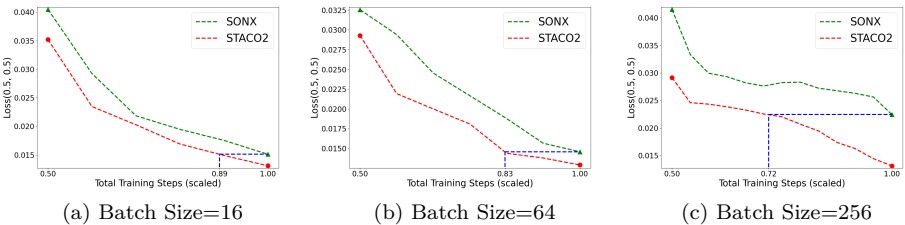

Figure 3: Negative sample batch size ($B$) benefits of STACO2 over SONX for training on ogbg-molmuv (t1) at 16, 64, and 256 batch size.

**Training Results.** Under two different metrics, we compare the training performance of the linear model between STACO1 and SONX in Figure 1, and the deep learning model among STACO2, SOTA, and SONX in Figure 2. We exclude SOTA from linear model experiments since SOTA is designed for optimizing deep learning models. In the linear model experiments as shown in Figure 1, we plot the TPAUC values throughout the training process. The results demonstrate that STACO1 exhibits strong and stable performance, consistently outperforming SONX on both the HIGGS and SUSY datasets in the (0.5, 0.5) and (0.75, 0.75) settings. These findings indicate that STACO1 is more efficient than SONX in maximizing TPAUC. We also observed that in Figure 1, across all datasets, there is an abrupt drop and subsequent rise in performance. This is due to the excessively large step size. Once the step size is reduced, training returns to normal.

Table 2: TPAUC on the test data of linear and deep models. $(\theta_0, \theta_1)$ represents TPR $\geq 1 - \theta_0$, FPR $\leq \theta_1$. Results are reported as mean(std).

| Metrics | Methods | Linear Model | | | Deep Model | | | |
|---|---|---|---|---|---|---|---|---|
| | | HIGGS | SUSY | ijcnn1 | molmuv(t1) | moltox21(t0) | nodulemnist3d | adrenalmnist3d |
| (0.5, 0.5) | CE | 0.041(0.001) | 0.300(0.010) | 0.230(0.017) | 0.715(0.166) | 0.267(0.042) | 0.657(0.037) | 0.507(0.094) |
| | AUCM | 0.122(0.001) | 0.512(0.015) | 0.487(0.098) | 0.722(0.114) | 0.279(0.038) | 0.672(0.021) | **0.554(0.022)** |
| | SOTAs | 0.108(0.001) | 0.484(0.001) | 0.637(0.030) | 0.821(0.110) | 0.325(0.030) | 0.688(0.019) | 0.498(0.090) |
| | PAUCI | 0.138(0.002) | 0.519(0.001) | 0.664(0.018) | 0.820(0.046) | 0.283(0.032) | 0.684(0.021) | 0.541(0.042) |
| | SONX | 0.110(0.009) | 0.516(0.001) | 0.633(0.094) | 0.865(0.061) | 0.286(0.023) | 0.654(0.035) | 0.540(0.042) |
| | **STACO** | **0.158(0.003)** | **0.520(0.001)** | **0.682(0.054)** | **0.904(0.048)** | **0.325(0.023)** | **0.707(0.005)** | 0.546(0.047) |
| (0.75, 0.75) | CE | 0.354(0.002) | 0.612(0.006) | 0.581(0.014) | 0.871(0.058) | 0.627(0.035) | 0.825(0.016) | 0.750(0.055) |
| | AUCM | 0.435(0.004) | 0.726(0.002) | 0.728(0.061) | 0.851(0.066) | 0.630(0.027) | 0.831(0.016) | 0.772(0.014) |
| | SOTAs | 0.441(0.002) | 0.746(0.009) | 0.813(0.016) | 0.821(0.070) | 0.614(0.056) | 0.838(0.012) | 0.763(0.054) |
| | PAUCI | 0.4742(0.003) | 0.749(0.007) | 0.830(0.030) | 0.883(0.024) | 0.616(0.030) | 0.823(0.014) | 0.7642(0.015) |
| | SONX | 0.447(0.009) | 0.748(0.000) | 0.810(0.049) | 0.927(0.029) | 0.626(0.028) | 0.832(0.013) | 0.772(0.021) |
| | **STACO** | **0.484(0.004)** | **0.752(0.000)** | **0.839(0.024)** | **0.945(0.024)** | **0.638(0.041)** | **0.856(0.003)** | **0.780(0.013)** |

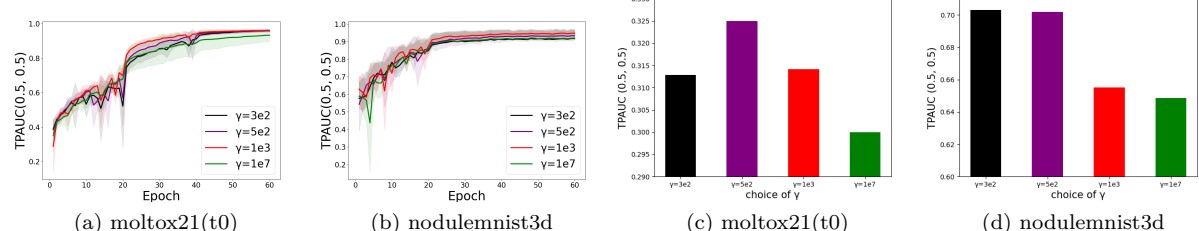

(a) moltox21(t0)  (b) nodulemnist3d  (c) moltox21(t0)  (d) nodulemnist3d

Figure 4: First two figures shows the TPAUC (0.5,0.5) training curves of STACO2 with different $\gamma$; last two figures shows the TPAUC (0.5, 0.5) testing results of STACO2 with different $\gamma$. The experiment is conducted on datasets ogbg-moltox21(t0) and nodulemnist3d.

In the nonlinear model experiments as shown in Figure 2, STACO2 demonstrates competitive performance in terms of training loss reduction across all four datasets compared to SONX and SOTA. In both the (0.5, 0.5) and (0.75, 0.75) settings, STACO2 achieves lower or comparable loss values while maintaining a stable training trajectory. These results indicate that STACO2 is effective in minimizing loss and optimizing model performance, further supporting its advantage over SONX and SOTA.

Due to space limit, we present more training results in Figure 5, 6 in Appendix B.

**Testing Results.** Under two different metrics, we present the testing results for linear and deep learning models in Table 2. For the linear model, STACO1 consistently outperforms the baseline methods across various datasets, demonstrating its robustness and strong generalization capability across different datasets and evaluation criteria. Similarly, for the nonlinear model, STACO2 achieves significant improvements over existing methods. Notably, compared to SONX, STACO2 exhibits a more pronounced advantage in testing performance than in training, suggesting superior generalization when optimizing the exact TPAUC loss.

We do not include SOTA (Zhu et al., 2022) in the above comparison, since SOTA is quite similar to STACO2 thus they have similar testing results. However, we must point out that the convergence of SOTA is much slower than STACO2 since it has to update all the coordinates of $\mathbf{s}$ in problem (2). As shown in Figure 2, STACO2 is significantly faster than SOTA.

## 5.3 Ablation Study

**Effect of Batch Size.** We examine the impact of negative batch size $B$ on the performance of STACO2 and SONX to verify the mini-batch speedup of STACO2 over SONX. Specifically, we tune the negative batch size $B$ in [16, 64, 256]. In Figure 3, we present the training loss curve for STACO2 and SONX on dataset ogbg-molmuv (t1). Our results show that as batch size increases, STACO2 exhibits greater convergence

improvement compared to SONX, indicating that it benefits more from a larger batch size. This observation is consistent with Theorem 4.5, i.e., STACO2 can achieve full mini-batch speedup than SONX.

**Effect of Epoch Decay Factor.** We examine the impact of epoch decay parameter $\gamma$ on the training performance of STACO2. In Theorem 4.5, $\gamma$ must be less or equal than $\frac{1}{2C_f \rho}$, where $C_f$ is the Lipschitz constant for function $f_i$ and $\rho$ is the weakly-convexity parameter for function $g_i$. In TPAUC maximization problem, $C_f$ is 1. However, $\rho$ in practice is difficult to determine. Therefore, we tune $\gamma$ in the range {300, 500, 1000} in the experiment. Additionally, we conduct $\gamma$ =1e7 case for our ablation study. Notably, STACO2 reduces to STACO1 if $\gamma$ equals an infinitely large number. The results are presented in Figure 4. We observe that an appropriate value of $\gamma$ can yield better training results, verifying Theorem 4.5 and demonstrating the importance of the epoch decay parameter $\gamma$ for primal-dual algorithms in deep learning.

**Effect of Surrogate Loss $\ell$.** We investigate how the choice of surrogate loss function $\ell$ influences the final experimental results. Specifically, we consider three common losses: square hinge loss, square loss, and hinge loss, and evaluate their performance across various datasets. The results show that our algorithm STACO performs consistently well and remains stable across different surrogate losses, indicating that the choice of $\ell$ has limited impact on the final performance.

Table 3: Comparison of performance metrics using different nonsmooth losses across datasets. Each entry is reported as mean(std).

| Methods | HIGGS | | | SUSY | | | ijcnn1 | | |
|---|---|---|---|---|---|---|---|---|---|
| | hinge square | square | hinge | hinge square | square | hinge | hinge square | square | hinge |
| CE | 0.354(0.002) | 0.376(0.003) | 0.341(0.004) | 0.612(0.004) | 0.590(0.004) | 0.639(0.005) | 0.581(0.003) | 0.560(0.004) | 0.604(0.003) |
| AUCM | 0.435(0.002) | 0.462(0.003) | 0.411(0.004) | 0.726(0.003) | 0.748(0.004) | 0.699(0.004) | 0.728(0.004) | 0.752(0.003) | 0.701(0.005) |
| SOTAs | 0.441(0.003) | 0.467(0.004) | 0.415(0.004) | 0.746(0.003) | 0.773(0.003) | 0.719(0.002) | 0.813(0.002) | 0.840(0.004) | 0.787(0.004) |
| PAUCI | 0.474(0.003) | 0.500(0.004) | 0.451(0.003) | 0.749(0.004) | 0.724(0.003) | **0.777(0.004)** | 0.830(0.002) | 0.857(0.003) | 0.808(0.003) |
| SONX | 0.447(0.003) | 0.472(0.003) | 0.420(0.004) | 0.748(0.003) | 0.773(0.003) | 0.723(0.003) | 0.810(0.002) | 0.785(0.003) | **0.836(0.004)** |
| STACO | **0.484(0.004)** | **0.511(0.003)** | **0.458(0.004)** | **0.752(0.000)** | **0.779(0.003)** | 0.725(0.004) | **0.839(0.024)** | **0.866(0.004)** | 0.812(0.004) |

## 5.4 Training Efficiency

To demonstrate the training efficiency of our algorithm, we compare the per-iteration runtime of STACO, PAUCI, SONX, and SOTA across four benchmark datasets, as shown in Table 4. STACO consistently achieves the lowest runtime per iteration across all datasets. Notably, it surpasses the second-best method, SONX, by a substantial margin, particularly on larger datasets such as molmuv and moltox21. These results highlight the superior computational efficiency of STACO, making it a compelling choice for large-scale or time-sensitive applications.

## 6 Conclusion

In this paper, we proposed two novel stochastic primal-dual double block-coordinate algorithms for optimizing two-way partial AUC (TPAUC), effectively addressing imbalanced data classification. By leveraging stochastic updates for both primal and dual variables, our methods achieve improved convergence rates in both convex and non-convex settings. Empirical results demonstrate faster convergence and superior generalization across benchmark datasets, establishing a new state-of-the-art in TPAUC optimization for real-world applications.

Table 4: Training time per iteration (in seconds) on different datasets.

| Methods | molmuv | moltox21 | nodulemnist3d | adrenalmnist3d |
|---|---|---|---|---|
| SOTA | 14.80 | 8.01 | 2.02 | 2.23 |
| SONX | 9.78 | 4.54 | 1.62 | 1.76 |
| PAUCI | 12.54 | 5.72 | 2.51 | 2.74 |
| STACO | **8.72** | **3.96** | **1.40** | **1.48** |

## Acknowledgments

We are grateful to the reviewers' comments. LZ, BW, TY were partially supported by NSF grants 2306572 and 2147253.

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

# A  Vanilla Algorithm

---

**Algorithm 3** Simplified STACO1

---

1: Initialize $\mathbf{u}_0 \in \mathcal{U}$, $\mathbf{s}_0 \in \mathcal{S}$, $\mathbf{y}_0 \in \mathcal{Y}$
2: **for** $t = 0, 1, \ldots, T - 1$ **do**
3:    Sample a batch $\mathcal{S}_t \subset \{1, \ldots, n\}$, $|\mathcal{S}_t| = S$
4:    **for** each $i \in \mathcal{S}_t$ **do**
5:       Sample independent size-$B$ mini-batches $\mathcal{B}_t^{(i)}, \tilde{\mathcal{B}}_t^{(i)}$ from $\mathbb{P}_i$
6:       Compute $\hat{g}_t^{(i)}(\mathcal{B}_t^{(i)}) = g_i(\mathbf{u}_t, \mathbf{s}_t^{(i)}; \mathcal{B}_t^{(i)})$
7:       Compute $\hat{G}_{t,1}^{(i)}(\tilde{\mathcal{B}}_t^{(i)}) \in \partial_{\mathbf{u}} g_i(\mathbf{u}_t, \mathbf{s}_t^{(i)}; \tilde{\mathcal{B}}_t^{(i)})$, $\hat{G}_{t,2}^{(i)}(\tilde{\mathcal{B}}_t^{(i)}) \in \partial_{\mathbf{s}^{(i)}} g_i(\mathbf{u}_t, \mathbf{s}_t^{(i)}; \tilde{\mathcal{B}}_t^{(i)})$
8:       $\mathbf{y}_{t+1}^{(i)} = \arg\max_{\mathbf{y}^{(i)} \in \mathcal{Y}_i} \left\{ \mathbf{y}^{(i)} \hat{g}_t^{(i)}(\mathcal{B}_t^{(i)}) - f_i^*(\mathbf{y}^{(i)}) - \frac{1}{2\alpha} \left( \mathbf{y}^{(i)} - \mathbf{y}_t^{(i)} \right)^2 \right\}$
9:       $\mathbf{s}_{t+1}^{(i)} = \mathbf{s}_t^{(i)} - \beta \frac{1}{S} \sum_{i \in \mathcal{S}_t} \mathbf{y}_{t+1}^{(i)} \hat{G}_{t,2}^{(i)}(\tilde{\mathcal{B}}_t^{(i)})$
10:    **end for**
11:    For each $i \notin \mathcal{S}_t$, $\mathbf{y}_{t+1}^{(i)} = \mathbf{y}_t^{(i)}, \mathbf{s}_{t+1}^{(i)} = \mathbf{s}_t^{(i)}$
12:    $\mathbf{u}_{t+1} = \mathbf{u}_t - \eta \frac{1}{S} \sum_{i \in \mathcal{S}_t} \mathbf{y}_{t+1}^{(i)} \hat{G}_{t,1}^{(i)}(\tilde{\mathcal{B}}_t^{(i)})$
13: **end for**
14: $\bar{\mathbf{u}} = \frac{1}{T} \sum_{t=0}^{T-1} \mathbf{u}_{t+1}, \bar{\mathbf{s}} = \frac{1}{T} \sum_{t=0}^{T-1} \mathbf{s}_{t+1}$
15: Return $\bar{\mathbf{u}}, \bar{\mathbf{s}}$

---

**Algorithm 4** Simplified STACO2

---

1: Initialize $\mathbf{u}_0 \in \mathcal{U}, \mathbf{s}_0 \in \mathcal{S}$
2: **for** $t = 0, 1, \ldots, T - 1$ **do**
3:    Initialize $\mathbf{y}_{t,0} \in \mathcal{Y}$
4:    Set $\mathbf{u}_{t,0} = \mathbf{u}_t, \mathbf{s}_{t,0} = \mathbf{s}_t$
5:    **for** $k = 0, 1, \ldots, K_t - 1$ **do**
6:       Sample a batch $\mathcal{S}_{t,k} \subset \{1, \ldots, n\}$, $|\mathcal{S}_{t,k}| = S$
7:       **for** each $i \in \mathcal{S}_{t,k}$ **do**
8:          Sample independent size-$B$ mini-batches $\mathcal{B}_{t,k}^{(i)}, \tilde{\mathcal{B}}_{t,k}^{(i)}$ from $\mathbb{P}_i$
9:          Compute $\hat{g}_{t,k}^{(i)}(\mathcal{B}_{t,k}^{(i)}) = g_i(\mathbf{u}_{t,k}, \mathbf{s}_{t,k}^{(i)}; \mathcal{B}_{t,k}^{(i)})$
10:         Compute $\hat{G}_{t,k,1}^{(i)}(\tilde{\mathcal{B}}_{t,k}^{(i)}) \in \partial_{\mathbf{u}} g_i(\mathbf{u}_{t,k}, \mathbf{s}_{t,k}^{(i)}; \tilde{\mathcal{B}}_{t,k}^{(i)}), \hat{G}_{t,k,2}^{(i)}(\tilde{\mathcal{B}}_{t,k}^{(i)}) \in \partial_{\mathbf{s}^{(i)}} g_i(\mathbf{u}_{t,k}, \mathbf{s}_{t,k}^{(i)}; \tilde{\mathcal{B}}_{t,k}^{(i)})$
11:         $\mathbf{y}_{t,k+1}^{(i)} = \arg\max_{\mathbf{y}^{(i)} \in \mathcal{Y}_i} \left\{ \mathbf{y}^{(i)} \hat{g}_{t,k}^{(i)}(\mathcal{B}_{t,k}^{(i)}) - f_i^*(\mathbf{y}^{(i)}) - \frac{1}{2\alpha_t} \left( \mathbf{y}^{(i)} - \mathbf{y}_{t,k}^{(i)} \right)^2 \right\}$
12:         $\mathbf{s}_{t,k+1}^{(i)} = \arg\min_{\mathbf{s}^{(i)} \in \mathcal{S}_i} \left\{ \left\langle \mathbf{s}^{(i)}, \frac{1}{S} \sum_{i \in \mathcal{S}_{t,k}} \mathbf{y}_{t,k+1}^{(i)} \hat{G}_{t,k,2}^{(i)}(\tilde{\mathcal{B}}_{t,k}^{(i)}) + \frac{1}{\gamma}(\mathbf{s}_{t,k}^{(i)} - \mathbf{s}_{t,0}^{(i)}) \right\rangle + \frac{1}{2\beta_t} \left( \mathbf{s}^{(i)} - \mathbf{s}_{t,k}^{(i)} \right)^2 \right\}$
13:      **end for**
14:      For each $i \notin \mathcal{S}_{t,k}$, $\mathbf{y}_{t,k+1}^{(i)} = \mathbf{y}_{t,k}^{(i)}, \mathbf{s}_{t,k+1}^{(i)} = \mathbf{s}_{t,k}^{(i)}$
15:      $\mathbf{u}_{t,k+1} = \arg\min_{\mathbf{u} \in \mathcal{U}} \left\{ \left\langle \mathbf{u}, \frac{1}{S} \sum_{i \in \mathcal{S}_{t,k}} \mathbf{y}_{t,k+1}^{(i)} \hat{G}_{t,k,1}^{(i)}(\tilde{\mathcal{B}}_{t,k}^{(i)}) + \frac{1}{\gamma}(\mathbf{u}_{t,k} - \mathbf{u}_{t,0}) \right\rangle + \frac{1}{2\eta_t} \|\mathbf{u} - \mathbf{u}_{t,k}\|_2^2 \right\}$
16:    **end for**
17:    Compute $\bar{\mathbf{u}}_t = \frac{1}{K_t} \sum_{k=0}^{K_t-1} \mathbf{u}_{t,k+1}, \bar{\mathbf{s}}_t = \frac{1}{K_t} \sum_{k=0}^{K_t-1} \mathbf{s}_{t,k+1}$
18:    Set $\mathbf{u}_{t+1} = \bar{\mathbf{u}}_t, \mathbf{s}_{t+1} = \bar{\mathbf{s}}_t$
19: **end for**
20: Return $\mathbf{u}_T, \mathbf{s}_T$

---

Table 5: Datasets Statistics (for nodulemnist3d and adrenalmnist3d, we follow the given training, validation and testing split). The percentage in parenthesis represents the proportion of positive samples.

| Dataset | Train (Validation) | Test |
|---|---|---|
| HIGGS | 4157561 (0.5%) | 1039299 (0.5%) |
| SUSY | 2181312 (0.5%) | 544489 (0.5%) |
| ijcnn1 | 49990 (9.71%) | 91701 (9.5%) |
| ogbg-moltox21 (t0) | 6556 (4.2%) | 709 (4.5%) |
| ogbg-molmuv (t1) | 13025 (0.17%) | 1709 (0.35%) |
| nodulemnist3d | 1,158 (25.4%) / 165 (25.4%) | 310 (20.6%) |
| adrenalmnist3d | 1,188 (21.8%) / 98 (22.4%) | 298 (23.1%) |

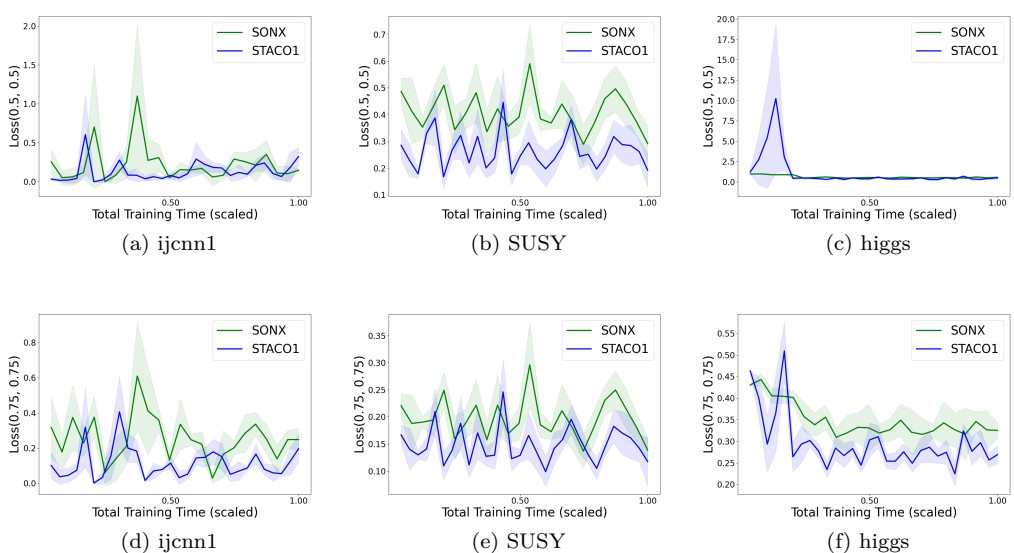

Figure 5: Training loss Curves of STACO1 and SONX on three different datasets. The first row shows the Loss (0.5, 0.5) results, and the second row shows the Loss (0.75, 0.75) results.

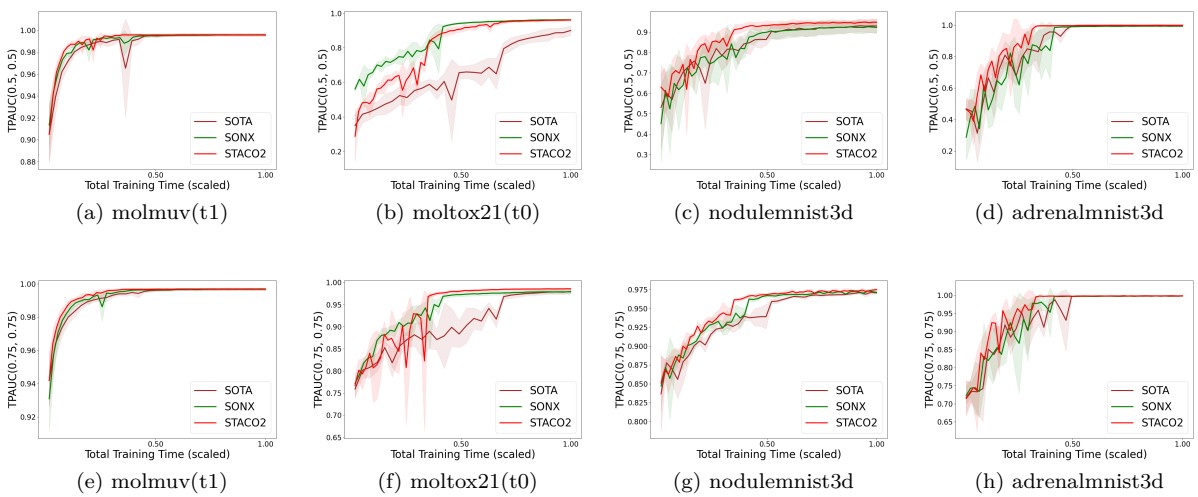

Figure 6: Training TPAUC Curves of STACO2, SOTA, and SONX on four different datasets. The first row shows the TPAUC (0.5, 0.5) results, and the second row shows the TPAUC (0.75, 0.75) results.

# B   More Experiment Results

## B.1   Additional plots for training loss curves

Figure 5 presents the training loss curves of STACO1 and SONX across three different datasets under two evaluation settings, $(0.5, 0.5)$ and $(0.75, 0.75)$. The results indicate that STACO1 consistently achieves lower and more stable loss values compared to SONX across all datasets. Notably, the variance in training loss is lower for STACO1, suggesting improved stability during optimization. The only exception is Figure 5c, which corresponds to optimizing the loss with $(0.5, 0.5)$ weights. We believe this is a limitation of the primal-dual algorithm, which involves two learning rates. In practical applications, improper tuning of these rates may lead to training instability. Besides, it is important to note that the loss curve is less stable compared to deep learning experiments, primarily due to the absence of pretraining for the linear model.

## B.2   Additional plots for training TPAUC curves

Figure 6 presents the training TPAUC curves for STACO2, SOTA, and SONX across the four datasets. In both TPAUC $(0.5, 0.5)$ and TPAUC $(0.75, 0.75)$ settings, STACO2 demonstrates competitive performance compared to SOTA and SONX, with better stability and faster convergence speed. Specifically, in some cases, STACO2 achieves superior results, particularly in later training stages, indicating its effectiveness in optimizing TPAUC objectives.

# C   Proof

## C.1   Preliminary Lemmas

Throughout the proof, for a space $\mathcal{X}$, we define its diameter with respect to the measure $\psi(\cdot) = \frac{1}{2}\|\cdot\|_2^2$ as $D_{\mathcal{X}} \coloneqq \left[\max_{\mathbf{x}\in\mathcal{X}}\psi(\mathbf{x}) - \min_{\mathbf{x}\in\mathcal{X}}\psi(\mathbf{x})\right]^{1/2}$. Besides, $a \asymp b$ means that there exists $c, C > 0$ such that $cb \le a \le Cb$. We first present a lemma here that will be useful in our later analysis.

**Lemma C.1** (Lemma 4 in Wang & Yang). *Suppose that the function $\phi : \mathcal{X} \to \mathbb{R}$ is on a convex, closed domain $\mathcal{X}$ and $\phi$ is $\mu$-convex with respect to Euclidean distance function $d(\mathbf{x}, \mathbf{y}) \coloneqq \frac{1}{2}\|\mathbf{x} - \mathbf{y}\|_2^2$ for any $\mathbf{x}, \mathbf{x}' \in \mathcal{X}$, i.e., $\phi(\mathbf{x}) \ge \phi(\mathbf{x}') + \langle \phi'(\mathbf{x}'), \mathbf{x} - \mathbf{x}'\rangle + \mu d(\mathbf{x}, \mathbf{x}')$, $\forall \mathbf{x}, \mathbf{x}' \in \mathcal{X}$. For $\hat{\mathbf{x}} = \arg\min_{\mathbf{x}\in\mathcal{X}}\{\phi(\mathbf{x}) + \eta d(\underline{\mathbf{x}}, \mathbf{x})\}$, we obtain*

$$\phi(\hat{\mathbf{x}}) - \phi(\mathbf{x}) \le \eta d(\mathbf{x}, \underline{\mathbf{x}}) - (\eta + \mu)d(\mathbf{x}, \hat{\mathbf{x}}) - \eta d(\hat{\mathbf{x}}, \underline{\mathbf{x}}), \quad \forall \mathbf{x} \in \mathcal{X}. \tag{21}$$

## C.2   Convex Case

In this section, we present the proof of the convex case. We begin by defining virtual sequences for Algorithm 3. The virtual sequences $\bar{\mathbf{y}}$ and $\bar{\mathbf{s}}$ are calculated with full coordinates, which is easier to bound in analyze. Thus, we also hope to bound the difference between true sequences and virtual sequences.

**Definition C.2** (virtual sequence). In Algorithm 3, a virtual sequence $\{\bar{\mathbf{y}}_t\}$ is defined as follows:

$$\bar{\mathbf{y}}_{t+1}^{(i)} = \arg\max_{\mathbf{y}^{(i)}\in\mathcal{Y}_i}\left\{\mathbf{y}^{(i)}\hat{g}_t^{(i)}(\mathcal{B}_t^{(i)}) - f_i^*(\mathbf{y}^{(i)}) - \frac{1}{2\alpha}\left(\mathbf{y}^{(i)} - \mathbf{y}_t^{(i)}\right)^2\right\} \quad i \in [n]. \tag{22}$$

Additionally, a virtual sequence $\{\bar{\mathbf{s}}_t\}$ is defined as follows:

$$\bar{\mathbf{s}}_{t+1}^{(i)} = \arg\min_{\mathbf{s}^{(i)}\in\mathcal{S}_i}\left\{\left(\mathbf{y}_{t+1}^{(i)}\hat{G}_{t,2}^{(i)}(\tilde{\mathcal{B}}_t^{(i)})\right)\cdot\mathbf{s}^{(i)} + \frac{1}{2\beta}\left(\mathbf{s}^{(i)} - \mathbf{s}_t^{(i)}\right)^2\right\} \quad i \in [n]. \tag{23}$$

Next, we present a useful lemma, which is helpful in bounding $\mathbf{y}$ related error term.

**Lemma C.3** (Lemma 9 in Wang & Yang). *Suppose* $\{\bar{\mathbf{y}}_t\}, \{\hat{\mathbf{y}}_t\}$ *are virtual sequences for any* $t \geq 0$ *in Algorithm 3. Then, for any* $\lambda_1 > 0, \mathbf{y} \in \mathcal{Y}$, *it follows that:*

$$\mathbf{E}\left[\frac{1}{2n\alpha}\left(\|\mathbf{y} - \mathbf{y}_t\|_2^2 - \|\mathbf{y} - \bar{\mathbf{y}}_t\|_2^2 - \|\bar{\mathbf{y}}_t - \mathbf{y}_t\|_2^2\right)\right] \leq \frac{1}{2\alpha S}\left(\|\mathbf{y} - \mathbf{y}_t\|_2^2 - \|\mathbf{y} - \mathbf{y}_{t+1}\|_2^2\right) + \frac{\lambda_1}{2\alpha S}\left(\|\mathbf{y} - \hat{\mathbf{y}}_t\|_2^2 - \|\mathbf{y} - \hat{\mathbf{y}}_{t+1}\|_2^2\right)$$
$$- \frac{1}{2\alpha n}(1 - \frac{1}{\lambda_1 S})\|\bar{\mathbf{y}}_{t+1} - \mathbf{y}_t\|_2^2. \tag{24}$$

We define that $\mathcal{G}_t$ is the $\sigma$-algebra generated by $\{\mathcal{B}_0, \mathcal{S}_0, \cdots, \mathcal{B}_{t-1}, \mathcal{S}_{t-1}, \mathcal{B}_t\}$ and $\mathcal{F}_t$ is the $\sigma$-algebra generated by $\{\mathcal{B}_0, \mathcal{S}_0, \cdots, \mathcal{B}_{t-1}, \mathcal{S}_{t-1}, \mathcal{B}_t, \mathcal{S}_t\}$. Note that $\mathcal{G}_t \subset \mathcal{F}_t$ and $\mathbf{y}_{t+1}$ is $\mathcal{F}_t$-measurable. Now we proceed to show the descent lemma.

**Lemma C.4** (Descent Lemma). *Under Assumption 4.1 and 4.2, suppose* $\{\bar{\mathbf{y}}_t\}, \{\tilde{\mathbf{y}}_t\}, \{\hat{\mathbf{y}}_t\}, \{\bar{\mathbf{s}}_t\}$ *are virtual sequences for Algorithm 3. Then, for any* $t \in [0, T-1]$, *taking expectation over* $\mathcal{F}_t$, *it holds that:*

$$\mathbf{E}\left[L(\mathbf{u}_{t+1}, \mathbf{s}_{t+1}, \mathbf{y}) - L(\mathbf{u}, \mathbf{s}, \bar{\mathbf{y}}_{t+1})\right]$$
$$\leq \frac{1}{2\eta}\left(\|\mathbf{u} - \mathbf{u}_t\|_2^2 - \mathbf{E}\|\mathbf{u} - \mathbf{u}_{t+1}\|_2^2\right) + \frac{1}{2\beta S}\left(\|\mathbf{s} - \mathbf{s}_t\|_2^2 - \mathbf{E}\|\mathbf{s} - \mathbf{s}_{t+1}\|_2^2\right) + \frac{1}{\alpha S}\left(\|\mathbf{y} - \mathbf{y}_t\|_2^2 - \mathbf{E}\|\mathbf{y} - \mathbf{y}_{t+1}\|_2^2\right)$$
$$+ \frac{1}{\alpha S}\left(\|\mathbf{y} - \hat{\mathbf{y}}_t\|_2^2 - \mathbf{E}\|\mathbf{y} - \hat{\mathbf{y}}_{t+1}\|_2^2\right) + \frac{1}{\alpha S}\left(\|\mathbf{y} - \tilde{\mathbf{y}}_t\|_2^2 - \mathbf{E}\|\mathbf{y} - \tilde{\mathbf{y}}_{t+1}\|_2^2\right)$$
$$+ 64\Omega C_f^2 C_g^2 + \frac{S\alpha\sigma_0^2}{2Bn} + \frac{\alpha\sigma_0^2}{2B} + \frac{\eta C_f^2 \sigma_1^2}{B} + \frac{\eta \delta^2}{S} + \frac{\beta C_f^2 \sigma_2^2}{B}, \tag{25}$$

*where* $\Omega = \max\{\eta, \beta\}$.

*Proof.* See Appendix C.2.1. $\qquad\qquad\square$

### C.2.1 Proof of Lemma C.4

*Proof.* By Definition, we have

$$L(\mathbf{u}_{t+1}, \mathbf{s}_{t+1}, \mathbf{y}) - L(\mathbf{u}, \mathbf{s}, \bar{\mathbf{y}}_{t+1})$$
$$= \frac{1}{n}\sum_{i=1}^n\left(\mathbf{y}^{(i)}g_i(\mathbf{u}_{t+1}, \mathbf{s}_{t+1}) - f_i^*(\mathbf{y}^{(i)})\right) - \frac{1}{n}\sum_{i=1}^n\left(\bar{\mathbf{y}}_{t+1}^{(i)}g_i(\mathbf{u}, \mathbf{s}) - f_i^*(\bar{\mathbf{y}}_{t+1}^{(i)})\right)$$
$$= \frac{1}{n}\sum_{i=1}^n(\mathbf{y}^{(i)} - \bar{\mathbf{y}}_{t+1}^{(i)})g_i(\mathbf{u}_{t+1}, \mathbf{s}_{t+1}) - \frac{1}{n}\sum_{i=1}^n f_i^*(\mathbf{y}^{(i)}) + \frac{1}{n}\sum_{i=1}^n f_i^*(\bar{\mathbf{y}}_{t+1}^{(i)})$$
$$+ \frac{1}{n}\sum_{i=1}^n \bar{\mathbf{y}}_{t+1}^{(i)}\left(g_i(\mathbf{u}_{t+1}, \mathbf{s}_{t+1}) - g_i(\mathbf{u}_t, \mathbf{s}_t)\right) + \frac{1}{n}\sum_{i=1}^n \bar{\mathbf{y}}_{t+1}^{(i)}\left(g_i(\mathbf{u}_t, \mathbf{s}_t) - g_i(\mathbf{u}, \mathbf{s})\right). \tag{26}$$

Using the convexity of $g_i$, we obtain the following upper bound:

$$L(\mathbf{u}_{t+1}, \mathbf{s}_{t+1}, \mathbf{y}) - L(\mathbf{u}, \mathbf{s}, \bar{\mathbf{y}}_{t+1})$$
$$\leq \underbrace{\frac{1}{n}\sum_{i=1}^n(\mathbf{y}^{(i)} - \bar{\mathbf{y}}_{t+1}^{(i)})g_i(\mathbf{u}_{t+1}, \mathbf{s}_{t+1}) - \frac{1}{n}\sum_{i=1}^n f_i^*(\mathbf{y}^{(i)}) + \frac{1}{n}\sum_{i=1}^n f_i^*(\bar{\mathbf{y}}_{t+1}^{(i)})}_{\text{I}}$$
$$+ \frac{1}{n}\sum_{i=1}^n \bar{\mathbf{y}}_{t+1}^{(i)}\left(g_i(\mathbf{u}_{t+1}, \mathbf{s}_{t+1}) - g_i(\mathbf{u}_t, \mathbf{s}_t)\right) + \underbrace{\frac{1}{n}\sum_{i=1}^n \bar{\mathbf{y}}_{t+1}^{(i)}\left(\left\langle G_{t,1}^{(i)}, \mathbf{u}_t - \mathbf{u}\right\rangle + G_{t,2}^{(i)}(\mathbf{s}_t - \mathbf{s})\right)}_{\text{II}}. \tag{27}$$

We now analyze terms I and II separately. For term I, we decompose as follows:

$$\text{I} = \frac{1}{n}\sum_{i=1}^n(\mathbf{y}^{(i)} - \bar{\mathbf{y}}_{t+1}^{(i)})g_i(\mathbf{u}_{t+1}, \mathbf{s}_{t+1})$$
$$= \frac{1}{n}\sum_{i=1}^n(\mathbf{y}^{(i)} - \bar{\mathbf{y}}_{t+1}^{(i)})\hat{g}_t^{(i)}(\mathcal{B}_t^{(i)}) + \frac{1}{n}\sum_{i=1}^n(\mathbf{y}^{(i)} - \bar{\mathbf{y}}_{t+1}^{(i)})\left(g_i(\mathbf{u}_{t+1}, \mathbf{s}_{t+1}) - \hat{g}_t^{(i)}(\mathcal{B}_t^{(i)})\right)$$

For term II, by decomposition we obtain:

$$\text{II} = \left\langle \frac{1}{S}\sum_{i\in\mathcal{S}_t}\mathbf{y}_{t+1}^{(i)}\hat{G}_{t,1}^{(i)}(\tilde{\mathcal{B}}_t^{(i)}) - \frac{1}{n}\sum_{i=1}^n \bar{\mathbf{y}}_{t+1}^{(i)}G_{t,1}^{(i)}, \mathbf{u}-\mathbf{u}_{t+1}\right\rangle - \left\langle \frac{1}{S}\sum_{i\in\mathcal{S}_t}\mathbf{y}_{t+1}^{(i)}\hat{G}_{t,1}^{(i)}(\tilde{\mathcal{B}}_t^{(i)}), \mathbf{u}-\mathbf{u}_{t+1}\right\rangle$$

$$+ \frac{1}{n}\sum_{i=1}^n \left\langle \bar{\mathbf{y}}_{t+1}^{(i)}G_{t,1}^{(i)}, \mathbf{u}_t-\mathbf{u}_{t+1}\right\rangle$$

$$+ \frac{1}{n}\sum_{i=1}^n \left\langle \bar{\mathbf{y}}_{t+1}^{(i)}\left(\hat{G}_{t,2}^{(i)}(\tilde{\mathcal{B}}_t^{(i)}) - G_{t,2}^{(i)}\right), \mathbf{s}-\mathbf{s}_{t+1}\right\rangle - \frac{1}{n}\sum_{i=1}^n \left\langle \bar{\mathbf{y}}_{t+1}^{(i)}\hat{G}_{t,2}^{(i)}(\tilde{\mathcal{B}}_t^{(i)}), \mathbf{s}-\mathbf{s}_{t+1}\right\rangle$$

$$+ \frac{1}{n}\sum_{i=1}^n \left\langle \bar{\mathbf{y}}_{t+1}^{(i)}G_{t,2}^{(i)}, \mathbf{s}_t-\mathbf{s}_{t+1}\right\rangle. \tag{28}$$

Combining all terms above, we arrive at the key inequality:

$$L(\mathbf{u}_{t+1}, \mathbf{s}_{t+1}, \mathbf{y}) - L(\mathbf{u}, \mathbf{s}, \bar{\mathbf{y}}_{t+1})$$

$$\leq \underbrace{\frac{1}{n}\sum_{i=1}^n (\mathbf{y}^{(i)} - \bar{\mathbf{y}}_{t+1}^{(i)})\hat{g}_t^{(i)}(\mathcal{B}_t^{(i)}) - \frac{1}{n}\sum_{i=1}^n f_i^*(\mathbf{y}^{(i)}) + \frac{1}{n}\sum_{i=1}^n f_i^*(\bar{\mathbf{y}}_{t+1}^{(i)})}_{\mathcal{C}_1}$$

$$+ \frac{1}{n}\sum_{i=1}^n (\mathbf{y}^{(i)} - \bar{\mathbf{y}}_{t+1}^{(i)})\left(g_i(\mathbf{u}_{t+1}, \mathbf{s}_{t+1}) - \hat{g}_t^{(i)}(\mathcal{B}_t^{(i)})\right) + \frac{1}{n}\sum_{i=1}^n \bar{\mathbf{y}}_{t+1}^{(i)}\left(g_i(\mathbf{u}_{t+1}, \mathbf{s}_{t+1}) - g_i(\mathbf{u}_t, \mathbf{s}_{t,k})\right)$$

$$+ \underbrace{\left\langle \frac{1}{S}\sum_{i\in\mathcal{S}_t}\mathbf{y}_{t+1}^{(i)}\hat{G}_{t,1}^{(i)}(\tilde{\mathcal{B}}_t^{(i)}) - \frac{1}{n}\sum_{i=1}^n \bar{\mathbf{y}}_{t+1}^{(i)}G_{t,1}^{(i)}, \mathbf{u}-\mathbf{u}_{t+1}\right\rangle - \left\langle \frac{1}{S}\sum_{i\in\mathcal{S}_t}\mathbf{y}_{t+1}^{(i)}\hat{G}_{t,1}^{(i)}(\tilde{\mathcal{B}}_t^{(i)}), \mathbf{u}-\mathbf{u}_{t+1}\right\rangle}_{\mathcal{C}_2}$$

$$+ \frac{1}{n}\sum_{i=1}^n \left\langle \bar{\mathbf{y}}_{t+1}^{(i)}G_{t,1}^{(i)}, \mathbf{u}_t-\mathbf{u}_{t+1}\right\rangle$$

$$+ \frac{1}{n}\sum_{i=1}^n \left\langle \bar{\mathbf{y}}_{t+1}^{(i)}\left(\hat{G}_{t,2}^{(i)}(\tilde{\mathcal{B}}_t^{(i)}) - G_{t,2}^{(i)}\right), \mathbf{s}-\mathbf{s}_{t+1}\right\rangle \underbrace{- \frac{1}{n}\sum_{i=1}^n \left\langle \bar{\mathbf{y}}_{t+1}^{(i)}\hat{G}_{t,2}^{(i)}(\tilde{\mathcal{B}}_t^{(i)}), \mathbf{s}-\mathbf{s}_{t+1}\right\rangle}_{\mathcal{C}_3}$$

$$+ \frac{1}{n}\sum_{i=1}^n \left\langle \bar{\mathbf{y}}_{t+1}^{(i)}G_{t,2}^{(i)}, \mathbf{s}_t-\mathbf{s}_{t+1}\right\rangle. \tag{29}$$

We now analyze the upper bounds of $\mathcal{C}_1$, $\mathcal{C}_2$, and $\mathcal{C}_3$ in turn. For $\mathcal{C}_1$, invoking Lemma C.1 and Lemma C.3, it holds that

$$\mathcal{C}_1 \underset{\text{Lemma C.1}}{\leq} \left[\frac{1}{2n\alpha}\left(\|\mathbf{y}-\mathbf{y}_t\|_2^2 - \|\mathbf{y}-\bar{\mathbf{y}}_{t+1}\|_2^2 - \|\bar{\mathbf{y}}_{t+1}-\mathbf{y}_t\|_2^2\right)\right]$$

$$\underset{\text{Lemma C.3}}{\leq} \frac{1}{2\alpha S}\left(\|\mathbf{y}-\mathbf{y}_t\|_2^2 - \|\mathbf{y}-\mathbf{y}_{t+1}\|_2^2\right) + \frac{\lambda_1}{2\alpha S}\left(\|\mathbf{y}-\hat{\mathbf{y}}_t\|_2^2 - \|\mathbf{y}-\hat{\mathbf{y}}_{t+1}\|_2^2\right)$$

$$- \frac{1}{2\alpha_t n}(1 - \frac{1}{\lambda_1 S})\|\bar{\mathbf{y}}_{t+1}-\mathbf{y}_t\|_2^2. \tag{30}$$

For $\mathcal{C}_2$, noticing here we have

$$\phi(\mathbf{u}) := \left\langle \frac{1}{S}\sum_{i\in\mathcal{S}_t}\mathbf{y}_{t+1}^{(i)}\hat{G}_{t,1}^{(i)}(\tilde{\mathcal{B}}_t^{(i)}), \mathbf{u}\right\rangle, \tag{31}$$

and $\phi(\cdot)$ is convex in Lemma C.1, it follows that

$$\mathcal{C}_2 \underset{\text{Lemma C.1}}{\leq} \frac{1}{2\eta}\left(\|\mathbf{u}-\mathbf{u}_t\|_2^2 - \|\mathbf{u}-\mathbf{u}_{t+1}\|_2^2\right) - \frac{1}{2\eta}\|\mathbf{u}_{t+1}-\mathbf{u}_t\|_2^2. \tag{32}$$

For $\mathcal{C}_3$, in a similar manner, we can obtain

$$\mathcal{C}_3 \underset{\text{Lemma C.1}}{\leq} \frac{1}{2n\beta}\left(\|\mathbf{s}-\mathbf{s}_t\|_2^2 - \|\mathbf{s}-\bar{\mathbf{s}}_{t+1}\|_2^2\right) - \frac{1}{2n\beta}\|\bar{\mathbf{s}}_{t+1}-\mathbf{s}_t\|_2^2. \tag{33}$$

Substituting the above inequality into (29), and taking expectation over $\mathcal{F}_t$, we can get

$$\mathbf{E}\left[L(\mathbf{u}_{t+1}, \mathbf{s}_{t+1}, \mathbf{y}) - L(\mathbf{u}, \mathbf{s}, \bar{\mathbf{y}}_{t+1})\right]$$

$$\leq \frac{1}{2\alpha S}\left(\|\mathbf{y} - \mathbf{y}_t\|_2^2 - \mathbf{E}\|\mathbf{y} - \mathbf{y}_{t+1}\|_2^2\right) + \frac{\lambda_1}{2\alpha S}\left(\|\mathbf{y} - \hat{\mathbf{y}}_t\|_2^2 - \mathbf{E}\|\mathbf{y} - \hat{\mathbf{y}}_{t+1}\|_2^2\right) - \frac{1}{2\alpha n}(1 - \frac{1}{\lambda_1 S})\mathbf{E}\|\bar{\mathbf{y}}_{t+1} - \mathbf{y}_t\|_2^2$$

$$+ \frac{1}{2\eta}\left(\|\mathbf{u} - \mathbf{u}_t\|_2^2 - \mathbf{E}\|\mathbf{u} - \mathbf{u}_{t+1}\|_2^2\right) - \frac{1}{2\eta}\mathbf{E}\|\mathbf{u}_{t+1} - \mathbf{u}_t\|_2^2 + \underbrace{\frac{1}{2n\beta}\left(\|\mathbf{s} - \mathbf{s}_t\|_2^2 - \mathbf{E}\|\mathbf{s} - \bar{\mathbf{s}}_{t+1}\|_2^2\right) - \frac{1}{2n\beta}\mathbf{E}\|\bar{\mathbf{s}}_{t+1} - \mathbf{s}_t\|_2^2}_{\mathcal{D}_1}$$

$$+ \underbrace{\frac{1}{n}\sum_{i=1}^n \mathbf{E}\left[(\mathbf{y}^{(i)} - \bar{\mathbf{y}}_{t+1}^{(i)})\left(g_i(\mathbf{u}_{t+1}, \mathbf{s}_{t+1}) - \hat{g}_t^{(i)}(\mathcal{B}_t^{(i)})\right)\right]}_{\mathcal{D}_2} + \underbrace{\frac{1}{n}\sum_{i=1}^n \mathbf{E}\left[\bar{\mathbf{y}}_{t+1}^{(i)}\left(g_i(\mathbf{u}_{t+1}, \mathbf{s}_{t+1}) - g_i(\mathbf{u}_t, \mathbf{s}_t)\right)\right]}_{\mathcal{D}_3}$$

$$+ \underbrace{\mathbf{E}\left\langle\frac{1}{n}\sum_{i=1}^n \bar{\mathbf{y}}_{t+1}^{(i)} G_{t,1}^{(i)}, \mathbf{u}_t - \mathbf{u}_{t+1}\right\rangle + \mathbf{E}\left\langle\frac{1}{n}\sum_{i=1}^n \bar{\mathbf{y}}_{t+1}^{(i)} G_{t,2}^{(i)}, \mathbf{s}_t - \mathbf{s}_{t+1}\right\rangle}_{\mathcal{D}_4}$$

$$+ \underbrace{\mathbf{E}\left\langle\frac{1}{S}\sum_{i\in\mathcal{S}_t} \mathbf{y}_{t+1}^{(i)} \hat{G}_{t,1}^{(i)}(\tilde{\mathcal{B}}_t^{(i)}) - \frac{1}{n}\sum_{i=1}^n \bar{\mathbf{y}}_{t+1}^{(i)} G_{t,1}^{(i)}, \mathbf{u} - \mathbf{u}_{t+1}\right\rangle}_{\mathcal{D}_5} + \underbrace{\frac{1}{n}\sum_{i=1}^n \left\langle\bar{\mathbf{y}}_{t+1}^{(i)}\left(\hat{G}_{t,2}^{(i)}(\tilde{\mathcal{B}}_t^{(i)}) - G_{t,2}^{(i)}\right), \mathbf{s} - \mathbf{s}_{t+1}\right\rangle}_{\mathcal{D}_6}. \quad (34)$$

For $\mathcal{D}_1$, noticing that $\mathbf{E}\left[(\mathbf{s}^{(i)} - \bar{\mathbf{s}}_{t+1}^{(i)})^2\right] = \frac{S}{n}(\mathbf{s}^{(i)} - \bar{\mathbf{s}}_{t+1}^{(i)})^2 + \frac{n-S}{n}(\mathbf{s}^{(i)} - \mathbf{s}_t^{(i)})^2$ for any $i \in [n]$, then we obtain

$$\mathcal{D}_1 \leq \frac{1}{2S\beta}\left(\|\mathbf{s} - \mathbf{s}_t\|_2^2 - \mathbf{E}\|\mathbf{s} - \mathbf{s}_{t+1}\|_2^2\right) - \frac{1}{2n\beta}\mathbf{E}\|\bar{\mathbf{s}}_{t+1} - \mathbf{s}_t\|_2^2. \quad (35)$$

Inspired by Lemma 10 in Wang & Yang, we bound $\mathcal{D}_2$ as following.

$$\mathcal{D}_2 = \frac{1}{n}\sum_{i=1}^n \mathbf{E}\left[(\mathbf{y}^{(i)} - \bar{\mathbf{y}}_{t+1}^{(i)})\left(g_i(\mathbf{u}_{t+1}, \mathbf{s}_{t+1}) - \hat{g}_t^{(i)}(\mathcal{B}_t^{(i)})\right)\right]$$

$$= \frac{1}{n}\sum_{i=1}^n \mathbf{E}\left[(\mathbf{y}^{(i)} - \bar{\mathbf{y}}_{t+1}^{(i)})\left(g_i(\mathbf{u}_{t+1}, \mathbf{s}_{t+1}) - g_i(\mathbf{u}_t, \mathbf{s}_t)\right)\right] - \frac{1}{n}\sum_{i=1}^n \mathbf{E}\left[(\mathbf{y}^{(i)} - \bar{\mathbf{y}}_{t+1}^{(i)})\left(g_i(\mathbf{u}_t, \mathbf{s}_t) - \hat{g}_t^{(i)}(\mathcal{B}_t^{(i)})\right)\right]$$

$$\leq \frac{1}{n}\sum_{i=1}^n \mathbf{E}\left[\left\|\mathbf{y}^{(i)} - \bar{\mathbf{y}}_{t+1}^{(i)}\right\|_2 \|g_i(\mathbf{u}_{t+1}, \mathbf{s}_{t+1}) - g_i(\mathbf{u}_t, \mathbf{s}_t)\|_2\right] - \frac{1}{n}\sum_{i=1}^n \mathbf{E}\left[(\mathbf{y}^{(i)} - \bar{\mathbf{y}}_{t+1}^{(i)})\left(g_i(\mathbf{u}_t, \mathbf{s}_t) - \hat{g}_t^{(i)}(\mathcal{B}_t^{(i)})\right)\right]$$

$$\leq \frac{C_g^2}{\lambda_4}\mathbf{E}\|\mathbf{u}_{t+1} - \mathbf{u}_t\|_2^2 + \frac{SC_g^2}{\lambda_4 n^2}\mathbf{E}\|\bar{\mathbf{s}}_{t+1} - \mathbf{s}_t\|_2^2 + 4\lambda_4 C_f^2 - \frac{1}{n}\sum_{i=1}^n \mathbf{E}\left[(\mathbf{y}^{(i)} - \bar{\mathbf{y}}_{t+1}^{(i)})\left(g_i(\mathbf{u}_t, \mathbf{s}_t) - \hat{g}_t^{(i)}(\mathcal{B}_t^{(i)})\right)\right]. \quad (36)$$

The last term in (36) is bounded as

$$\frac{1}{n}\sum_{i=1}^n \mathbf{E}\left[(\mathbf{y}^{(i)} - \bar{\mathbf{y}}_{t+1}^{(i)})\left(g_i(\mathbf{u}_t, \mathbf{s}_t) - \hat{g}_t^{(i)}(\mathcal{B}_t^{(i)})\right)\right]$$

$$= \frac{1}{n}\sum_{i=1}^n \mathbf{E}\left[(\mathbf{y}^{(i)} - \mathbf{y}_t^{(i)})\left(g_i(\mathbf{u}_{t+1}, \mathbf{s}_{t+1}) - \hat{g}_t^{(i)}(\mathcal{B}_t^{(i)})\right)\right] - \frac{1}{n}\sum_{i=1}^n \mathbf{E}\left[(\mathbf{y}_t^{(i)} - \bar{\mathbf{y}}_{t+1}^{(i)})\left(g_i(\mathbf{u}_{t+1}, \mathbf{s}_{t+1}) - \hat{g}_t^{(i)}(\mathcal{B}_t^{(i)})\right)\right]. \quad (37)$$

To bound the first term in (37), we have $\mathbf{E}\left[\mathbf{y}_t^{(i)}\left(g_i(\mathbf{u}_{t+1}, \mathbf{s}_{t+1}) - \hat{g}_t^{(i)}(\mathcal{B}_t^{(i)})\right)|\mathcal{F}_t\right] = 0$. Besides, according to Corollary 12 in Juditsky et al. (2011), for some $\lambda_2 > 0$, we have

$$\mathbf{E}\left[\mathbf{y}^{(i)}\left(g_i(\mathbf{u}_{t+1}, \mathbf{s}_{t+1}) - \hat{g}_t^{(i)}(\mathcal{B}_t^{(i)})\right)\right] \leq \lambda_2 \mathbf{E}\left\|\mathbf{y}^{(i)} - \tilde{\mathbf{y}}_t^{(i)}\right\|_2^2 - \lambda_2 \mathbf{E}\left\|\mathbf{y}^{(i)} - \tilde{\mathbf{y}}_t^{(i)}\right\|_2^2 + \frac{1}{2\lambda_2}\mathbf{E}\left\|g_i(\mathbf{u}_{t+1}, \mathbf{s}_{t+1}) - \hat{g}_t^{(i)}(\mathcal{B}_t^{(i)})\right\|_2^2$$

such that

$$\frac{1}{n}\sum_{i=1}^n \mathbf{E}\left[\mathbf{y}^{(i)}\left(g_i(\mathbf{u}_{t+1}, \mathbf{s}_{t+1}) - \hat{g}_t^{(i)}(\mathcal{B}_t^{(i)})\right)\right] \leq \frac{\lambda_2}{n}\mathbf{E}\left\|\mathbf{y}^{(i)} - \tilde{\mathbf{y}}_t^{(i)}\right\|_2^2 - \frac{\lambda_2}{n}\mathbf{E}\left\|\mathbf{y}^{(i)} - \tilde{\mathbf{y}}_t^{(i)}\right\|_2^2 + \frac{\sigma_0^2}{2\lambda_2 B}, \quad (38)$$

where $\{\tilde{\mathbf{y}}_t\}$ is also a virtual sequence for Algorithm 3. For any $\lambda_3 > 0$, the second term can be bounded as:

$$\frac{1}{n}\sum_{i=1}^{n}\mathbf{E}\left[\left(\mathbf{y}_t^{(i)} - \bar{\mathbf{y}}_{t+1}^{(i)}\right)\left(g_i(\mathbf{u}_{t+1}, \mathbf{s}_{t+1}) - \hat{g}_t^{(i)}(\mathcal{B}_t^{(i)})\right)\right] \le \frac{\lambda_3\sigma_0^2}{2B} + \frac{\mathbf{E}\|\bar{\mathbf{y}}_{t+1} - \mathbf{y}_t\|_2^2}{4\lambda_3 n}. \tag{39}$$

Put (36), (37), (38), and (39) together,

$$\mathcal{D}_2 \le \frac{C_g^2}{\lambda_4}\mathbf{E}\|\mathbf{u}_{t+1} - \mathbf{u}_t\|_2^2 + \frac{SC_g^2}{\lambda_4 n^2}\mathbf{E}\|\bar{\mathbf{s}}_{t+1} - \mathbf{s}_t\|_2^2 + 4\lambda_4 C_f^2 + \frac{\lambda_2}{2n}\left(\mathbf{E}\|\mathbf{y} - \tilde{\mathbf{y}}_t\|_2^2 - \mathbf{E}\|\mathbf{y} - \tilde{\mathbf{y}}_{t+1}\|_2^2\right)$$
$$+ \frac{\sigma_0^2}{2B\lambda_2} + \frac{\lambda_3\sigma_0^2}{2B} + \frac{\mathbf{E}\|\bar{\mathbf{y}}_{t+1} - \mathbf{y}_t\|_2^2}{4\lambda_3 n}. \tag{40}$$

For $\mathcal{D}_3$, under Assumption 4.1, for some $\lambda_5, \lambda_6 > 0$, we have

$$\mathcal{D}_3 \le C_f C_g \mathbf{E}\|\mathbf{u}_{t+1} - \mathbf{u}_t\|_2 + \frac{SC_f C_g}{n^2}\mathbf{E}\left[\sum_{i=1}^{n}\left|\bar{\mathbf{s}}_{t+1}^{(i)} - \mathbf{s}_t^{(i)}\right|\right]$$
$$\le \frac{\lambda_5}{2\eta}\mathbf{E}\|\mathbf{u}_{t+1} - \mathbf{u}_t\|_2^2 + \frac{\eta C_f^2 C_g^2}{2\lambda_5} + \frac{S\lambda_6}{2n^2\beta}\mathbf{E}\|\bar{\mathbf{s}}_{t+1} - \mathbf{s}_t\|_2^2 + \frac{S\beta C_f^2 C_g^2}{2n\lambda_6}. \tag{41}$$

For $\mathcal{D}_4$, similar to the derivations on $\mathcal{D}_3$, it holds that

$$\mathcal{D}_4 \le \frac{\lambda_5}{2\eta}\mathbf{E}\|\mathbf{u}_{t+1} - \mathbf{u}_t\|_2^2 + \frac{\eta C_f^2 C_g^2}{2\lambda_5} + \frac{\lambda_6}{2n\beta}\mathbf{E}\|\bar{\mathbf{s}}_{t+1} - \mathbf{s}_t\|_2^2 + \frac{\beta C_f^2 C_g^2}{2\lambda_6}. \tag{42}$$

Finally invoking Lemma 4 in Juditsky et al. (2011)(as well as Lemma 7 in Zhang & Lan (2020)) on $\mathcal{D}_5$ and $\mathcal{D}_6$, we have

$$\mathcal{D}_5 = -\mathbf{E}\left\langle\frac{1}{S}\sum_{i\in\mathcal{S}_t}\mathbf{y}_{t+1}^{(i)}\hat{G}_{t,1}^{(i)}(\tilde{\mathcal{B}}_t^{(i)}) - \frac{1}{n}\sum_{i=1}^{n}\bar{\mathbf{y}}_{t+1}^{(i)}G_{t,1}^{(i)}, \mathbf{u}_{t+1}\right\rangle \le \frac{\eta C_f^2\sigma_1^2}{B} + \frac{\eta\delta^2}{S}$$
$$\mathcal{D}_6 = -\mathbf{E}\left\langle\frac{1}{n}\sum_{i=1}^{n}\bar{\mathbf{y}}_{t+1}^{(i)}\hat{G}_{t,2}^{(i)}(\tilde{\mathcal{B}}_t^{(i)}) - \frac{1}{n}\sum_{i=1}^{n}\bar{\mathbf{y}}_{t+1}^{(i)}G_{t,2}^{(i)}, \mathbf{s}_{t+1}\right\rangle \le \frac{\beta C_f^2\sigma_2^2}{B}. \tag{43}$$

Supposing $\lambda_1 = 1 + \frac{1}{S}, \lambda_2 = \frac{n}{S\alpha}, \lambda_3 = \alpha, \lambda_4 = 8C_g^2\max\{\beta, \eta\}, \lambda_5 = \lambda_6 = \frac{1}{8}$ and substituting $\mathcal{D}_1, \mathcal{D}_2, \mathcal{D}_3, \mathcal{D}_4, \mathcal{D}_5, \mathcal{D}_6$ into equation (34) yields desired result. $\qquad\square$

### C.2.2 Proof of Theorem 4.3

*Proof.* Fix any $t \ge 0$. Applying Lemma C.4 with $(\mathbf{u}, \mathbf{s}) = (\mathbf{u}^*, \mathbf{s}^*)$, where $(\mathbf{u}^*, \mathbf{s}^*) \coloneqq \arg\min_{\mathbf{u}\in\mathcal{U}, \mathbf{s}\in\mathcal{S}} F(\mathbf{u}, \mathbf{s})$, and summing from $t = 0$ to $T - 1$, taking expectation on $\mathcal{F}_0$, we obtain

$$\sum_{t=0}^{T-1}\mathbf{E}\left[L(\mathbf{u}_{t+1}, \mathbf{s}_{t+1}, \mathbf{y}) - L(\mathbf{u}^*, \mathbf{s}^*, \bar{\mathbf{y}}_{t+1})\right]$$
$$\le \frac{1}{2\eta}\|\mathbf{u}^* - \mathbf{u}_0\|_2^2 + \frac{1}{2S\beta}\|\mathbf{s}^* - \mathbf{s}_0\|_2^2 + \frac{1}{\alpha S}\left(\|\mathbf{y} - \mathbf{y}_0\|_2^2 + \|\mathbf{y} - \hat{\mathbf{y}}_0\|_2^2 + \|\mathbf{y} - \tilde{\mathbf{y}}_0\|_2^2\right)$$
$$+ T\left(64\Omega C_g^2 C_f^2 + \frac{S\alpha\sigma_0^2}{2Bn} + \frac{\alpha\sigma_0^2}{2B} + \frac{\eta C_f^2\sigma_1^2}{B} + \frac{\eta\delta^2}{S} + \frac{\beta C_f^2\sigma_2^2}{B}\right). \tag{44}$$

Since $L(\mathbf{u}, \mathbf{s}, \mathbf{y})$ is convex on $\mathbf{u}, \mathbf{s}$ and linear on $\mathbf{y}$, we have

$$\max_{y}L(\bar{\mathbf{u}}, \bar{\mathbf{s}}, \mathbf{y}) - L(\mathbf{u}^*, \mathbf{s}^*, \bar{\bar{\mathbf{y}}}) \le \max_{\mathbf{y}}\frac{1}{T}\sum_{t=0}^{T-1}L(\mathbf{u}_{t+1}, \mathbf{s}_{t+1}, \mathbf{y}) - L(\mathbf{u}^*, \mathbf{s}^*, \bar{\mathbf{y}}_{t+1}), \tag{45}$$

where $\bar{\mathbf{u}} = \frac{1}{T}\sum_{t=0}^{T-1}\mathbf{u}_{t+1}, \bar{\mathbf{s}} = \frac{1}{T}\sum_{t=0}^{T-1}\mathbf{s}_{t+1}, \bar{\bar{\mathbf{y}}} = \frac{1}{T}\sum_{t=0}^{T-1}\bar{\mathbf{y}}_{t+1}$. Next, consider the left-hand side (LHS):

$$L(\bar{\mathbf{u}}, \bar{\mathbf{s}}, \mathbf{y}) - L(\mathbf{u}^*, \mathbf{s}^*, \bar{\bar{\mathbf{y}}}) = \frac{1}{n}\sum_{i=1}^{n}\left(\mathbf{y}^{(i)}g_i(\bar{\mathbf{u}}, \bar{\mathbf{s}}^{(i)}) - f_i^*(\mathbf{y}^{(i)})\right) - \frac{1}{n}\sum_{i=1}^{n}\left(\bar{\bar{\mathbf{y}}}^{(i)}g_i(\mathbf{u}^*, \mathbf{s}^{*(i)}) - f_i^*(\bar{\bar{\mathbf{y}}}^{(i)})\right). \tag{46}$$

Choose $\mathbf{y}^{(i)} = \tilde{\mathbf{y}}^{(i)} \in \arg\max_{\mathbf{v}}\{\mathbf{v}^{(i)}g_i(\bar{\mathbf{u}}, \bar{\mathbf{s}}^{(i)}) - f_i^*(\mathbf{v}^{(i)})\}$, By the definition of conjugate, we have $\mathbf{y}^{(i)}g_i(\bar{\mathbf{u}}, \bar{\mathbf{s}}^{(i)}) - f_i^*(\mathbf{y}^{(i)}) = f_i(g_i(\bar{\mathbf{u}}, \bar{\mathbf{s}}^{(i)}))$. By Fenchel-Young inequality, it holds that $\bar{\bar{\mathbf{y}}}^{(i)}g_i(\mathbf{u}^*, \mathbf{s}^{*(i)}) - f_i^*(\bar{\bar{\mathbf{y}}}^{(i)}) \leq f_i(g_i(\mathbf{u}^*, \mathbf{s}^{*(i)}))$. Combining the above $F(\bar{\mathbf{u}}, \bar{\mathbf{s}}) - F(\mathbf{u}^*, \mathbf{s}^*) \leq \max_{\mathbf{y}} \frac{1}{T}\sum_{t=0}^{T-1} L(\mathbf{u}_{t+1}, \mathbf{s}_{t+1}, \mathbf{y}) - L(\mathbf{u}^*, \mathbf{s}^*, \bar{\mathbf{y}}_{t+1})$, it follows that

$$\mathbf{E}F(\bar{\mathbf{u}}, \bar{\mathbf{s}}) - F(\mathbf{u}^*, \mathbf{s}^*) \leq \frac{1}{2\eta T}\|\mathbf{u}^* - \mathbf{u}_0\|_2^2 + \frac{1}{2S\beta T}\|\mathbf{s}^* - \mathbf{s}_0\|_2^2 + \frac{3D_{\mathcal{Y}}^2}{\alpha ST} + 64\Omega C_g^2 C_f^2$$
$$+ \frac{\alpha\sigma_0^2}{B} + \frac{\eta C_f^2\sigma_1^2}{B} + \frac{\eta\delta^2}{S} + \frac{\beta C_f^2\sigma_2^2}{B}. \tag{47}$$

Choose $\alpha \asymp \frac{B\epsilon}{\sigma_0^2}, \eta \asymp \min\{\frac{\epsilon}{C_g^2 C_f^2}, \frac{B\epsilon}{C_f^2\sigma_1^2}, \frac{S\epsilon}{\delta^2}\}, \beta \asymp \min\{\frac{\epsilon}{C_g^2 C_f^2}, \frac{B\epsilon}{C_f^2\sigma_2^2}\}$

and $T \asymp \max\{\frac{C_g^2 C_f^2}{\epsilon^2}, \frac{\mathcal{D}_{\mathcal{S}}^2 C_g^2 C_f^2}{S\epsilon^2}, \frac{C_f^2\sigma_1^2}{B\epsilon^2}, \frac{\mathcal{D}_{\mathcal{S}}^2 C_f^2\sigma_2^2}{BS\epsilon^2}, \frac{\delta^2}{S\epsilon^2}, \frac{D_{\mathcal{Y}}^2\sigma_0^2}{BS\epsilon^2}\}$ completes the proof. $\qquad\square$

## C.3 Non-convex Case

In this section, we present the proof of the non-convex case. The key to the analysis is to apply the convergence analysis of STACO1 for the regularized problem at each stage. However, there is a gap as STACO1 requires $g_i(\mathbf{w}, \mathbf{s}^{(i)})$ to be convex. To address this gap, we reformulate the problem in (8) as the following:

$$L_t(\mathbf{u}, \mathbf{s}, \mathbf{y}) = \frac{1}{n}\sum_{i=1}^{n}\mathbf{y}^{(i)}\left(g_i(\mathbf{u}, \mathbf{s}^{(i)}) + \frac{1}{2\tau^{(i)}}\|\mathbf{u} - \mathbf{u}_{t,0}\|_2^2 + \frac{1}{2\tau^{(i)}}\left(\mathbf{s}^{(i)} - \mathbf{s}_{t,0}^{(i)}\right)^2\right) - f_i^*(\mathbf{y}^{(i)})$$
$$+ \left(\frac{1}{2\gamma} - \frac{1}{2n}\sum_{i=1}^{n}\frac{\mathbf{y}^{(i)}}{\tau^{(i)}}\right)\|\mathbf{u} - \mathbf{u}_{t,0}\|_2^2 + \frac{1}{2n\gamma}\|\mathbf{s} - \mathbf{s}_{t,0}\|_2^2 - \frac{1}{2n}\sum_{i=1}^{n}\frac{\mathbf{y}^{(i)}}{\tau^{(i)}}\left(\mathbf{s}^{(i)} - \mathbf{s}_{t,0}^{(i)}\right)^2, \tag{48}$$

where $\tau^{(i)}$ is a proper constant. By carefully choosing the value of $\tau^{(i)}$, we can make $g_i(\mathbf{u}, \mathbf{s}^{(i)}) + \frac{1}{2\tau^{(i)}}\|\mathbf{u} - \mathbf{u}_{t,0}\|_2^2 + \frac{1}{2\tau^{(i)}}\left(\mathbf{s}^{(i)} - \mathbf{s}_{t,0}^{(i)}\right)^2$ to be convex in terms of $\mathbf{u}, \mathbf{s}^{(i)}$ such that we can leverage the convergence analysis of STACO1. Nevertheless, our algorithm does not depend on $\tau^{(i)}$ as computing the gradient of $\mathbf{u}$ and $\mathbf{s}^{(i)}$ will remove $\tau^{(i)}$. We now introduce some definitions and notations for our later analysis.

$$\Phi_\gamma(\mathbf{u}, \mathbf{s}; \mathbf{u}', \mathbf{s}') \coloneqq F(\mathbf{u}, \mathbf{s}) + \frac{1}{2\gamma}\|\mathbf{u} - \mathbf{u}'\|_2^2 + \frac{1}{2n\gamma}\|\mathbf{s} - \mathbf{s}'\|_2^2$$
$$\mathbf{u}_t = \mathbf{u}_{t,0}$$
$$\mathbf{s}_t = \mathbf{s}_{t,0}$$
$$(\mathbf{u}_t^\dagger, \mathbf{s}_t^\dagger) = \underset{\mathbf{u}\in\mathcal{U}, \mathbf{s}\in\mathcal{S}}{\arg\min}\left\{F(\mathbf{u}, \mathbf{s}) + \frac{1}{2\gamma}\|\mathbf{u} - \mathbf{u}_t\|_2^2 + \frac{1}{2n\gamma}\|\mathbf{s} - \mathbf{s}_t\|_2^2\right\}. \tag{49}$$

Since $f_i$ is convex and $g_i$ is non-convex, the function $F(\cdot, \cdot)$ is non-convex with respect to $\mathbf{u} \in \mathcal{U}$ and $\mathbf{s} \in \mathcal{S}$.

**Lemma C.5.** *Under Assumption 4.1 and 4.4, $F(\cdot, \cdot)$ is $C_f\rho$-weakly convex on $\mathbf{u} \in \mathcal{U}$ and $\frac{C_f\rho}{n}$-weakly convex on $\mathbf{s} \in \mathcal{S}$.*

*Proof.* See Appendix C.3.1. $\qquad\square$

Now we define the virtual sequence for inner loop update in STACO2.

**Definition C.6** (virtual sequence)**.** In Algorithm 4, for any $t$, a virtual sequence $\{\bar{\mathbf{y}}_{t,k}\}_k$ is defined as follows:

$$\bar{\mathbf{y}}_{t,k+1}^{(i)} = \underset{\mathbf{y}^{(i)}\in\mathcal{Y}_i}{\arg\max}\left\{\mathbf{y}^{(i)}\hat{g}_t^{(i)}(\mathcal{B}_{t,k}^{(i)}) - f_i^*(\mathbf{y}^{(i)}) - \frac{1}{2\alpha_t}\left(\mathbf{y}^{(i)} - \mathbf{y}_{t,k}^{(i)}\right)^2\right\} \quad i \in [n], \quad \forall k \geq 0$$
$$\bar{\mathbf{y}}_{t,0}^{(i)} = \mathbf{y}_{t,0}^{(i)} \quad i \in [n], \tag{50}$$

and a virtual sequence $\{\bar{\mathbf{s}}_{t,k}\}_k$ is defined as follows:

$$\bar{\mathbf{s}}_{t,k+1}^{(i)} = \underset{\mathbf{s}^{(i)}\in\mathcal{S}_i}{\arg\min}\left\{\left(\mathbf{y}_{t,k+1}^{(i)}\hat{G}_{t,k,2}^{(i)}(\tilde{\mathcal{B}}_{t,k}^{(i)}) + \frac{1}{\gamma}(\mathbf{s}_{t,k}^{(i)} - \mathbf{s}_{t,0}^{(i)})\right)\cdot\mathbf{s}^{(i)} + \frac{1}{2\beta_t}\left(\mathbf{s}^{(i)} - \mathbf{s}_{t,k}^{(i)}\right)^2\right\} \quad i \in [n], \quad \forall k \geq 0$$
$$\bar{\mathbf{s}}_{t,0}^{(i)} = \mathbf{s}_{t,0}^{(i)} \quad i \in [n]. \tag{51}$$

Lemma C.7 is similar to Lemma C.3, but this is for the inner loop in STACO2.

**Lemma C.7** (Lemma 9 in Wang & Yang). *Suppose $\{\bar{\mathbf{y}}_{t,k}\}_k, \{\hat{\mathbf{y}}_{t,k}\}_k$ are virtual sequences for any $t \geq 0$ in Algorithm 4. Then, for any $\lambda_1 > 0, \mathbf{y} \in \mathcal{Y}, t \in [0, T-1]$, the following holds:*

$$\mathbf{E}\left[\frac{1}{2n\alpha_t}\left(\|\mathbf{y} - \mathbf{y}_{t,k}\|_2^2 - \|\mathbf{y} - \bar{\mathbf{y}}_{t,k+1}\|_2^2 - \|\bar{\mathbf{y}}_{t,k+1} - \mathbf{y}_{t,k}\|_2^2\right)\right]$$

$$\leq \frac{1}{2\alpha_t S}\left(\|\mathbf{y} - \mathbf{y}_{t,k}\|_2^2 - \|\mathbf{y} - \mathbf{y}_{t,k+1}\|_2^2\right) + \frac{\lambda_1}{2\alpha_t S}\left(\|\mathbf{y} - \hat{\mathbf{y}}_{t,k}\|_2^2 - \|\mathbf{y} - \hat{\mathbf{y}}_{t,k+1}\|_2^2\right)$$

$$- \frac{1}{2\alpha_t n}(1 - \frac{1}{\lambda_1 S})\|\bar{\mathbf{y}}_{t,k+1} - \mathbf{y}_{t,k}\|_2^2. \tag{52}$$

There are two loops update in Algorithm 4. We first present the descent lemma of the inner loop. Its analysis is similar to that of Lemma C.4. However, since $g_i$ is not convex on $(\mathbf{u}, \mathbf{s})$, we cannot directly apply Lemma C.4. By carefully reformulating the regularized problem, we can leverage the convergence analysis from the convex case. For the inner loop, we define that $\mathcal{G}_{t,k}$ is the $\sigma$-algebra generated by $\{\mathcal{B}_{t,0}, \mathcal{S}_{t,0}, \cdots, \mathcal{B}_{t,k-1}, \mathcal{S}_{t-1}, \mathcal{B}_{t,k}\}$ and $\mathcal{F}_{t,k}$ is the $\sigma$-algebra generated by $\{\mathcal{B}_{t,0}, \mathcal{S}_{t,0}, \cdots, \mathcal{B}_{t,k-1}, \mathcal{S}_{t,k-1}, \mathcal{B}_{t,k}, \mathcal{S}_{t,k}\}$. Note that $\mathcal{G}_{t,k} \subset \mathcal{F}_{t,k}$ and $\mathbf{y}_{t,k+1}$ is $\mathcal{F}_{t,k}$-measurable.

**Lemma C.8** (Descent Lemma for Inner Loop). *Under Assumption 4.1,4.2 and 4.4, suppose that $\{\bar{\mathbf{y}}_{t,k}\}_k, \{\tilde{\mathbf{y}}_{t,k}\}_k, \{\hat{\mathbf{y}}_{t,k}\}_k, \{\bar{\mathbf{s}}_{t,k}\}_k$ are virtual sequences for Algorithm 4, and let $\gamma \leq \frac{1}{2C_f \rho}$ and $\eta_t, \beta_t \leq \frac{\gamma}{8}$. Then, for any $t \in [0, T-1]$ and $k \in [0, K_T - 2]$, the following holds:*

$$\mathbf{E}\left[L_\gamma(\mathbf{u}_{t,k+1}, \mathbf{s}_{t,k+1}, \mathbf{y}; \mathbf{u}_{t,0}, \mathbf{s}_{t,0}) - L_\gamma(\mathbf{u}, \mathbf{s}, \bar{\mathbf{y}}_{t,k+1}; \mathbf{u}_{t,0}, \mathbf{s}_{t,0})\right]$$

$$\leq \frac{1}{2\eta_t}\left(\|\mathbf{u} - \mathbf{u}_{t,k}\|_2^2 - \mathbf{E}\|\mathbf{u} - \mathbf{u}_{t,k+1}\|_2^2\right) + \frac{1}{2S\beta_t}\left(\|\mathbf{s} - \mathbf{s}_{t,k}\|_2^2 - \mathbf{E}\|\mathbf{s} - \mathbf{s}_{t,k+1}\|_2^2\right) + \frac{1}{\alpha_t S}\left(\|\mathbf{y} - \mathbf{y}_{t,k}\|_2^2 - \mathbf{E}\|\mathbf{y} - \mathbf{y}_{t,k+1}\|_2^2\right)$$

$$+ \frac{1}{\alpha_t S}\left(\|\mathbf{y} - \hat{\mathbf{y}}_{t,k}\|_2^2 - \mathbf{E}\|\mathbf{y} - \hat{\mathbf{y}}_{t,k+1}\|_2^2\right) + \frac{1}{\alpha_t S}\left(\|\mathbf{y} - \tilde{\mathbf{y}}_{t,k}\|_2^2 - \mathbf{E}\|\mathbf{y} - \tilde{\mathbf{y}}_{t,k+1}\|_2^2\right)$$

$$+ 64\Omega_t C_g^2 C_f^2 + \frac{S\alpha_t \sigma_0^2}{2Bn} + \frac{\alpha_t \sigma_0^2}{2B} + \frac{\eta_t C_f^2 \sigma_1^2}{B} + \frac{\eta_t \delta^2}{S} + \frac{\beta_t C_f^2 \sigma_2^2}{B}, \tag{53}$$

*where $\Omega_t = \max\{\eta_t, \beta_t\}$.*

*Proof.* See Appendix C.3.2. □

**Lemma C.9** (Proximal Error Bound). *Under Assumption 4.1,4.2 and 4.4, letting $\gamma \leq \frac{1}{2C_f \rho}$ and $\eta_t, \beta_t \leq \frac{\gamma}{8}$ for any $t$ in Algorithm 4, the following holds:*

$$\mathbf{E}\Phi_\gamma(\mathbf{u}_{t+1}, \mathbf{s}_{t+1}; \mathbf{u}_t, \mathbf{s}_t) \leq \Phi_\gamma(\mathbf{u}_t^\dagger, \mathbf{s}_t^\dagger; \mathbf{u}_t, \mathbf{s}_t) + \frac{1}{2\eta_t K_t}\left\|\mathbf{u}_t^\dagger - \mathbf{u}_t\right\|_2^2 + \frac{1}{2S\beta_t K_t}\left\|\mathbf{s}_t^\dagger - \mathbf{s}_t\right\|_2^2$$

$$+ \frac{3D_\mathcal{Y}^2}{\alpha_t S K_t} + 64\Omega_t C_g^2 C_f^2 + \frac{S\alpha_t \sigma_0^2}{2Bn} + \frac{\alpha_t \sigma_0^2}{2B} + \frac{\eta_t C_f^2 \sigma_1^2}{B} + \frac{\beta_t C_f^2 \sigma_2^2}{B} + \frac{\eta_t \delta^2}{S}, \tag{54}$$

*where $\Omega_t = \max\{\eta_t, \beta_t\}$.*

*Proof.* See Appendix C.3.3. □

### C.3.1 Proof of Lemma C.5

*Proof.* We first show $F(\mathbf{u}, \mathbf{s}) := \frac{1}{n}\sum_{i=1}^n f_i(g_i(\mathbf{u}, \mathbf{s}^{(i)}))$ is weakly convex on $\mathbf{u}$. For convenience, we denote $g_i(\cdot, \mathbf{s}^{(i)})$ as $g_i(\cdot)$. Then $\forall i \in [n]$ and $\mathbf{x}, \mathbf{y} \in \mathcal{U}$, we want to establish: $\forall \lambda \in [0, 1]$,

$$f_i(g_i(\lambda\mathbf{x} + (1-\lambda)\mathbf{y})) \leq \lambda f_i(g_i(\mathbf{x})) + (1-\lambda)f_i(g_i(\mathbf{y})) + \frac{\mu}{2}\lambda(1-\lambda)\|\mathbf{x} - \mathbf{y}\|_2^2, \tag{55}$$

for some $\mu > 0$. Since $g_i(\mathbf{x})$ is $\rho$-weakly convex, for any $\mathbf{x}, \mathbf{y} \in \mathcal{U}$ and $\lambda \in [0, 1]$, we have

$$g_i(\lambda \mathbf{x} + (1 - \lambda)\mathbf{y}) \leq \lambda g_i(\mathbf{x}) + (1 - \lambda)g_i(\mathbf{y}) + \frac{\rho}{2}\lambda(1 - \lambda)\|\mathbf{x} - \mathbf{y}\|_2^2. \tag{56}$$

Noticing $f_i(\cdot)$ is monotone non-decreasing, it holds that

$$f_i(g_i(\lambda \mathbf{x} + (1 - \lambda)\mathbf{y})) \leq f_i\left(\lambda g_i(\mathbf{x}) + (1 - \lambda)g_i(\mathbf{y}) + \frac{\rho}{2}\lambda(1 - \lambda)\|\mathbf{x} - \mathbf{y}\|_2^2\right). \tag{57}$$

By the $C_f$-Lipschitz continuity of $f_i$, we have:

$$f_i(a + \delta) \leq f_i(a) + C_f|\delta|, \tag{58}$$

where $a = \lambda g_i(\mathbf{x}) + (1 - \lambda)g_i(\mathbf{y})$ and $\delta = \frac{\rho}{2}\lambda(1 - \lambda)\|\mathbf{x} - \mathbf{y}\|_2^2$. Applying above inequality into (57), we can obtain:

$$\begin{aligned} f_i(g_i(\lambda \mathbf{x} + (1 - \lambda)\mathbf{y})) &\leq f_i(\lambda g_i(\mathbf{x}) + (1 - \lambda)g_i(\mathbf{y})) + C_f \cdot \frac{\rho}{2}\lambda(1 - \lambda)\|\mathbf{x} - \mathbf{y}\|_2^2 \\ &\leq \lambda f_i(g_i(\mathbf{x})) + (1 - \lambda)f_i(g_i(\mathbf{y})) + C_f \cdot \frac{\rho}{2}\lambda(1 - \lambda)\|\mathbf{x} - \mathbf{y}\|_2^2, \end{aligned} \tag{59}$$

where the last inequality holds due to the convexity of $f_i$. Therefore, $f_i(g_i(\mathbf{u}, \mathbf{s}))$ is $C_f\rho$-weakly convex on $\mathbf{u}$. Summing above inequality from $i = 1$ to $n$ and averaging, we obtain that $F(\mathbf{u}, \mathbf{s})$ is $C_f\rho$-weakly convex on $\mathbf{u}$. Next, we show that $F(\mathbf{u}, \mathbf{s})$ is weakly convex on $\mathbf{s}$. By denoting $g_i(\mathbf{u}, \cdot)$ as $g_i(\cdot)$ and following the similar approach, for any $i \in [n]$ and $x, y \in \mathcal{S}_i$, we have

$$f_i(g_i(\lambda x + (1 - \lambda)y)) \leq \lambda f_i(g_i(x)) + (1 - \lambda)f_i(g_i(y)) + C_f \cdot \frac{\rho}{2}\lambda(1 - \lambda)(x - y)^2. \tag{60}$$

Noticing $f_i(g_i(\mathbf{u}, \mathbf{s}))$ is weakly-convex to each coordinate of $\mathbf{s}$, $F(\mathbf{u}, \mathbf{s})$ is $\frac{C_f\rho}{n}$-weakly convex on $\mathbf{s}$. $\qquad \square$

### C.3.2 Proof of Lemma C.8

*Proof.* For analysis, we introduce two auxiliary variables $\tau_{t,k}$ and $\bar{\tau}_{t,k}$, where $\tau_{t,k}^{(i)} := \gamma \mathbf{y}_{t,k+1}^{(i)}$ for any $i \in \mathcal{S}_{t,k}$, $\bar{\tau}_{t,k}^{(i)} := \gamma \bar{\mathbf{y}}_{t,k+1}^{(i)}$ for any $i \in [n]$. Before delving into the formal proof of Lemma C.8, we briefly highlight its role: it provides a variation bound for stochastic gradients under block-coordinate updates, which is the central technical novelty enabling us to prove parallel mini-batch speedup. The proof proceeds in three steps: (i) decomposing the variation of block-coordinate updates, and (ii) bounding the dependence on different batches.

**Step 1: Decomposing the variation of block-coordinate updates.** By definition, we have

$$
L_\gamma(\mathbf{u}_{t,k+1}, \mathbf{s}_{t,k+1}, \mathbf{y}; \mathbf{u}_{t,0}, \mathbf{s}_{t,0}) - L_\gamma(\mathbf{u}, \mathbf{s}, \bar{\mathbf{y}}_{t,k+1}; \mathbf{u}_{t,0}, \mathbf{s}_{t,0})
$$

$$
= \frac{1}{n} \sum_{i=1}^{n} \left( \mathbf{y}^{(i)} g_i(\mathbf{u}_{t,k+1}, \mathbf{s}_{t,k+1}^{(i)}) - f_i^*(\mathbf{y}^{(i)}) \right) + \frac{1}{2\gamma} \|\mathbf{u}_{t,k+1} - \mathbf{u}_{t,0}\|_2^2 + \frac{1}{2n\gamma} \|\mathbf{s}_{t,k+1} - \mathbf{s}_{t,0}\|_2^2
$$

$$
- \frac{1}{n} \sum_{i=1}^{n} \left( \bar{\mathbf{y}}_{t,k+1}^{(i)} g_i(\mathbf{u}, \mathbf{s}) - f_i^*(\bar{\mathbf{y}}_{t,k+1}^{(i)}) \right) - \frac{1}{2\gamma} \|\mathbf{u} - \mathbf{u}_{t,0}\|_2^2 - \frac{1}{2n\gamma} \|\mathbf{s} - \mathbf{s}_{t,0}\|_2^2
$$

$$
= \frac{1}{n} \sum_{i=1}^{n} \left( \mathbf{y}^{(i)} \left( g_i(\mathbf{u}_{t,k+1}, \mathbf{s}_{t,k+1}^{(i)}) + \frac{1}{2\bar{\tau}_{t,k}^{(i)}} \|\mathbf{u}_{t,k+1} - \mathbf{u}_{t,0}\|_2^2 + \frac{1}{2\bar{\tau}_{t,k}^{(i)}} (\mathbf{s}_{t,k+1}^{(i)} - \mathbf{s}_{t,0}^{(i)})^2 \right) - f_i^*(\mathbf{y}^{(i)}) \right)
$$

$$
- \frac{1}{n} \sum_{i=1}^{n} \left( \bar{\mathbf{y}}_{t,k+1}^{(i)} \left( g_i(\mathbf{u}, \mathbf{s}) + \frac{1}{2\bar{\tau}_{t,k}^{(i)}} \|\mathbf{u} - \mathbf{u}_{t,0}\|_2^2 + \frac{1}{2\bar{\tau}_{t,k}^{(i)}} (\mathbf{s}^{(i)} - \mathbf{s}_{t,0}^{(i)})^2 \right) - f_i^*(\bar{\mathbf{y}}_{t,k+1}^{(i)}) \right)
$$

$$
+ \frac{1}{2\gamma} \|\mathbf{u}_{t,k+1} - \mathbf{u}_{t,0}\|_2^2 - \frac{1}{n} \sum_{i=1}^{n} \frac{\mathbf{y}^{(i)}}{2\bar{\tau}_{t,k}^{(i)}} \|\mathbf{u}_{t,k+1} - \mathbf{u}_{t,0}\|_2^2 - \frac{1}{2\gamma} \|\mathbf{u} - \mathbf{u}_{t,0}\|_2^2 + \frac{1}{n} \sum_{i=1}^{n} \frac{\bar{\mathbf{y}}_{t,k+1}^{(i)}}{2\bar{\tau}_{t,k}^{(i)}} \|\mathbf{u} - \mathbf{u}_{t,0}\|_2^2
$$

$$
+ \frac{1}{2n\gamma} \|\mathbf{s}_{t,k+1} - \mathbf{s}_{t,0}\|_2^2 - \frac{1}{n} \sum_{i=1}^{n} \frac{\mathbf{y}^{(i)}}{2\bar{\tau}_{t,k}^{(i)}} (\mathbf{s}_{t,k+1}^{(i)} - \mathbf{s}_{t,0}^{(i)})^2 - \frac{1}{2n\gamma} \|\mathbf{s} - \mathbf{s}_{t,0}\|_2^2 + \frac{1}{n} \sum_{i=1}^{n} \frac{\bar{\mathbf{y}}_{t,k+1}^{(i)}}{2\bar{\tau}_{t,k}^{(i)}} (\mathbf{s}^{(i)} - \mathbf{s}_{t,0}^{(i)})^2
$$

$$
= \frac{1}{n} \sum_{i=1}^{n} (\mathbf{y}^{(i)} - \bar{\mathbf{y}}_{t,k+1}^{(i)}) \left( g_i(\mathbf{u}_{t,k+1}, \mathbf{s}_{t,k+1}^{(i)}) + \frac{1}{2\bar{\tau}_{t,k}^{(i)}} \|\mathbf{u}_{t,k+1} - \mathbf{u}_{t,0}\|_2^2 + \frac{1}{2\bar{\tau}_{t,k}^{(i)}} (\mathbf{s}_{t,k+1}^{(i)} - \mathbf{s}_{t,0}^{(i)})^2 \right)
$$

$$
+ \frac{1}{n} \sum_{i=1}^{n} \bar{\mathbf{y}}_{t,k+1}^{(i)} \left( g_i(\mathbf{u}_{t,k+1}, \mathbf{s}_{t,k+1}^{(i)}) + \frac{1}{2\bar{\tau}_{t,k}^{(i)}} \|\mathbf{u}_{t,k+1} - \mathbf{u}_{t,0}\|_2^2 + \frac{1}{2\bar{\tau}_{t,k}^{(i)}} (\mathbf{s}_{t,k+1}^{(i)} - \mathbf{s}_{t,0}^{(i)})^2 \right.
$$

$$
\left. - g_i(\mathbf{u}_{t,k}, \mathbf{s}_{t,k}^{(i)}) - \frac{1}{2\bar{\tau}_{t,k}^{(i)}} \|\mathbf{u}_{t,k} - \mathbf{u}_{t,0}\|_2^2 - \frac{1}{2\bar{\tau}_{t,k}^{(i)}} (\mathbf{s}_{t,k}^{(i)} - \mathbf{s}_{t,0}^{(i)})^2 \right)
$$

$$
+ \frac{1}{n} \sum_{i=1}^{n} \bar{\mathbf{y}}_{t,k+1}^{(i)} \left( g_i(\mathbf{u}_{t,k}, \mathbf{s}_{t,k}^{(i)}) + \frac{1}{2\bar{\tau}_{t,k}^{(i)}} \|\mathbf{u}_{t,k} - \mathbf{u}_{t,0}\|_2^2 + \frac{1}{2\bar{\tau}_{t,k}^{(i)}} (\mathbf{s}_{t,k}^{(i)} - \mathbf{s}_{t,0}^{(i)})^2 \right.
$$

$$
\left. - g_i(\mathbf{u}, \mathbf{s}^{(i)}) - \frac{1}{2\bar{\tau}_{t,k}^{(i)}} \|\mathbf{u} - \mathbf{u}_{t,0}\|_2^2 - \frac{1}{2\bar{\tau}_{t,k}^{(i)}} (\mathbf{s}^{(i)} - \mathbf{s}_{t,0}^{(i)})^2 \right)
$$

$$
+ \frac{1}{2\gamma} \|\mathbf{u}_{t,k+1} - \mathbf{u}_{t,0}\|_2^2 - \frac{1}{n} \sum_{i=1}^{n} \frac{\mathbf{y}^{(i)}}{2\bar{\tau}_{t,k}^{(i)}} \|\mathbf{u}_{t,k+1} - \mathbf{u}_{t,0}\|_2^2 - \frac{1}{2\gamma} \|\mathbf{u} - \mathbf{u}_{t,0}\|_2^2 + \frac{1}{n} \sum_{i=1}^{n} \frac{\bar{\mathbf{y}}_{t,k+1}^{(i)}}{2\bar{\tau}_{t,k}^{(i)}} \|\mathbf{u} - \mathbf{u}_{t,0}\|_2^2
$$

$$
+ \frac{1}{2n\gamma} \|\mathbf{s}_{t,k+1} - \mathbf{s}_{t,0}\|_2^2 - \frac{1}{n} \sum_{i=1}^{n} \frac{\mathbf{y}^{(i)}}{2\bar{\tau}_{t,k}^{(i)}} (\mathbf{s}_{t,k+1}^{(i)} - \mathbf{s}_{t,0}^{(i)})^2 - \frac{1}{2n\gamma} \|\mathbf{s} - \mathbf{s}_{t,0}\|_2^2 + \frac{1}{n} \sum_{i=1}^{n} \frac{\bar{\mathbf{y}}_{t,k+1}^{(i)}}{2\bar{\tau}_{t,k}^{(i)}} (\mathbf{s}^{(i)} - \mathbf{s}_{t,0}^{(i)})^2
$$

$$
- \frac{1}{n} \sum_{i=1}^{n} f_i^*(\mathbf{y}^{(i)}) + \frac{1}{n} \sum_{i=1}^{n} f_i^*(\bar{\mathbf{y}}_{t,k+1}^{(i)}). \tag{61}
$$

Observe that for any $(\mathbf{u}, \mathbf{s}) \in (\mathcal{U}, \mathcal{S})$, the function $g_i(\mathbf{u}, \mathbf{s}^{(i)}) + \frac{1}{2\bar{\tau}_{t,k}^{(i)}} \|\mathbf{u} - \mathbf{u}'\|_2^2 + \frac{1}{2\bar{\tau}_{t,k}^{(i)}} (\mathbf{s}^{(i)} - \mathbf{s}'^{(i)})^2$ is convex with respect to $\mathbf{u}$ and $\mathbf{s}^{(i)}$, for any fixed $(\mathbf{u}', \mathbf{s}') \in (\mathcal{U}, \mathcal{S})$, since $\frac{1}{\bar{\tau}_{t,k}^{(i)}} = \frac{1}{\gamma \bar{\mathbf{y}}_{t,k+1}^{(i)}} \geq \rho$. This convexity enables us to

apply a first-order approximation and derive the following bound:

$$L_\gamma(\mathbf{u}_{t,k+1}, \mathbf{s}_{t,k+1}, \mathbf{y}; \mathbf{u}_{t,0}, \mathbf{s}_{t,0}) - L_\gamma(\mathbf{u}, \mathbf{s}, \bar{\mathbf{y}}_{t,k+1}; \mathbf{u}_{t,0}, \mathbf{s}_{t,0})$$

$$\leq \underbrace{\frac{1}{n}\sum_{i=1}^{n}(\mathbf{y}^{(i)} - \bar{\mathbf{y}}_{t,k+1}^{(i)})\left(g_i(\mathbf{u}_{t,k+1}, \mathbf{s}_{t,k+1}^{(i)}) + \frac{1}{2\bar{\tau}_{t,k}^{(i)}}\|\mathbf{u}_{t,k+1} - \mathbf{u}_{t,0}\|_2^2 + \frac{1}{2\bar{\tau}_{t,k}^{(i)}}(\mathbf{s}_{t,k+1}^{(i)} - \mathbf{s}_{t,0}^{(i)})^2\right)}_{\text{I}}$$

$$+\underbrace{\frac{1}{n}\sum_{i=1}^{n}\bar{\mathbf{y}}_{t,k+1}^{(i)}\left(g_i(\mathbf{u}_{t,k+1}, \mathbf{s}_{t,k+1}^{(i)}) + \frac{1}{2\bar{\tau}_{t,k}^{(i)}}\|\mathbf{u}_{t,k+1} - \mathbf{u}_{t,0}\|_2^2 + \frac{1}{2\bar{\tau}_{t,k}^{(i)}}(\mathbf{s}_{t,k+1}^{(i)} - \mathbf{s}_{t,0}^{(i)})^2\right.}_{\text{III}}$$

$$\underbrace{\left.- g_i(\mathbf{u}_{t,k}, \mathbf{s}_{t,k}^{(i)}) - \frac{1}{2\bar{\tau}_{t,k}^{(i)}}\|\mathbf{u}_{t,k} - \mathbf{u}_{t,0}\|_2^2 - \frac{1}{2\bar{\tau}_{t,k}^{(i)}}(\mathbf{s}_{t,k}^{(i)} - \mathbf{s}_{t,0}^{(i)})^2\right)}_{\text{III}}$$

$$+\underbrace{\frac{1}{n}\sum_{i=1}^{n}\bar{\mathbf{y}}_{t,k+1}^{(i)}\left[\left\langle G_{t,k,1}^{(i)} + \frac{1}{\bar{\tau}_{t,k}^{(i)}}(\mathbf{u}_{t,k} - \mathbf{u}_{t,0}), \mathbf{u}_{t,k} - \mathbf{u}\right\rangle + \left(G_{t,k,2}^{(i)} + \frac{1}{\bar{\tau}_{t,k}^{(i)}}(\mathbf{s}_{t,k}^{(i)} - \mathbf{s}_{t,0}^{(i)})\right)(\mathbf{s}_{t,k}^{(i)} - \mathbf{s}^{(i)})\right]}_{\text{II}}$$

$$\underbrace{-\frac{1}{n}\sum_{i=1}^{n}\frac{\mathbf{y}^{(i)}}{2\bar{\tau}_{t,k}^{(i)}}\|\mathbf{u}_{t,k+1} - \mathbf{u}_{t,0}\|_2^2}_{\text{I}} + \frac{1}{n}\sum_{i=1}^{n}\frac{\bar{\mathbf{y}}_{t,k+1}^{(i)}}{2\bar{\tau}_{t,k}^{(i)}}\|\mathbf{u} - \mathbf{u}_{t,0}\|_2^2 \underbrace{-\frac{1}{n}\sum_{i=1}^{n}\frac{\bar{\mathbf{y}}_{t,k+1}^{(i)}}{2\bar{\tau}_{t,k}^{(i)}}\|\mathbf{u}_{t,k+1} - \mathbf{u}_{t,0}\|_2^2}_{\text{III}}$$

$$\underbrace{+\frac{1}{n}\sum_{i=1}^{n}\frac{\bar{\mathbf{y}}_{t,k+1}^{(i)}}{2\bar{\tau}_{t,k}^{(i)}}\|\mathbf{u}_{t,k+1} - \mathbf{u}_{t,0}\|_2^2}_{\text{I}} - \frac{1}{n}\sum_{i=1}^{n}\frac{\bar{\mathbf{y}}_{t,k+1}^{(i)}}{2\bar{\tau}_{t,k}^{(i)}}\|\mathbf{u}_{t,k} - \mathbf{u}_{t,0}\|_2^2 \underbrace{+\frac{1}{n}\sum_{i=1}^{n}\frac{\bar{\mathbf{y}}_{t,k+1}^{(i)}}{2\bar{\tau}_{t,k}^{(i)}}\|\mathbf{u}_{t,k} - \mathbf{u}_{t,0}\|_2^2}_{\text{III}}$$

$$\underbrace{-\frac{1}{n}\sum_{i=1}^{n}\frac{\mathbf{y}^{(i)}}{2\bar{\tau}_{t,k}^{(i)}}(\mathbf{s}_{t,k+1}^{(i)} - \mathbf{s}_{t,0}^{(i)})^2}_{\text{I}} + \frac{1}{n}\sum_{i=1}^{n}\frac{\bar{\mathbf{y}}_{t,k+1}^{(i)}}{2\bar{\tau}_{t,k}^{(i)}}(\mathbf{s}^{(i)} - \mathbf{s}_{t,0}^{(i)})^2 \underbrace{-\frac{1}{n}\sum_{i=1}^{n}\frac{\bar{\mathbf{y}}_{t,k+1}^{(i)}}{2\bar{\tau}_{t,k}^{(i)}}(\mathbf{s}_{t,k+1}^{(i)} - \mathbf{s}_{t,0}^{(i)})^2}_{\text{III}}$$

$$\underbrace{+\frac{1}{n}\sum_{i=1}^{n}\frac{\bar{\mathbf{y}}_{t,k+1}^{(i)}}{2\bar{\tau}_{t,k}^{(i)}}(\mathbf{s}_{t,k+1}^{(i)} - \mathbf{s}_{t,0}^{(i)})^2}_{\text{I}} - \frac{1}{n}\sum_{i=1}^{n}\frac{\bar{\mathbf{y}}_{t,k+1}^{(i)}}{2\bar{\tau}_{t,k}^{(i)}}(\mathbf{s}_{t,k}^{(i)} - \mathbf{s}_{t,0}^{(i)})^2 \underbrace{+\frac{1}{n}\sum_{i=1}^{n}\frac{\bar{\mathbf{y}}_{t,k+1}^{(i)}}{2\bar{\tau}_{t,k}^{(i)}}(\mathbf{s}_{t,k}^{(i)} - \mathbf{s}_{t,0}^{(i)})^2}_{\text{III}}$$

$$+\frac{1}{2\gamma}\|\mathbf{u}_{t,k+1} - \mathbf{u}_{t,0}\|_2^2 - \frac{1}{2\gamma}\|\mathbf{u} - \mathbf{u}_{t,0}\|_2^2 + \frac{1}{2n\gamma}\|\mathbf{s}_{t,k+1} - \mathbf{s}_{t,0}\|_2^2 - \frac{1}{2n\gamma}\|\mathbf{s} - \mathbf{s}_{t,0}\|_2^2$$

$$-\frac{1}{n}\sum_{i=1}^{n}f_i^*(\mathbf{y}^{(i)}) + \frac{1}{n}\sum_{i=1}^{n}f_i^*(\bar{\mathbf{y}}_{t,k+1}^{(i)}). \tag{62}$$

Noticing

$$
\begin{aligned}
\text{I} &= \frac{1}{n}\sum_{i=1}^{n}(\mathbf{y}^{(i)} - \bar{\mathbf{y}}_{t,k+1}^{(i)})g_i(\mathbf{u}_{t,k+1}, \mathbf{s}_{t,k+1}^{(i)}) \\
&= \frac{1}{n}\sum_{i=1}^{n}(\mathbf{y}^{(i)} - \bar{\mathbf{y}}_{t,k+1}^{(i)})\hat{g}_{t,k}^{(i)}(\mathcal{B}_{t,k}^{(i)}) + \frac{1}{n}\sum_{i=1}^{n}(\mathbf{y}^{(i)} - \bar{\mathbf{y}}_{t,k+1}^{(i)})\left(g_i(\mathbf{u}_{t,k+1}, \mathbf{s}_{t,k+1}^{(i)}) - \hat{g}_{t,k}^{(i)}(\mathcal{B}_{t,k}^{(i)})\right)
\end{aligned}
$$

$$
\begin{aligned}
\text{II} &= \left\langle \frac{1}{S}\sum_{i\in\mathcal{S}_{t,k}}\mathbf{y}_{t,k+1}^{(i)}\left(\hat{G}_{t,k,1}^{(i)}(\tilde{\mathcal{B}}_{t,k}^{(i)}) + \frac{1}{\tau_{t,k}^{(i)}}(\mathbf{u}_{t,k} - \mathbf{u}_{t,0})\right) - \frac{1}{n}\sum_{i=1}^{n}\bar{\mathbf{y}}_{t,k+1}^{(i)}\left(G_{t,k,1}^{(i)} + \frac{1}{\bar{\tau}_{t,k}^{(i)}}(\mathbf{u}_{t,k} - \mathbf{u}_{t,0})\right), \mathbf{u} - \mathbf{u}_{t,k+1}\right\rangle \\
&\quad - \frac{1}{S}\sum_{i\in\mathcal{S}_{t,k}}\left\langle \mathbf{y}_{t,k+1}^{(i)}\left(\hat{G}_{t,k,1}^{(i)}(\tilde{\mathcal{B}}_{t,k}^{(i)}) + \frac{1}{\tau_{t,k}^{(i)}}(\mathbf{u}_{t,k} - \mathbf{u}_{t,0})\right), \mathbf{u} - \mathbf{u}_{t,k+1}\right\rangle \\
&\quad + \frac{1}{n}\sum_{i=1}^{n}\left\langle \bar{\mathbf{y}}_{t,k+1}^{(i)}\left(G_{t,k,1}^{(i)} + \frac{1}{\bar{\tau}_{t,k}^{(i)}}(\mathbf{u}_{t,k} - \mathbf{u}_{t,0})\right), \mathbf{u}_{t,k} - \mathbf{u}_{t,k+1}\right\rangle \\
&\quad + \frac{1}{n}\sum_{i=1}^{n}\bar{\mathbf{y}}_{t,k+1}^{(i)}(\hat{G}_{t,k,2}^{(i)}(\tilde{\mathcal{B}}_{t,k}^{(i)}) - G_{t,k,2}^{(i)})(\mathbf{s}^{(i)} - \bar{\mathbf{s}}_{t,k+1}^{(i)}) \\
&\quad - \frac{1}{n}\sum_{i=1}^{n}\bar{\mathbf{y}}_{t,k+1}^{(i)}\left(\hat{G}_{t,k,2}^{(i)}(\tilde{\mathcal{B}}_{t,k}^{(i)}) + \frac{1}{\bar{\tau}_{t,k}^{(i)}}(\mathbf{s}_{t,k}^{(i)} - \mathbf{s}_{t,0}^{(i)})\right)(\mathbf{s}^{(i)} - \bar{\mathbf{s}}_{t,k+1}^{(i)}) \\
&\quad + \frac{1}{n}\sum_{i=1}^{n}\bar{\mathbf{y}}_{t,k+1}^{(i)}\left(G_{t,k,2}^{(i)} + \frac{1}{\bar{\tau}_{t,k}^{(i)}}(\mathbf{s}_{t,k}^{(i)} - \mathbf{s}_{t,0}^{(i)})\right)(\mathbf{s}_{t,k}^{(i)} - \bar{\mathbf{s}}_{t,k+1}^{(i)}) \\
\text{III} &= \frac{1}{n}\sum_{i=1}^{n}\bar{\mathbf{y}}_{t,k+1}^{(i)}\left(g_i(\mathbf{u}_{t,k+1}, \mathbf{s}_{t,k+1}^{(i)}) - g_i(\mathbf{u}_{t,k}, \mathbf{s}_{t,k}^{(i)})\right),
\end{aligned}
\tag{63}
$$

and replacing $\tau_{t,k}^{(i)} = \gamma\mathbf{y}_{t,k+1}^{(i)}$ and $\bar{\tau}_{t,k}^{(i)} = \gamma\bar{\mathbf{y}}_{t,k+1}^{(i)}$, it follows that

$$
\begin{aligned}
&L_\gamma(\mathbf{u}_{t,k+1}, \mathbf{s}_{t,k+1}, \mathbf{y}; \mathbf{u}_{t,0}, \mathbf{s}_{t,0}) - L_\gamma(\mathbf{u}, \mathbf{s}, \bar{\mathbf{y}}_{t,k+1}; \mathbf{u}_{t,0}, \mathbf{s}_{t,0}) \\
&\leq \underbrace{\frac{1}{n}\sum_{i=1}^{n}(\mathbf{y}^{(i)} - \bar{\mathbf{y}}_{t,k+1}^{(i)})\hat{g}_{t,k}^{(i)}(\mathcal{B}_{t,k}^{(i)}) - \frac{1}{n}\sum_{i=1}^{n}f_i^*(\mathbf{y}^{(i)}) + \frac{1}{n}\sum_{i=1}^{n}f_i^*(\bar{\mathbf{y}}_{t,k+1}^{(i)})}_{\mathcal{C}_1} \\
&\quad + \frac{1}{n}\sum_{i=1}^{n}(\mathbf{y}^{(i)} - \bar{\mathbf{y}}_{t,k+1}^{(i)})\left(g_i(\mathbf{u}_{t,k+1}, \mathbf{s}_{t,k+1}^{(i)}) - \hat{g}_{t,k}^{(i)}(\mathcal{B}_{t,k}^{(i)})\right) + \frac{1}{n}\sum_{i=1}^{n}\bar{\mathbf{y}}_{t,k+1}^{(i)}\left(g_i(\mathbf{u}_{t,k+1}, \mathbf{s}_{t,k+1}^{(i)}) - g_i(\mathbf{u}_{t,k}, \mathbf{s}_{t,k}^{(i)})\right) \\
&\quad + \underbrace{\left\langle \frac{1}{S}\sum_{i\in\mathcal{S}_{t,k}}\left(\mathbf{y}_{t,k+1}^{(i)}\hat{G}_{t,k,1}^{(i)}(\tilde{\mathcal{B}}_{t,k}^{(i)}) + \frac{1}{\gamma}(\mathbf{u}_{t,k} - \mathbf{u}_{t,0})\right) - \frac{1}{n}\sum_{i=1}^{n}\left(\bar{\mathbf{y}}_{t,k+1}^{(i)}G_{t,k,1}^{(i)} + \frac{1}{\gamma}(\mathbf{u}_{t,k} - \mathbf{u}_{t,0})\right), \mathbf{u} - \mathbf{u}_{t,k+1}\right\rangle}_{} \\
&\quad \underbrace{- \frac{1}{S}\sum_{i\in\mathcal{S}_{t,k}}\left\langle \mathbf{y}_{t,k+1}^{(i)}\hat{G}_{t,k,1}^{(i)}(\tilde{\mathcal{B}}_{t,k}^{(i)}) + \frac{1}{\gamma}(\mathbf{u}_{t,k} - \mathbf{u}_{t,0}), \mathbf{u} - \mathbf{u}_{t,k+1}\right\rangle}_{\mathcal{C}_2} \\
&\quad + \underbrace{\frac{1}{n}\sum_{i=1}^{n}\bar{\mathbf{y}}_{t,k+1}^{(i)}(\hat{G}_{t,k,2}^{(i)}(\tilde{\mathcal{B}}_{t,k}^{(i)}) - G_{t,k,2}^{(i)})(\mathbf{s}^{(i)} - \bar{\mathbf{s}}_{t,k+1}^{(i)})}_{} \\
&\quad \underbrace{- \frac{1}{n}\sum_{i=1}^{n}\left(\bar{\mathbf{y}}_{t,k+1}^{(i)}\hat{G}_{t,k,2}^{(i)}(\tilde{\mathcal{B}}_{t,k}^{(i)}) + \frac{1}{\gamma}(\mathbf{s}_{t,k}^{(i)} - \mathbf{s}_{t,0}^{(i)})\right)(\mathbf{s}^{(i)} - \bar{\mathbf{s}}_{t,k+1}^{(i)})}_{\mathcal{C}_3} \\
&\quad + \frac{1}{n}\sum_{i=1}^{n}\left\langle \bar{\mathbf{y}}_{t,k+1}^{(i)}G_{t,k,1}^{(i)} + \frac{1}{\gamma}(\mathbf{u}_{t,k} - \mathbf{u}_{t,0}), \mathbf{u}_{t,k} - \mathbf{u}_{t,k+1}\right\rangle + \frac{1}{n}\sum_{i=1}^{n}\left(\bar{\mathbf{y}}_{t,k+1}^{(i)}G_{t,k,2}^{(i)} + \frac{1}{\gamma}(\mathbf{s}_{t,k}^{(i)} - \mathbf{s}_{t,0}^{(i)})\right)(\mathbf{s}_{t,k}^{(i)} - \bar{\mathbf{s}}_{t,k+1}^{(i)}) \\
&\quad + \frac{1}{2n\gamma}\sum_{i=1}^{n}\|\mathbf{u}_{t,k+1} - \mathbf{u}_{t,0}\|_2^2 - \frac{1}{2n\gamma}\sum_{i=1}^{n}\|\mathbf{u}_{t,k} - \mathbf{u}_{t,0}\|_2^2 + \frac{1}{2n\gamma}\sum_{i=1}^{n}(\bar{\mathbf{s}}_{t,k+1}^{(i)} - \mathbf{s}_{t,0}^{(i)})^2 - \frac{1}{2n\gamma}\sum_{i=1}^{n}(\mathbf{s}_{t,k}^{(i)} - \mathbf{s}_{t,0}^{(i)})^2.
\end{aligned}
\tag{64}
$$

For $\mathcal{C}_1$, by applying Lemma C.1 followed by Lemma C.7, we obtain:

$$
\begin{aligned}
\mathcal{C}_1 \underset{\text{Lemma C.1}}{\leq} & \left[ \frac{1}{2n\alpha_t} \left( \|\mathbf{y} - \mathbf{y}_{t,k}\|_2^2 - \|\mathbf{y} - \bar{\mathbf{y}}_{t,k+1}\|_2^2 - \|\bar{\mathbf{y}}_{t,k+1} - \mathbf{y}_{t,k}\|_2^2 \right) \right] \\
\underset{\text{Lemma C.7}}{\leq} & \frac{1}{2\alpha_t S} \left( \|\mathbf{y} - \mathbf{y}_{t,k}\|_2^2 - \|\mathbf{y} - \mathbf{y}_{t,k+1}\|_2^2 \right) + \frac{\lambda_1}{2\alpha_t S} \left( \|\mathbf{y} - \hat{\mathbf{y}}_{t,k}\|_2^2 - \|\mathbf{y} - \hat{\mathbf{y}}_{t,k+1}\|_2^2 \right) \\
& - \frac{1}{2\alpha_t n}(1 - \frac{1}{\lambda_1 S}) \|\bar{\mathbf{y}}_{t,k+1} - \mathbf{y}_{t,k}\|_2^2.
\end{aligned}
\tag{65}
$$

For $\mathcal{C}_2$, we define the auxiliary function:

$$
\phi(\mathbf{u}) := \left\langle \frac{1}{S} \sum_{i \in \mathcal{S}_{t,k}} \mathbf{y}_{t,k+1}^{(i)} \left( \hat{G}_{t,k,1}^{(i)}(\tilde{\mathcal{B}}_{t,k}^{(i)}) + \frac{1}{\tau_{t,k}^{(i)}}(\mathbf{u}_{t,k} - \mathbf{u}_{t,0}) \right), \mathbf{u} \right\rangle.
\tag{66}
$$

Substituting $\tau_{t,k}^{(i)} = \gamma \mathbf{y}_{t,k+1}^{(i)}$, this becomes

$$
\phi(\mathbf{u}) = \left\langle \frac{1}{S} \sum_{i \in \mathcal{S}_{t,k}} \mathbf{y}_{t,k+1}^{(i)} \hat{G}_{t,k,1}^{(i)}(\tilde{\mathcal{B}}_{t,k}^{(i)}) + \frac{1}{\gamma}(\mathbf{u}_{t,k} - \mathbf{u}_{t,0}), \mathbf{u} \right\rangle.
\tag{67}
$$

Since $\phi(\cdot)$ is convex, applying Lemma C.1 yields:

$$
\mathcal{C}_2 \leq \frac{1}{2\eta_t} \left( \|\mathbf{u} - \mathbf{u}_{t,k}\|_2^2 - \|\mathbf{u} - \mathbf{u}_{t,k+1}\|_2^2 \right) - \frac{1}{2\eta_t} \|\mathbf{u}_{t,k+1} - \mathbf{u}_{t,k}\|_2^2.
\tag{68}
$$

For $\mathcal{C}_3$, following the similar manner with $\mathcal{C}_2$, for any $i$ in $\mathcal{S}_{t,k}$, we can get

$$
\mathcal{C}_3 \underset{\text{Lemma C.1}}{\leq} \frac{1}{2n\beta_t} \left( \|\mathbf{s} - \mathbf{s}_{t,k}\|_2^2 - \|\mathbf{s} - \bar{\mathbf{s}}_{t,k+1}\|_2^2 \right) - \frac{1}{2n\beta_t} \|\bar{\mathbf{s}}_{t,k+1} - \mathbf{s}_{t,k}\|_2^2.
\tag{69}
$$

Substituting the above inequalities into (64) and taking expectation with respect to $\mathcal{F}_{t,k}$ yields:

$$
\mathbf{E}\left[L_\gamma(\mathbf{u}_{t,k+1}, \mathbf{s}_{t,k+1}, \mathbf{y}; \mathbf{u}_{t,0}, \mathbf{s}_{t,0}) - L_\gamma(\mathbf{u}, \mathbf{s}, \bar{\mathbf{y}}_{t,k+1}; \mathbf{u}_{t,0}, \mathbf{s}_{t,0})\right]
$$

$$
\leq \frac{1}{2\alpha_t S}\left(\|\mathbf{y} - \mathbf{y}_{t,k}\|_2^2 - \mathbf{E}\|\mathbf{y} - \mathbf{y}_{t,k+1}\|_2^2\right) + \frac{\lambda_1}{2\alpha_t S}\left(\|\mathbf{y} - \hat{\mathbf{y}}_{t,k}\|_2^2 - \mathbf{E}\|\mathbf{y} - \hat{\mathbf{y}}_{t,k+1}\|_2^2\right)
$$

$$
- \frac{1}{2\alpha_t n}(1 - \frac{1}{\lambda_1 S})\mathbf{E}\|\bar{\mathbf{y}}_{t,k+1} - \mathbf{y}_{t,k}\|_2^2
$$

$$
+ \frac{1}{2\eta_t}\left(\|\mathbf{u} - \mathbf{u}_{t,k}\|_2^2 - \mathbf{E}\|\mathbf{u} - \mathbf{u}_{t,k+1}\|_2^2\right) - \frac{1}{2\eta_t}\mathbf{E}\|\mathbf{u}_{t,k+1} - \mathbf{u}_{t,k}\|_2^2
$$

$$
+ \underbrace{\frac{1}{2n\beta_t}\left(\|\mathbf{s} - \mathbf{s}_{t,k}\|_2^2 - \mathbf{E}\|\mathbf{s} - \bar{\mathbf{s}}_{t,k+1}\|_2^2\right) - \frac{1}{2n\beta_t}\mathbf{E}\|\bar{\mathbf{s}}_{t,k+1} - \mathbf{s}_{t,k}\|_2^2}_{\mathcal{D}_1}
$$

$$
+ \underbrace{\frac{1}{n}\sum_{i=1}^n \mathbf{E}\left[(\mathbf{y}^{(i)} - \bar{\mathbf{y}}_{t,k+1}^{(i)})\left(g_i(\mathbf{u}_{t,k+1}, \mathbf{s}_{t,k+1}^{(i)}) - \hat{g}_{t,k}^{(i)}(\mathcal{B}_{t,k}^{(i)})\right)\right]}_{\mathcal{D}_2}
$$

$$
+ \underbrace{\frac{1}{n}\sum_{i=1}^n \mathbf{E}\left[\bar{\mathbf{y}}_{t,k+1}^{(i)}\left(g_i(\mathbf{u}_{t,k+1}, \mathbf{s}_{t,k+1}^{(i)}) - g_i(\mathbf{u}_{t,k}, \mathbf{s}_{t,k}^{(i)})\right)\right]}_{\mathcal{D}_3}
$$

$$
+ \underbrace{\frac{1}{n}\sum_{i=1}^n \mathbf{E}\left\langle\bar{\mathbf{y}}_{t,k+1}^{(i)} G_{t,k,1}^{(i)}, \mathbf{u}_{t,k} - \mathbf{u}_{t,k+1}\right\rangle + \frac{1}{n}\sum_{i=1}^n \mathbf{E}\left\langle\bar{\mathbf{y}}_{t,k+1}^{(i)} G_{t,k,2}^{(i)}, \mathbf{s}_{t,k}^{(i)} - \bar{\mathbf{s}}_{t,k+1}^{(i)}\right\rangle}_{\mathcal{D}_4}
$$

$$
+ \underbrace{\mathbf{E}\left[\frac{1}{\gamma}\langle\mathbf{u}_{t,k} - \mathbf{u}_{t,0}, \mathbf{u}_{t,k} - \mathbf{u}_{t,k+1}\rangle\right] + \mathbf{E}\left[\frac{1}{2\gamma}\left(\|\mathbf{u}_{t,k+1} - \mathbf{u}_{t,0}\|_2^2 - \|\mathbf{u}_{t,k} - \mathbf{u}_{t,0}\|_2^2\right)\right]}_{\mathcal{D}_5}
$$

$$
+ \underbrace{\mathbf{E}\left[\frac{1}{n\gamma}\langle\mathbf{s}_{t,k} - \mathbf{s}_{t,0}, \mathbf{s}_{t,k} - \bar{\mathbf{s}}_{t,k+1}\rangle\right] + \mathbf{E}\left[\frac{1}{2n\gamma}\left(\|\bar{\mathbf{s}}_{t,k+1} - \mathbf{s}_{t,0}\|_2^2 - \|\mathbf{s}_{t,k} - \mathbf{s}_{t,0}\|_2^2\right)\right]}_{\mathcal{D}_6}
$$

$$
+ \underbrace{\mathbf{E}\left\langle\frac{1}{S}\sum_{i \in \mathcal{S}_{t,k}} \mathbf{y}_{t,k+1}^{(i)} \hat{G}_{t,k,1}^{(i)}(\tilde{\mathcal{B}}_{t,k}^{(i)}) - \frac{1}{n}\sum_{i=1}^n \bar{\mathbf{y}}_{t,k+1}^{(i)} G_{t,k,1}^{(i)}, \mathbf{u} - \mathbf{u}_{t,k+1}\right\rangle}_{\mathcal{D}_7}
$$

$$
+ \underbrace{\frac{1}{n}\sum_{i=1}^n \mathbf{E}\left[\bar{\mathbf{y}}_{t,k+1}^{(i)}\left(\hat{G}_{t,k,2}^{(i)}(\tilde{\mathcal{B}}_{t,k}^{(i)}) - G_{t,k,2}^{(i)}\right)(\mathbf{s}^{(i)} - \bar{\mathbf{s}}_{t,k+1}^{(i)})\right]}_{\mathcal{D}_8}. \tag{70}
$$

**Step 2: Bounding the dependence on different batches.** For $\mathcal{D}_1$, notice that $\mathbf{E}\left[(\mathbf{s}^{(i)} - \bar{\mathbf{s}}_{t,k+1}^{(i)})^2\right] = \frac{S}{n}(\mathbf{s}^{(i)} - \bar{\mathbf{s}}_{t,k+1}^{(i)})^2 + \frac{n-S}{n}(\mathbf{s}^{(i)} - \mathbf{s}_{t,k}^{(i)})^2$ for any $i \in [n]$. Then, it holds that

$$
\mathcal{D}_1 \leq \frac{1}{2S\beta_t}\left(\|\mathbf{s} - \mathbf{s}_{t,k}\|_2^2 - \mathbf{E}\|\mathbf{s} - \mathbf{s}_{t,k+1}\|_2^2\right) - \frac{1}{2n\beta_t}\mathbf{E}\|\bar{\mathbf{s}}_{t,k+1} - \mathbf{s}_{t,k}\|_2^2. \tag{71}
$$

For $\mathcal{D}_2$, following a similar manner we show in (40), for some $\lambda_2, \lambda_3, \lambda_4 > 0$, we have

$$
\mathcal{D}_2 \leq \frac{C_g^2}{\lambda_4}\mathbf{E}\|\mathbf{u}_{t,k+1} - \mathbf{u}_{t,k}\|_2^2 + \frac{SC_g^2}{\lambda_4 n^2}\mathbf{E}\|\bar{\mathbf{s}}_{t,k+1} - \mathbf{s}_{t,k}\|_2^2 + 4\lambda_4 C_f^2 + \frac{\lambda_2}{2n}\left(\mathbf{E}\|\mathbf{y} - \tilde{\mathbf{y}}_{t,k}\|_2^2 - \mathbf{E}\|\mathbf{y} - \tilde{\mathbf{y}}_{t,k+1}\|_2^2\right)
$$

$$
+ \frac{\sigma_0^2}{2B\lambda_2} + \frac{\lambda_3\sigma_0^2}{2B} + \frac{\mathbf{E}\|\bar{\mathbf{y}}_{t,k+1} - \mathbf{y}_{t,k}\|_2^2}{4\lambda_3 n}. \tag{72}
$$

For $\mathcal{D}_3$, by invoking Assumption 4.1, for some $\lambda_5, \lambda_6 > 0$, we have

$$\mathcal{D}_3 \le C_f C_g \mathbf{E} \left\| \mathbf{u}_{t,k+1} - \mathbf{u}_{t,k} \right\|_2 + \frac{SC_f C_g}{n^2} \mathbf{E} \left[ \sum_{i=1}^{n} \left| \bar{\mathbf{s}}_{t,k+1}^{(i)} - \mathbf{s}_{t,k}^{(i)} \right| \right]$$

$$\le \frac{\lambda_5}{2\eta_t} \mathbf{E} \left\| \mathbf{u}_{t,k+1} - \mathbf{u}_{t,k} \right\|_2^2 + \frac{\eta_t C_f^2 C_g^2}{2\lambda_5} + \frac{S\lambda_6}{2n^2 \beta_t} \mathbf{E} \left\| \bar{\mathbf{s}}_{t,k+1} - \mathbf{s}_{t,k} \right\|_2^2 + \frac{S\beta_t C_f^2 C_g^2}{2n\lambda_6}. \tag{73}$$

For $\mathcal{D}_4$, same to the derivations for $\mathcal{D}_3$, it holds that

$$\mathcal{D}_4 \le \frac{\lambda_5}{2\eta_t} \mathbf{E} \left\| \mathbf{u}_{t,k+1} - \mathbf{u}_{t,k} \right\|_2^2 + \frac{\eta_t C_f^2 C_g^2}{2\lambda_5} + \frac{\lambda_6}{2n\beta_t} \mathbf{E} \left\| \bar{\mathbf{s}}_{t,k+1} - \mathbf{s}_{t,k} \right\|_2^2 + \frac{\beta_t C_f^2 C_g^2}{2\lambda_6}. \tag{74}$$

For $\mathcal{D}_5$, noticing

$$\frac{1}{2} \left( \left\| \mathbf{u}_{t,k+1} - \mathbf{u}_{t,0} \right\|_2^2 - \left\| \mathbf{u}_{t,k} - \mathbf{u}_{t,0} \right\|_2^2 \right) \le - \left\langle \mathbf{u}_{t,k+1} - \mathbf{u}_{t,0}, \, \mathbf{u}_{t,k} - \mathbf{u}_{t,k+1} \right\rangle, \tag{75}$$

then we have

$$\mathcal{D}_5 \le \mathbf{E} \left[ \frac{1}{\gamma} \left\langle \mathbf{u}_{t,k} - \mathbf{u}_{t,0}, \, \mathbf{u}_{t,k} - \mathbf{u}_{t,k+1} \right\rangle \right] - \mathbf{E} \left[ \frac{1}{\gamma} \left\langle \mathbf{u}_{t,k+1} - \mathbf{u}_{t,0}, \, \mathbf{u}_{t,k} - \mathbf{u}_{t,k+1} \right\rangle \right]$$

$$= \frac{1}{\gamma} \mathbf{E} \left\| \mathbf{u}_{t,k} - \mathbf{u}_{t,k+1} \right\|_2^2. \tag{76}$$

In the same manner as for $\mathcal{D}_5$, we obtain $\mathcal{D}_6 \le \frac{1}{n\gamma} \mathbf{E} \left\| \mathbf{s}_{t,k} - \bar{\mathbf{s}}_{t,k+1} \right\|_2^2$. Next, applying Lemma 4 in Juditsky et al. (2011)(as well as Lemma 7 in Zhang & Lan (2020)) on $\mathcal{D}_7$, we have

$$\mathcal{D}_7 = -\mathbf{E} \left\langle \frac{1}{S} \sum_{i \in \mathcal{S}_{t,k}} \mathbf{y}_{t,k+1}^{(i)} \left( \hat{G}_{t,k,1}^{(i)}(\tilde{\mathcal{B}}_{t,k}^{(i)}) + \frac{1}{\tau_{t,k}} (\mathbf{u}_{t,k} - \mathbf{u}_{t,0}) \right) - \frac{1}{n} \sum_{i=1}^{n} \bar{\mathbf{y}}_{t,k+1}^{(i)} \left( G_{t,k,1}^{(i)} + \frac{1}{\bar{\tau}_{t,k}} (\mathbf{u}_{t,k} - \mathbf{u}_{t,0}) \right), \, \mathbf{u}_{t,k+1} \right\rangle$$

$$\le \frac{\eta_t C_f^2 \sigma_1^2}{B} + \frac{\eta_t \delta^2}{S}. \tag{77}$$

Finally, for $\mathcal{D}_8$, similar to $\mathcal{D}_7$, it follows that

$$\mathcal{D}_8 = \frac{1}{n} \sum_{i=1}^{n} \mathbf{E} \left\langle \bar{\mathbf{y}}_{t,k+1}^{(i)} \left( \hat{G}_{t,k,2}^{(i)}(\tilde{\mathcal{B}}_{t,k}^{(i)}) - G_{t,k,2}^{(i)} \right), \, \mathbf{s}^{(i)} - \bar{\mathbf{s}}_{t,k+1}^{(i)} \right\rangle$$

$$= \frac{1}{n} \sum_{i=1}^{n} \mathbf{E} \left\langle \bar{\mathbf{y}}_{t,k+1}^{(i)} \left( \hat{G}_{t,k,2}^{(i)}(\tilde{\mathcal{B}}_{t,k}^{(i)}) - G_{t,k,2}^{(i)} \right), \, -\bar{\mathbf{s}}_{t,k+1}^{(i)} \right\rangle$$

$$\le \frac{\beta_t C_f^2 \sigma_2^2}{B}. \tag{78}$$

By setting $\lambda_1 = 1 + \frac{1}{S}$, $\lambda_2 = \frac{n}{S\alpha_t}$, $\lambda_3 = \alpha_t$, $\lambda_4 = 8C_g^2 \max\{\eta_t, \beta_t\}$, and $\lambda_5 = \lambda_6 = \frac{1}{8}$, and substituting the bounds of $\mathcal{D}_1$ through $\mathcal{D}_8$ into inequality (70), we obtain the desired result. $\qquad\square$

### C.3.3 Proof of Lemma C.9

*Proof.* For any $t \ge 0$, by invoking Lemma C.8, choosing $\mathbf{u} = \mathbf{u}_t^\dagger, \mathbf{s} = \mathbf{s}_t^\dagger$, and summing from $k = 0$ to $K_t - 1$ while taking expectation over $\mathcal{F}_{t,0}$, it holds that

$$\sum_{k=0}^{K_t-1} \mathbf{E} \left[ L_\gamma(\mathbf{u}_{t,k+1}, \mathbf{s}_{t,k+1}, \mathbf{y}; \mathbf{u}_{t,0}, \mathbf{s}_{t,0}) - L_\gamma(\mathbf{u}_t^\dagger, \mathbf{s}_t^\dagger, \bar{\mathbf{y}}_{t,k+1}; \mathbf{u}_{t,0}, \mathbf{s}_{t,0}) \right]$$

$$\le \frac{1}{2\eta_t} \left\| \mathbf{u}_t^\dagger - \mathbf{u}_{t,0} \right\|_2^2 + \frac{1}{2S\beta_t} \left\| \mathbf{s}_t^\dagger - \mathbf{s}_{t,0} \right\|_2^2 + \frac{1}{\alpha_t S} \left( \left\| \mathbf{y} - \mathbf{y}_{t,0} \right\|_2^2 + \left\| \mathbf{y} - \hat{\mathbf{y}}_{t,0} \right\|_2^2 + \left\| \mathbf{y} - \tilde{\mathbf{y}}_{t,0} \right\|_2^2 \right)$$

$$+ K_t \left( 64\Omega_t C_g^2 C_f^2 + \frac{S\alpha_t \sigma_0^2}{2Bn} + \frac{\alpha_t \sigma_0^2}{2B} + \frac{\eta_t C_f^2 \sigma_1^2}{B} + \frac{\beta_t C_f^2 \sigma_2^2}{B} + \frac{\eta_t \delta^2}{S} \right). \tag{79}$$

Since $L_\gamma(\mathbf{u}, \mathbf{s}, \mathbf{y}; \mathbf{u}', \mathbf{s}')$ is convex on $\mathbf{u}$ and $\mathbf{s}$, and linear on $\mathbf{y}$, we have

$$\max_{\mathbf{y} \in \mathcal{Y}} L_\gamma(\bar{\mathbf{u}}_t, \bar{\mathbf{s}}_t, \mathbf{y}; \mathbf{u}_{t,0}, \mathbf{s}_{t,0}) - L_\gamma(\mathbf{u}_t^\dagger, \mathbf{s}_t^\dagger, \bar{\bar{\mathbf{y}}}_t; \mathbf{u}_{t,0}, \mathbf{s}_{t,0})$$

$$\leq \max_{\mathbf{y} \in \mathcal{Y}} \frac{1}{K_t} \sum_{k=0}^{K_t-1} L_\gamma(\mathbf{u}_{t,k+1}, \mathbf{s}_{t,k+1}, \mathbf{y}; \mathbf{u}_{t,0}, \mathbf{s}_{t,0}) - L_\gamma(\mathbf{u}_t^\dagger, \mathbf{s}_t^\dagger, \bar{\mathbf{y}}_{t,k+1}; \mathbf{u}_{t,0}, \mathbf{s}_{t,0}), \tag{80}$$

where $\bar{\mathbf{u}}_t = \frac{1}{K_t} \sum_{k=0}^{K_t-1} \mathbf{u}_{t,k+1}, \bar{\mathbf{s}}_t = \frac{1}{K_t} \sum_{k=0}^{K_t-1} \mathbf{s}_{t,k+1}, \bar{\bar{\mathbf{y}}}_t = \frac{1}{T} \sum_{k=0}^{K_t-1} \bar{\mathbf{y}}_{t,k+1}$. Next, for the left-hand side (LHS), we have

$$L_\gamma(\bar{\mathbf{u}}_t, \bar{\mathbf{s}}_t, \mathbf{y}; \mathbf{u}_{t,0}, \mathbf{s}_{t,0}) - L_\gamma(\mathbf{u}_t^\dagger, \mathbf{s}_t^\dagger, \bar{\bar{\mathbf{y}}}_t; \mathbf{u}_{t,0}, \mathbf{s}_{t,0})$$

$$= \frac{1}{n} \sum_{i=1}^n \left( \mathbf{y}^{(i)} g_i(\bar{\mathbf{u}}_t, \bar{\mathbf{s}}_t^{(i)}) - f_i^*(\mathbf{y}^{(i)}) \right) + \frac{1}{2\gamma} \left\| \bar{\mathbf{u}}_t - \mathbf{u}_{t,0} \right\|_2^2 + \frac{1}{2n\gamma} \left\| \bar{\mathbf{s}}_t - \mathbf{s}_{t,0} \right\|_2^2$$

$$- \frac{1}{n} \sum_{i=1}^n \left( \bar{\bar{\mathbf{y}}}_t^{(i)} g_i(\mathbf{u}_t^\dagger, \mathbf{s}_t^{\dagger(i)}) - f_i^*(\bar{\bar{\mathbf{y}}}_t^{(i)}) \right) - \frac{1}{2\gamma} \left\| \mathbf{u}_t^\dagger - \mathbf{u}_{t,0} \right\|_2^2 - \frac{1}{2n\gamma} \left\| \mathbf{s}_t^\dagger - \mathbf{s}_{t,0} \right\|_2^2. \tag{81}$$

Choose $\mathbf{y}^{(i)} = \tilde{\mathbf{y}}_t^{(i)} \in \arg\max_{\mathbf{v}} \{\mathbf{v}^{(i)} g_i(\bar{\mathbf{u}}_t, \bar{\mathbf{s}}_t^{(i)}) - f_i^*(\mathbf{v}^{(i)})\}$, then we have $\mathbf{y}^{(i)} g_i(\bar{\mathbf{u}}_t, \bar{\mathbf{s}}_t^{(i)}) - f_i^*(\mathbf{y}^{(i)}) = f_i(g_i(\bar{\mathbf{u}}_t, \bar{\mathbf{s}}_t^{(i)}))$. By Fenchel-Young inequality, it holds that $\bar{\bar{\mathbf{y}}}_t^{(i)} g_i(\mathbf{u}_t^\dagger, \mathbf{s}_t^{\dagger(i)}) - f_i^*(\bar{\mathbf{s}}_t^{(i)}) \leq f_i(g_i(\mathbf{u}_t^\dagger, \mathbf{s}_t^{\dagger(i)}))$. Thus, we have $\Phi_\gamma(\bar{\mathbf{u}}_t, \bar{\mathbf{s}}_t; \mathbf{u}_{t,0}, \mathbf{s}_{t,0}) - \Phi_\gamma(\mathbf{u}_t^\dagger, \mathbf{s}_t^\dagger; \mathbf{u}_{t,0}, \mathbf{s}_{t,0}) \leq \max_{\mathbf{y}} \frac{1}{K_t} \sum_{k=0}^{K_t-1} L_\gamma(\mathbf{u}_{t,k+1}, \mathbf{s}_{t,k+1}, \mathbf{y}; \mathbf{u}_{t,0}, \mathbf{s}_{t,0}) - L_\gamma(\mathbf{u}_t^\dagger, \mathbf{s}_t^\dagger, \bar{\mathbf{y}}_{t,k+1}; \mathbf{u}_{t,0}, \mathbf{s}_{t,0})$. Dividing both sides by $K_t$ completes the proof. $\qquad\square$

### C.3.4  Proof of Theorem 4.5

*Proof.* We begin by invoking Lemma C.9, which yields:

$$\mathbf{E} F(\mathbf{u}_{t+1}, \mathbf{s}_{t+1}) \leq F(\mathbf{u}_t^\dagger, \mathbf{s}_t^\dagger) - \frac{1}{2\gamma} \mathbf{E} \left\| \mathbf{u}_{t+1} - \mathbf{u}_t \right\|_2^2 - \frac{1}{2n\gamma} \mathbf{E} \left\| \mathbf{s}_{t+1} - \mathbf{s}_t \right\|_2^2 + \left( \frac{1}{2\eta_t K_t} + \frac{1}{2\gamma} \right) \left\| \mathbf{u}_t^\dagger - \mathbf{u}_t \right\|_2^2$$

$$+ \left( \frac{1}{2S\beta_t K_t} + \frac{1}{2n\gamma} \right) \left\| \mathbf{s}_t^\dagger - \mathbf{s}_t \right\|_2^2 + \frac{1}{\alpha_t S K_t} \left( \left\| \mathbf{y} - \mathbf{y}_{t,0} \right\|_2^2 + \left\| \mathbf{y} - \hat{\mathbf{y}}_{t,0} \right\|_2^2 + \left\| \mathbf{y} - \tilde{\mathbf{y}}_{t,0} \right\|_2^2 \right)$$

$$+ 64 \Omega_t C_g^2 C_f^2 + \frac{S\alpha_t \sigma_0^2}{2Bn} + \frac{\alpha_t \sigma_0^2}{2B} + \frac{\eta_t C_f^2 \sigma_1^2}{B} + \frac{\beta_t C_f^2 \sigma_2^2}{B} + \frac{\eta_t \delta^2}{S}. \tag{82}$$

Based on inequality (6) from Rafique et al. (2022), the following estimate holds:

$$\left\| \mathbf{u}_t^\dagger - \mathbf{u}_t \right\|_2^2 + \frac{1}{n} \left\| \mathbf{s}_t^\dagger - \mathbf{s}_t \right\|_2^2 - \left\| \mathbf{u}_{t+1} - \mathbf{u}_t \right\|_2^2 - \frac{1}{n} \left\| \mathbf{s}_{t+1} - \mathbf{s}_t \right\|_2^2$$

$$\leq \frac{1}{3} \left\| \mathbf{u}_t^\dagger - \mathbf{u}_t \right\|_2^2 + \frac{1}{3n} \left\| \mathbf{s}_t^\dagger - \mathbf{s}_t \right\|_2^2 + 4 \left\| \mathbf{u}_t^\dagger - \mathbf{u}_{t+1} \right\|_2^2 + \frac{4}{n} \left\| \mathbf{s}_t^\dagger - \mathbf{s}_{t+1} \right\|_2^2,$$

then

$$\mathbf{E} F(\mathbf{u}_{t+1}, \mathbf{s}_{t+1}) \leq F(\mathbf{u}_t^\dagger, \mathbf{s}_t^\dagger) + \left( \frac{1}{2\eta_t K_t} + \frac{1}{6\gamma} \right) \left\| \mathbf{u}_t^\dagger - \mathbf{u}_t \right\|_2^2 + \frac{2}{\gamma} \mathbf{E} \left\| \mathbf{u}_t^\dagger - \mathbf{u}_{t+1} \right\|_2^2$$

$$+ \left( \frac{1}{2S\beta_t K_t} + \frac{1}{6n\gamma} \right) \left\| \mathbf{s}_t^\dagger - \mathbf{s}_t \right\|_2^2 + \frac{2}{n\gamma} \mathbf{E} \left\| \mathbf{s}_t^\dagger - \mathbf{s}_{t+1} \right\|_2^2$$

$$+ \frac{3D_{\mathcal{Y}}^2}{\alpha_t S K_t} + 64 \Omega_t C_g^2 C_f^2 + \frac{S\alpha_t \sigma_0^2}{2Bn} + \frac{\alpha_t \sigma_0^2}{2B} + \frac{\eta_t C_f^2 \sigma_1^2}{B} + \frac{\beta_t C_f^2 \sigma_2^2}{B} + \frac{\eta_t \delta^2}{S}. \tag{83}$$

Next, we apply the strong convexity of the auxiliary potential function $\Phi_\gamma$, as established in Lemma C.5. Since $\Phi_\gamma(\mathbf{u}, \mathbf{s}; \mathbf{u}', \mathbf{s}')$ is $(\frac{1}{\gamma} - C_f \rho)$-strongly convex with respect to $\mathbf{u}$ and $(\frac{1}{n\gamma} - \frac{C_f \rho}{n})$-strongly convex with

respect to $\mathbf{s}$, it follows that:

$$(\frac{1}{2\gamma} - \frac{C_f \rho}{2})\mathbf{E}\left\|\mathbf{u}_t^\dagger - \mathbf{u}_{t+1}\right\|_2^2 + (\frac{1}{2n\gamma} - \frac{C_f \rho}{2n})\mathbf{E}\left\|\mathbf{s}_t^\dagger - \mathbf{s}_{t+1}\right\|_2^2$$

$$\leq \mathbf{E}\Phi_\gamma(\mathbf{u}_{t+1}, \mathbf{s}_{t+1}; \mathbf{u}_t, \mathbf{s}_t) - \Phi_\gamma(\mathbf{u}_t^\dagger, \mathbf{s}_t^\dagger; \mathbf{u}_t, \mathbf{s}_t)$$

$$\leq \frac{1}{2\eta_t K_t}\mathbf{E}\left\|\mathbf{u}_t^\dagger - \mathbf{u}_t\right\|_2^2 + \frac{1}{2S\beta_t K_t}\mathbf{E}\left\|\mathbf{s}_t^\dagger - \mathbf{s}_t\right\|_2^2 + \frac{3D_{\mathcal{Y}}^2}{\alpha_t S K_t}$$

$$+ 64\Omega_t C_g^2 C_f^2 + \frac{S\alpha_t \sigma_0^2}{2Bn} + \frac{\alpha_t \sigma_0^2}{2B} + \frac{\eta_t C_f^2 \sigma_1^2}{B} + \frac{\beta_t C_f^2 \sigma_2^2}{B} + \frac{\eta_t \delta^2}{S}. \tag{84}$$

Furthermore, using the assumption $\gamma \leq \frac{1}{2C_f \rho}$, the final descent bound simplifies to:

$$\mathbf{E}F(\mathbf{u}_{t+1}, \mathbf{s}_{t+1}) \leq F(\mathbf{u}_t^\dagger, \mathbf{s}_t^\dagger) + \left(\frac{9}{2\eta_t K_t} + \frac{1}{6\gamma}\right)\left\|\mathbf{u}_t^\dagger - \mathbf{u}_t\right\|_2^2 + \left(\frac{9}{2S\beta_t K_t} + \frac{1}{6n\gamma}\right)\left\|\mathbf{s}_t^\dagger - \mathbf{s}_t\right\|_2^2$$

$$+ \frac{27D_{\mathcal{Y}}^2}{\alpha_t S K_t} + 576\Omega_t C_g^2 C_f^2 + \frac{9S\alpha_t \sigma_0^2}{2Bn} + \frac{9\alpha_t \sigma_0^2}{2B} + \frac{9\eta_t C_f^2 \sigma_1^2}{B} + \frac{9\beta_t C_f^2 \sigma_2^2}{B} + \frac{9\eta_t \delta^2}{S}. \tag{85}$$

Since $F(\mathbf{u}_t^\dagger, \mathbf{s}_t^\dagger) \leq F(\mathbf{u}_t, \mathbf{s}_t) - \frac{1}{2\gamma}\left\|\mathbf{u}_t^\dagger - \mathbf{u}_t\right\|_2^2 - \frac{1}{2n\gamma}\left\|\mathbf{s}_t^\dagger - \mathbf{s}_t\right\|_2^2$, it follows that

$$\mathbf{E}F(\mathbf{u}_{t+1}, \mathbf{s}_{t+1}) \leq F(\mathbf{u}_t, \mathbf{s}_t) - \frac{1}{3\gamma}\left\|\mathbf{u}_t^\dagger - \mathbf{u}_t\right\|_2^2 + \frac{9}{2\eta_t K_t}\left\|\mathbf{u}_t^\dagger - \mathbf{u}_t\right\|_2^2 - \frac{1}{3n\gamma}\left\|\mathbf{s}_t^\dagger - \mathbf{s}_t\right\|_2^2 + \frac{9}{2S\beta_t K_t}\left\|\mathbf{s}_t^\dagger - \mathbf{s}_t\right\|_2^2$$

$$+ \frac{27D_{\mathcal{Y}}^2}{\alpha_t S K_t} + 576\Omega_t C_g^2 C_f^2 + \frac{9S\alpha_t \sigma_0^2}{2Bn} + \frac{9\alpha_t \sigma_0^2}{2B} + \frac{9\eta_t C_f^2 \sigma_1^2}{B} + \frac{9\beta_t C_f^2 \sigma_2^2}{B} + \frac{9\eta_t \delta^2}{S}. \tag{86}$$

Summing both sides over $t = 0, 1, \ldots, T-1$ and taking expectation with respect to $\mathcal{F}_{0,0}$, we conclude:

$$\frac{1}{T}\sum_{t=0}^{T-1}\left((\frac{1}{3\gamma} - \frac{9}{2\eta_t K_t})\mathbf{E}\left\|\mathbf{u}_t^\dagger - \mathbf{u}_t\right\|_2^2 + (\frac{1}{3n\gamma} - \frac{9}{2S\beta_t K_t})\mathbf{E}\left\|\mathbf{s}_t^\dagger - \mathbf{s}_t\right\|_2^2\right)$$

$$\leq \frac{F(\mathbf{u}_0, \mathbf{s}_0) - \mathbf{E}F(\mathbf{u}_T, \mathbf{s}_T)}{T} + \frac{27D_{\mathcal{Y}}^2}{\alpha_t S K_t} + 576\Omega_t C_g^2 C_f^2$$

$$+ \frac{9S\alpha_t \sigma_0^2}{2Bn} + \frac{9\alpha_t \sigma_0^2}{2B} + \frac{9\eta_t C_f^2 \sigma_1^2}{B} + \frac{9\beta_t C_f^2 \sigma_2^2}{B} + \frac{9\eta_t \delta^2}{S}. \tag{87}$$

Finally, choosing the step sizes and inner-loop iteration numbers appropriately as:

$$\eta_t \asymp \min\{\frac{\epsilon^2}{C_g^2 C_f^2}, \frac{B\epsilon^2}{C_f^2 \sigma_1^2}, \frac{S\epsilon^2}{\delta^2}, \gamma\epsilon^2\}$$

$$\beta_t \asymp \min\{\frac{\epsilon^2}{C_g^2 C_f^2}, \frac{B\epsilon^2}{C_f^2 \sigma_2^2}, \gamma\epsilon^2\}$$

$$\alpha_t \asymp \frac{B\epsilon^2}{\sigma_0^2}$$

$$K_t \asymp \max\{\frac{D_{\mathcal{Y}}^2 \sigma_0^2}{BS\epsilon^4}, \frac{\gamma}{\eta_t}, \frac{n\gamma}{S\beta_t}\}, \tag{88}$$

we arrive at the desired convergence rate:

$$\mathbf{E}\left[\text{dist}(\mathbf{0}, \partial F(\mathbf{u}_t^\dagger, \mathbf{s}_t^\dagger))^2\right] \leq \frac{1}{\gamma^2 T}\sum_{t=0}^{T-1}\mathbf{E}\left(\left\|\mathbf{u}_t^\dagger - \mathbf{u}_t\right\|_2^2 + \frac{1}{n}\left\|\mathbf{s}_t^\dagger - \mathbf{s}_t\right\|_2^2\right)$$

$$\leq \frac{6\left[F(\mathbf{u}_0, \mathbf{s}_0) - \mathbf{E}F(\mathbf{u}_T, \mathbf{s}_T)\right]}{\gamma T} + \mathcal{O}(\gamma^{-1}\epsilon^2), \tag{89}$$

where $\bar{t}$ is uniformly sampled from $\{0, 1, \cdots, T-1\}$. Then we can make $\mathbf{E}\left[\operatorname{dist}(\mathbf{0}, \partial F(\mathbf{u}_{\bar{t}}^{\dagger}, \mathbf{s}_{\bar{t}}^{\dagger}))\right] \leq \epsilon$ by choosing $T = \mathcal{O}\left(\frac{F(\mathbf{u}_0, \mathbf{s}_0) - \inf_{\mathbf{u}, \mathbf{s}} F(\mathbf{u}, \mathbf{s})}{\epsilon^2}\right)$. The total iteration complexity would be

$$\sum_{t=0}^{T-1} K_t = \mathcal{O}\left(\frac{C_g^2 C_f^2}{\epsilon^4} + \frac{C_f^2 \sigma_1^2}{B\epsilon^4} + \frac{\delta^2}{S\epsilon^4} + \frac{nC_g^2 C_f^2}{S\epsilon^4} + \frac{nC_f^2 \sigma_2^2}{BS\epsilon^4} + \frac{D_{\mathcal{Y}}^2 \sigma_0^2}{BS\epsilon^6}\right) \tag{90}$$

$\square$

