# OpenReview forum: "Stochastic Primal-Dual Double Block-Coordinate for Two- way Partial AUC Maximization"
_TMLR — Accepted by TMLR_

### Review · Reviewer_4bQu · 2025-06-10

**Summary Of Contributions:**

This paper proposes two stochastic primal-dual double block-coordinate algorithms (STACO-1 and STACO-2) for solving the two-way partial AUC (TPAUC) maximization problem. The authors reformulate the original  objective as a minimax problem, allowing the use of primal-dual methods with alternating updates on blocks of primal and dual variables. The paper provides convergence guarantees for both convex and nonconvex settings, and empirical evaluations on multiple benchmark datasets demonstrate the effectiveness of the proposed approaches.

**Audience:**

Yes

**Claims And Evidence:**

Yes

**Requested Changes:**

Please see above.

**Strengths And Weaknesses:**

Strengths:

* The paper is clearly written and well-structured, making the main contributions easy to understand.

* The proposed algorithms are novel and tailored to the structure of the TPAUC problem. They make clever use of primal-dual and block-coordinate techniques, which enable a parallel speed-up and are supported by solid theoretical analysis.

* The authors provide both theoretical convergence results and empirical evaluations, offering a comprehensive view of the method's performance.

Weaknesses:

* The main algorithmic design appears to be inspired by the proximal-point framework used in Rafique et al.'s work. It would strengthen the contribution to discuss and compare with other recent minimax algorithms, such as [1]. How would the modification of algorithms in [1] perform when applying to TPAUC?

* In Figure 1, across all datasets, there is an abrupt drop and subsequent rise in performance. This behavior is not discussed in the main text and warrants explanation—what causes such sharp changes?

* The comparison in terms of runtime is missing. Reporting the total optimization time would provide a clearer picture of practical efficiency.

* In Appendix, the authors claim that STACO-1 shows lower variance in training loss based on Figure 5. However, this claim is not convincingly supported by the plots, especially Figure 5(c), where STACO-1 seems to exhibit higher variance. Would you please clarify this observation?

[1] Li, J., Zhu, L., & So, A. M. C. (2025). Nonsmooth nonconvex–nonconcave minimax optimization: Primal–dual balancing and iteration complexity analysis. Mathematical Programming, 1-51.

---

> ### Author Response · Authors · 2025-07-20
>
> Thanks for your suggestions. The revision according to your suggestions will be highlighted in brown
>
> **Q1**
> **The main algorithmic design appears to be inspired by the proximal-point framework used in Rafique et al.'s work. It would strengthen the contribution to discuss and compare with other recent minimax algorithms, such as [1]. How would the modification of algorithms in [1] perform when applying to TPAUC?**
>
> A: Thanks for pointing out [1]. The main differences between our methods and [1] are:
>
> 1. **Setting**: PLDA is for deterministic optimization while our methods are for stochastic optimization. As we consider large-scale machine learning problems, deterministic algorithms are not efficient.
> 2. **Dual update**: In our problem, updating all coordinates of the dual variable is expensive. Hence, we need to perform stochastic coordinate update.
>
> Thad being said, the leverage of KL inequality for characterzing the dual function in [1] could be useful for developing efficient single-loop stochastic algorithms. We will consider this in the future work.
>
> We've cited and discussed [1] in Section 2 of the revision.
>
>
> **Q2**
> **In Figure 1, across all datasets, there is an abrupt drop and subsequent rise in performance. This behavior is not discussed in the main text and warrants explanation—what causes such sharp changes?**
>
> A: This is due to the step size schedule. We apply a decreasing step size strategy during training, where the learning rate is reduced by a factor of 10 at the 500th, 1500th, and 2500th iterations for all methods. The abrupt drop is caused by an initially large step size, and at the 500th iteration, after reducing the step size, the TPAUC metric increases again. This decreasing step size schedule is commonly used in practice.  We've added this explanation at the beginning of Section 5.2.
>
> **Q3**
> **The comparison in terms of runtime is missing. Reporting the total optimization time would provide a clearer picture of practical efficiency.**
>
>
> Q: We have added the comparison of the per-iteration running time of STACO, PAUCI, SONX, and SOTA in the following table. The unit is seconds.
>
> |Method\Dataset|molmuv|moltox21|nodulemnist3d|adrenalmnist3d|
> |:-:|:-:|:-:|:-:|:-:|
> |SOTA|14.80|8.01|2.02|2.23|
> |SONX|9.78|4.54|1.62|1.76|
> |PAUCI|12.54|5.72|2.51|2.74|
> |STACO|**8.72**|**3.96**|**1.40**|**1.48**|
>
> From the results, we can clearly see that our method demonstrates significant time efficiency over existing baselines. We've added this comparison to Section 5.4.
>
>
> **Q4**
> **In Appendix, the authors claim that STACO-1 shows lower variance in training loss based on Figure 5. However, this claim is not convincingly supported by the plots, especially Figure 5\(c\), where STACO-1 seems to exhibit higher variance. Would you please clarify this observation?**
>
> A: For most experiments shown in Figure 5, STACO-1 exhibits lower variance compared to SONX. The only exception is Figure 5c, which corresponds to optimizing the loss with $(0.5, 0.5)$ weights. We believe this is a limitation of the primal-dual algorithm, which involves two learning rates. In practical applications, improper tuning of these rates may lead to training instability. We've clarified this point in the Appendix.

---

### Review · Reviewer_3nWQ · 2025-06-17

**Summary Of Contributions:**

This manuscript presents two novel algorithms, STACO1 and STACO2, for TPAUC (two-way partial AUC) optimization. TPAUC has emerged as a useful alternative to AUC but has relatively fewer optimization methods in the literature especially on the stochastic front, and this paper aims to fill that gap by developing block-coordinate optimization methods. Both methods, one for the convex case and the other for the weakly convex case, are based on a recent min-max reformulation of the original TPAUC optimization problem, and improve on the respective theoretical state-of-the-art rates of prior algorithms. Some of the techniques used for the analysis could be useful in their own right. Finally, a substantial empirical evaluation of both proposed methods is given (for the both cases of linear/nonlinear predictors, on several datasets in either case), which showcases a variety of performance advantages that new methods offer over prior state-of-the-art.

**Audience:**

Yes

**Claims And Evidence:**

Yes

**Requested Changes:**

Overall, I am reasonably satisfied with the manuscript, but I believe the following modifications would strengthen the work in nontrivial ways and so I would like to ask the authors to implement most/all of these:

--- Larger-scale very imbalanced datasets: While the current dataset lineup appears to be consistent with some prior works in the AUC/TPAUC literature, the empirical section left me wondering as to why some of the well-established large-scale settings with imbalanced classes such as the click-through rate prediction task in e-commerce were not experimented with. The degree of imbalance in CTR prediction tasks is such that the motivation for TPAUC optimization is very strong. Further, the very large scale of several existing such datasets gives an opportunity to possibly showcase even greater efficiency advantage of the proposed block coordinate approach. So unless there is something that I'm missing here, I would request to add such an evaluation to further strengthen the paper --- e.g. I'd propose a dataset like Criteo.

--- Experimental ablation: It appears important, from the perspective of the theory of surrogate losses, to look into how the empirical results would change as a result of choosing a different \ell than squared hinge loss --- testing other nonlinearities as \ell would not only lead to different Lipschitz parameters etc, but might also conceivably result in distinct optimization dynamics. Again, if there is something that I'm missing in terms of background, e.g. if the squared hinge loss has been established as the clear winner among possible choices of \ell and other choices are somehow empirically less sensible for TPAUC optimization, please let me know --- but from a purely optimization standpoint, it is natural to wonder about this ablation. As another useful thing to try, different pairs of \theta_0, \theta_1 would be good to see: e.g. in the above-suggested Criteo dataset's case, the imbalance could be large enough to where more extreme values of \theta_0, \theta_1 than 50 or 75 would be the interesting ones.

--- Clarity: It would be good to improve the exposition at the beginning of Section 3, where relevant background (such as the reformulation of the problem in a min-max form) is given. In this case, only a brief recap of the reformulations of Zhu et al (2022) is given, which for a less-familiar reader may be too terse. Some further background would be good to briefly discuss right there for intuition such as: Any other reformulations that may have been used in the literature (and their relative advantages)? What's the role of \ell in theory and in practice wrt. the optimization methods in prior work?

**Strengths And Weaknesses:**

Strengths:

+ Improved theoretical rates of both STACO1 and STACO2 relative to prior work, that are achieved via analysis that as far as I could tell is meaningfully different/improved relative to past work. In particular, the nontrivial 1/B vs. 1/\sqrt{B} improvement is made, and some of the stronger boundedness assumptions from prior work are shaved off, in the non-convex case. And in the convex case, the improved technique helps tease out a bound whose leading term that matches an existing lower bound.

+ As one might expect from carefully designed block-coordinate methods, their improved empirical properties appear significant on the whole: as demonstrated in different comparisons in the experimental section, the proposed methods win out on one or more of the following, relative to prior methods, ---  training performance; generalization properties; speed/stability of convergence.

(Relatively minor) Weaknesses:

- The empirical section could use some beefing up in terms of the datasets/tasks tested, as well as testing with other surrogate losses and with other (\theta_0, \theta_1); see Requested Changes below. Furthermore (as a minor point), if one were to nitpick, one could say that in certain aspects, the performance advantage of the new methods over some of the older methods is not always as large: for instance, in the linear case, apparently PAUCI performs quite on par with STACO in Table 2.

- The presentation writing, as a reader from outside the AUC/TPAUC optimization area, is something that I found less clear, especially in terms of setting the stage at the beginning of the paper. To keep readers engaged, I would consider adding more details early on in the manuscript such as: Conceptually, what ideas are some of the previous state-of-the-art methods based on (current indications are a bit too brief)? For the cited state-of-the-art rates, which dependencies could one hope to improve and by how much, discussing relevant parallels/lower bounds from the general optimization literature whenever possible (e.g. B^{1/2} vs B; the \epsilon^{-6} rate in the non-convex setting)? More on this in Requested Changes.

---

> ### Author Response · Authors · 2025-07-20
>
> Thanks for your suggestions. The revision according to your suggestions will be highlighted in red
>
> **Q1**
> Thank you very much for the insightful suggestion. We agree that the Criteo dataset represents an excellent large-scale and highly imbalanced benchmark, where the advantages of TPAUC optimization are particularly well-motivated. In fact, we also considered evaluating our method on Criteo during our experiments.
>
> However, we encountered issues when attempting to download the dataset from Hugging Face(https://huggingface.co/datasets/criteo/CriteoClickLogs), the compressed .gz file appears to be corrupted or incomplete, and fails to decompress properly. We have noticed that other users have reported similar issues on the dataset's Hugging Face page, and at the time of submission, the problem had not been resolved. We have also reached out to the dataset uploader to inquire about this.
>
> If the issue is resolved in the near future, we would be very happy to include results on the Criteo dataset and compare our method with other baselines, as we agree this would further strengthen our empirical evaluation.
>
> Nevertheless, we would like to emphasize that this paper is theory-oriented, with a primary focus on developing theoretical results. The empirical studies serve as proof-of-concept validations.
>
> **Q2**
> We agree that the choice of surrogate loss $\ell$ is important. We have added a new ablation study on surrogate loss functions in Section 5.3. Specifically, we compare the testing performance using three different surrogate losses: square loss, squared hinge loss, and hinge loss. The results show that our algorithm STACO performs well and remains stable across different choices of surrogate losses, indicating that the surrogate loss has limited impact on the final performance. We try (0.5, 0.5) and (0.75, 0.75) for (\theta_0, \theta_1), and we agree that we have to choose extreme values when the imbalance is large enough. We would be very happy to try it on Criteo datase if the issue is resolved.
>
> **Q3**
> We have added an explanation at the beginning of Section 3 to better clarify the reformulation from equation (1) to equation (2), and from equation (2) to equation (3). To the best of our knowledge, the CVaR-based reformulation in equation (2) is the only exact reformulation available for the original TPAUC minimization problem. While some other works propose approximate formulations, they are not exact [1].
>
> Regarding the surrogate loss $\ell$, our method accommodates multiple choices, including square loss, hinge loss, and squared hinge loss. In terms of theory, prior works such as Zhu et al. (2022) support both smooth and non-smooth loss functions in their analysis. However, our method achieves better convergence rates. In practice, prior works (e.g., Zhu et al. (2022), Hu et al. (2023)) commonly use squared hinge loss in their experiments, and we follow this convention.
>
> [1] Huiyang Shao, Qianqian Xu, Zhiyong Yang, Shilong Bao, and Qingming Huang. Asymptotically unbiased instance-wise regularized partial AUC optimization: Theory and algorithm. Advances in Neural Information Processing Systems, 35:38667–38679, 2022.

---

### Review · Reviewer_YBcz · 2025-07-06

**Summary Of Contributions:**

Consider the problem of optimizing the two-way partial AUC (TPAUC). This paper adopts a surrogate loss approach and formulates it as a variant of a finite-sum coupled compositional stochastic optimization problem. Two algorithms are then proposed based on a primal-dual approach: STACO1 for the convex case and STACO2 for the non-convex case. Notably, the algorithms implement block coordinate updates for both the primal and dual variables. Convergence analyses for both algorithms are provided.

**Audience:**

Yes

**Broader Impact Concerns:**

This is a theoretical work, and I believe there are no significant concerns of this kind.

**Claims And Evidence:**

No

**Requested Changes:**

I do not believe this work is publishable after only one minor revision. The presentation of the submission does not appear to be ready for publication. True connections with existing work are neither adequately discussed nor clearly established. There are several gaps and points of confusion in the analysis, and the proofs are difficult to read. Therefore, rather than requesting a minor revision, I believe it would be more appropriate for the authors to withdraw the submission and resubmit it after making substantial revisions.

However, if the authors believe that the work can be made satisfactory with a minor revision, then I encourage them to address the weaknesses outlined above.

**Strengths And Weaknesses:**

**Strengths.**

It is notable that the proposed algorithms implement block coordinate updates for both the primal and dual variables.

**Weaknesses.**

1. **Connection to existing works.** The discussion of connections to existing work appears to be insufficient. Although related work is cited, the discussion remains superficial.
    - On p. 9, it is stated that “SOTA is quite similar to STACO2.” Is the difference merely that the latter adopts block coordinate updates on both the primal and dual variables? If so, is the analysis of STACO2 an extension of that of SOTA? Is the extension straightforward, or does it involve addressing non-trivial challenges?
    - The proofs in the appendix rely heavily on the analyses of Wang & Yang (2023), citing their work frequently. However, the paper lacks a discussion of the technical novelty in light of that prior work. It merely states that the formulation considered by Wang & Yang (2023) does not involve the parameter $s$, but it remains unclear whether this difference introduces any non-trivial challenges.
2. **Presentation.**
    - The English needs improvement. For example, there are phrases such as “convexity *to* $u$ and $s^{(i)}$” and “weak*ly* convexity.”
    - It would be beneficial if the authors could provide some high-level explanations, rather than simply presenting equations directly from their calculation sheets. In particular, since the work of Wang & Yang (2023) is cited several times, it would suffice to include a high-level explanation such as: “the proof essentially follows Wang & Yang (2023), with necessary modifications.”
    - The submission appears to have been completed several years ago and has not been revised since. For example, the paper by Wang & Yang (2023) has had five versions on arXiv, the latest of which was uploaded in 2025, but the authors cited only the first version from 2023.
    - The authors cited Rafique et al. (2018) and Rafique et al. (2022) separately, but according to the arXiv records, these are simply the first and most recent versions of the same work. Are there any essential differences between the two?
    - The proofs are nearly as readable as the authors’ raw calculation sheets. Please reorganize and revise them to improve clarity and readability.
    - “Parallel speed-up” is not a widely accepted or well-defined term. It should be introduced before being used.
    - Are the constraints on the parameters in Equation 2 the same as those in Equation 3?
    - The first equation in Section 4.3 is not a “regularized problem” but simply a function.
    - I don’t get the following statement:
        > Nevertheless, our algorithm does not depend on $τ_{(i)}$ as computing the gradient of $u$ and $s_{(i)}$ will remove $τ_{(i)}$.
    - Some explanations in Section C.3 regarding why an approximate stationary point of the Moreau envelope is meaningful should be moved to Section 4.3.
    - Equation 41 needs a citation.
    - The filtration $\mathcal{F}_t$ in Lemma C.4 is not defined.
    - The proof of Lemma C.4 cites Lemma 13 of Wang & Yang (2023), but that lemma is not proved by them; they, in turn, cite other works. It would be more appropriate to cite the original sources directly.
3. **Analysis.**
    - In Section 4, the authors do not analyze the algorithms for the original problem, but rather for a similar one that omits the parameter $s’$, stating that “the update of $s’$ is almost the same as $w$.” This is quite confusing, and it remains unclear whether there are any gaps in applying the analysis results to the original problem.
    - What is the purpose of defining the virtual sequences in Definition C.2? In particular, isn’t $\bar{y}_{t + 1}^{(i)}$ exactly $y_{t + 1}$?
    - The proof of Lemma C.4 cites Theorem 22 of Wang & Yang (2023) to bound $\mathcal{D}_2$, but the statement of that theorem provides an iteration complexity bound rather than a bound on $\mathcal{D}_2$.
4. **Some other issues.**
    - The reformulation of Equation 2 into Equation 3 in Lemma 3.1 is not analogous to the reformulation of Equation 13 into Equation 14, which is confusing.
    - Moreover, the equivalence between Equation 2 and Equation 3 appears to follow directly from the definition of $[x]_+$, rather than from the convexity of $f_i$.
    - On p. 2, it is claimed that the algorithm of Hu et al. (2023) is designed for the non-convex case and does not apply to the convex case. Their algorithm assumes weak convexity, and standard convexity is an extreme case of weak convexity, so it appears strange that their algorithm does not apply to the convex case. Explanations are needed.
    - What is the purpose of showing the first two results in Equation 41? Doesn’t the third inequality suffice?

---

> ### Author Response · Authors · 2025-07-20
>
> Thanks for your suggestions. The revision according to your suggestions will be highlighted in blue. Due to the character limit, we have omitted the statement of each reviewer comment. Our responses follow the same order as the original comments.
>
> **Connection to existing works.**
>
> **1.** The key difference lies in the update strategy: STACO2 applies block coordinate updates to both the primal variable $\mathbf{s}$ and dual variable $\mathbf{y}$, while SOTA updates only $\mathbf{y}$, leading to higher computational cost when updating $\mathbf{s}$. Moreover, SOTA’s analysis is relatively rough, with a basic convergence rate of $\mathcal{O}(1/\epsilon^6)$ and no mini-batch speed-up. In contrast, our analysis for STACO2 is novel and more involved, achieving a rate of $\mathcal{O}(1/BS\epsilon^6)$ with demonstrated mini-batch acceleration. Despite algorithmic similarities, their design motivations and theoretical analyses are fundamentally different.
>
> **2.** First, Wang & Yang (2023) do not include the variable $\mathbf{s}$. Introducing $\mathbf{s}$ in our setting makes the analysis more challenging, especially since it is updated coordinate-wise, so techniques for $\mathbf{w}$ cannot be directly applied. Second, they focus only on the convex setting, whereas we address both convex and weakly-convex cases, where the latter introduces additional terms that require careful handling. While we leverage some lemmas from Wang & Yang (2023), our analysis is novel and does not simply follow their framework.
>
> **Presentation**
>
> **1.** Thanks for pointing this out. We've corrected these issues in the revised version.
>
> **2.** Thanks for the suggestion. We've added several high-level explanations in the proofs to improve clarity.
>
> **3.** Thanks for pointing this out. This work was conducted earlier this year, before the latest version of Wang & Yang (2025) was published. We have updated our citation to reflect the most recent version.
>
> **4.** No differences. We have updated the reference to cite the Journal verision.
>
> **5.** Thanks for the suggestion. We've added several explanations to the proofs to make them clearer and more understandable.
>
> **6.** Thanks for the suggestion. By “parallel speed-up,” we mean that the complexity is linearly dependent on both the positive and negative mini-batch sizes. We've added this explanation at the beginning of Section 2.
>
> **7.** Yes, Equations (2) and (3) are mathematically equivalent. We've added clarification on this reformulation at the beginning of Section 3 (please see the red-highlighted part).
>
> **8.** We have removed the term "regularized".
>
> **9.** In the proof, we introduced $\tau_{(i)}$ in our reformulation of (8) in (48) to obtain the convexity of
> $g_i(\mathbf{u}, \mathbf{s}^{(i)}) + \frac{1}{2\tau_{(i)}}\|\mathbf{u} - \mathbf{u}_{t,0}\|_2^2 + \frac{1}{2\tau_{(i)}}\|\mathbf{s}^{(i)} - \mathbf{s}^{(i)}_{t,0}\|_2^2$.
> However, in implementation, our algorithm does not need to set up $\tau_{(i)}$; it is simply for the sake of analysis.
>
> **10.** Thanks for the suggestion. We've made the corresponding modification.
>
> **11.** We've added the appropriate citation (now it is Equation (18)).
>
> **12.** We've added the definition of the filtration $\mathcal{F}_t$ before introducing Lemma C.4.
>
>
> **13.** Thanks for pointing this out. We've added the appropriate original citations.
>
> **Analysis.**
>
> **1.** We omit $s'$ in our proof since its update rule is the same as the primal variable $\mathbf{w}$, and even simpler as it is a scalar. Including $s'$ in the analysis would not alter the final convergence results. Therefore, we omit it to improve the clarity and readability of the proof.
>
> **2.** $\bar{y}_{t + 1}^{(i)}$ is different from $y_{t + 1}$ because in the virtual sequence, $\bar{y}$ is updated across all coordinates, while $y$ is only updated on the selected block.
>
> **3.** Thanks for pointing this out. In Theorem 22 of Wang & Yang (2023), they also derive a similar bound (see the last equation in their proof), and their result can be adapted to bound $\mathcal{D}_2$. To make this clearer, we have provided a more detailed explanation of how to bound $\mathcal{D}_2$ in the revised version.
>
> **Some other issues.**
>
> **1.** Sorry for the confusion. The transformations are essentially the same. We have added more details about these reformulations at the beginning of Section 3 (please see the red-highlighted part).
>
> **2.** Yes, it is.
>
> **3.** You are correct.  However, since their method does not exploit convexity, their algorithm and convergence guarantee still exhibit a complexity of  $O(1/\epsilon^6)$ even in the convex setting. We have revised the sentence correctly.
>
> **4.** Yes, in Equation 41 (now Equation (18) in the revised version), the third inequality alone is sufficient. We include the first two results because all three are classic and standard properties of the proximal point $\mathbf{x}^{\dagger}$, and we present them together for completeness.

---

> > ### Author Response · Authors · 2025-07-20
> >
> > We noticed some Markdown formatting issues in **Presentation 9.** and **Analysis 2.**, and have updated our responses to these items accordingly, and they now display correctly
> >
> > **Presentation**
> >
> > 9. In the proof, we introduced $\tau_{(i)}$ in our reformulation of (8) in (48) to obtain the convexity of
> > $g_i(\mathbf{u}, \mathbf{s}^{(i)}) + \frac{1}{2\tau_{(i)}}$$\|\mathbf{u} - \mathbf{u}\_{t,0}\|\_2^2$$ + \frac{1}{2\tau\_{(i)}}\|\mathbf{s}^{(i)} - \mathbf{s}^{(i)}\_{t,0}\|\_2^2$.
> > However, in implementation, our algorithm does not need to set up $\tau_{(i)}$; it is simply for the sake of analysis.
> >
> > **Analysis**
> >
> > 2. $\bar{y}\_{t + 1}^{(i)}$ is different from $y_{t + 1}$ because in the virtual sequence, $\bar{y}$ is updated across all coordinates, while $y$ is only updated on the selected block.

---

### Decision · Action_Editor_7wGy · 2025-08-27

**Recommendation:** Accept with minor revision

**Additional Comments:**

The authors consider the two-way partial AUC optimization and propose two novel algorithms based on the min-max reformulation through primal-dual structure. The authors provide both theoretical convergence guarantee and empirical evaluation on several benchmarks to demonstrate the effectiveness of the proposed algorithms.

Most of the reviewers acknowledge the contribution of the work, and recommend for acceptance. The major concerns raise in following several points:

- The related work is not inadequately discussed as pointed by Reviewer YBcz: "Please incorporate your answers about related work into the revision. Moreover, your answers, just like your original submission, only state that existing work provides inferior results. Please highlight the technical challenges and your key ideas for solving them".  Please consider the suggestion by the reviewer to "highlight the technical challenges and your key ideas for solving them" to emphasize the contribution.

- The presentation and organization can be further improved: " the proof of Lemma C.8 (Appendix C.3.2)" and "the technical novelty of the proposed methods" should be improved, as pointed by reviewers.

- Although all the reviewers mentioned about the concern on novelty (not uniformly novel with supporting lemmas lean on prior work), but "novelty of the studied method is not a necessary criteria for acceptance" for TMLR.

Please follow these suggestions from reviewers to improve the camera-ready version.

**Audience:**

Yes

**Audience Explanation:**

This paper will be interested to both machine learning and optimization community.

**Claims And Evidence:**

Yes

**Claims Explanation:**

The authors provide both theoretical convergence guarantee and empirical evaluation on several benchmarks to demonstrate the effectiveness of the proposed algorithms.